

# Algebraic Bethe ansatz for spinor R-matrices

**Vidas Regelskis**

Department of Physics, Astronomy and Mathematics, University of Hertfordshire,
Hatfield AL10 9AB, UK
Institute of Theoretical Physics and Astronomy, Vilnius University,
Saulėtekio av. 3, Vilnius 10257, Lithuania

vidas.regelskis@gmail.com

## Abstract

We present a supermatrix realisation of $q$-deformed spinor-spinor and spinor-vector $R$-matrices. These $R$-matrices are then used to construct transfer matrices for $U_{q^2}(\mathfrak{so}_{2n+1})$- and $U_q(\mathfrak{so}_{2n+2})$-symmetric closed spin chains. Their eigenvectors and eigenvalues are computed.



# 1 Introduction

In [1], Reshetikhin proposed a method of diagonalizing spin chain transfer matrices that obey quadratic relations defined by the $\mathfrak{so}_{2n+1}$- and $\mathfrak{so}_{2n}$-invariant spinor-spinor $R$-matrices. The key observation was that these matrices exhibit a nested six-vertex type structure thus allowing one to apply principles of the XXX Bethe ansatz at each level of the nesting. In the $\mathfrak{so}_{2n+1}$-invariant case the nesting truncates at the $\mathfrak{so}_3$-invariant spinor-spinor $R$-matrix which is equivalent to the well known Yang's $R$-matrix of the XXX spin chain. In the $\mathfrak{so}_{2n}$-invariant case the nesting truncates at the $\mathfrak{so}_4$-invariant spinor-spinor $R$-matrix which factorises into a tensor product of two Yang's $R$-matrices. It is important to note that the Lie algebra $\mathfrak{so}_{2n}$ has two spinor representations specified by the chirality property. As a consequence, there are four $\mathfrak{so}_{2n}$-invariant spinor-spinor $R$-matrices indexed by chirality of the corresponding spinor representations thus adding extra difficulties to the nesting procedure.

This diagonalization procedure was recently addressed in a new perspective in [2] by Karakhanyan and Kirschner. An important novelty in their work was that the spinor-spinor $R$-matrices were written in terms of the Euler Beta function rather than in terms of recurrent relations presented in [1] (see also [3]). The authors provided explicit examples of spinor-spinor $R$-matrices of low rank and commented on the corresponding cases of the algebraic Bethe ansatz. Similar spectral problems were also addressed by Reshetikhin in [4], De Vega and Karowski in [5], Babujian, Foerster and Karowski in [6,7], Ferrando, Frassek and Kazakov in [8], Liashyk and Pakuliak in [9], and Gerrard together with the author in [10].

In the present paper we address the long-standing problem of diagonalizing transfer matrices that obey quadratic relations defined by the $q$-deformed $\mathfrak{so}_{2n+1}$- and $\mathfrak{so}_{2n}$-invariant spinor-spinor $R$-matrices. We propose a new construction of spinor-spinor and spinor-vector $R$-matrices in terms of supermatrices (this replaces gamma matrices used in [1] and [2]) and provide explicit recurrence relations. These $R$-matrices are then used to construct spinor-type transfer matrices for $U_{q^2}(\mathfrak{so}_{2n+1})$- and $U_q(\mathfrak{so}_{2n})$-symmetric spin chains with twisted diagonal periodic boundary conditions. The deformation parameter in the former case is set to $q^2$ to avoid having $\sqrt{q}$ in the spinor-spinor $R$-matrix and the corresponding exchange relations. The square root of the deformation parameter arises because the root system of $\mathfrak{so}_{2n+1}$ has a short root. We then employ algebraic Bethe ansatz techniques similar to those in [1] to construct Bethe vectors and derive the corresponding Bethe ansatz equations. Our main results are stated in Theorems 3.3, 4.4 and 4.5.

The paper is organised as follows. Section 2 is devoted to the spinor $R$-matrices and various associated identities. Sections 3 and 4 contain the main results of the paper, diagonalization of the spinor-type transfer matrices. In Appendix A, we provide the semi-classical $q \to 1$ limit of the main results of this paper.

## 2 Spinor $R$-matrices

### 2.1 Matrices and supermatrices

Consider vector space $\mathbb{C}^N$ with $N \geq 3$. We will denote the standard basis vectors of $\mathbb{C}^N$ by $e_i$ and the standard matrix units of $\text{End}(\mathbb{C}^N)$ by $e_{ij}$ where indices $i, j$ are allowed to run from $-n$ to $n$ with $n = N \div 2$, and $0$ will only be included when $N$ is odd. We will use $\otimes$ to denote the usual tensor product over $\mathbb{C}$.

Next, consider vector superspace $\mathbb{C}^{1|1}$ with basis vectors $e_{-1}^{(1)}$ and $e_{+1}^{(1)}$. We will denote the standard matrix superunits of $\text{End}(\mathbb{C}^{1|1})$ by $e_{ij}^{(1)}$ where $i, j = \pm 1$. We define a $\mathbb{Z}_2$-grading on $\mathbb{C}^{1|1}$ by $\deg(e_i^{(1)}) = (1+i)/2$, and on $\text{End}(\mathbb{C}^{1|1})$ by $\deg(e_{ij}^{(1)}) = (1-ij)/2$. We also define a mapping $\gamma$ on $\text{End}(\mathbb{C}^{1|1})$ via $\gamma(e_{ij}^{(1)}) = ij\, e_{ij}^{(1)}$.

For any $n \geq 2$ we set $\mathbb{C}^{n|n} := (\mathbb{C}^{1|1})^{\hat{\otimes} n}$ where $\hat{\otimes}$ denotes a graded tensor product over $\mathbb{C}$, that is

$$(1 \,\hat{\otimes}\, e_j^{(1)})(e_i^{(1)} \,\hat{\otimes}\, 1) = (-1)^{\deg(e_j^{(1)})\deg(e_i^{(1)})} e_i^{(1)} \,\hat{\otimes}\, e_j^{(1)}. \tag{1}$$

We will write matrix superunits of $\text{End}(\mathbb{C}^{n|n})$ as

$$e_{\boldsymbol{ij}}^{(n)} := e_{i_1 j_1}^{(1)} \,\hat{\otimes}\, \cdots \,\hat{\otimes}\, e_{i_n j_n}^{(1)} \quad \text{with} \quad \boldsymbol{i}, \boldsymbol{j} \in (\pm 1, \ldots, \pm 1).$$

The degree of $e_{\boldsymbol{ij}}^{(n)}$ is $\deg(e_{\boldsymbol{ij}}^{(n)}) = (1 - \theta_{\boldsymbol{ij}})/2$ and $\gamma(e_{\boldsymbol{ij}}^{(n)}) = \theta_{\boldsymbol{ij}}\, e_{\boldsymbol{ij}}^{(n)}$ where $\theta_{\boldsymbol{ij}} = \theta_{\boldsymbol{i}}\theta_{\boldsymbol{j}}$ with $\theta_{\boldsymbol{i}} = i_1 i_2 \cdots i_n$. We will write supermatrices in $\text{End}(\mathbb{C}^{n|n})$ as

$$A^{(n)} = \sum_{\boldsymbol{i},\boldsymbol{j}} a_{\boldsymbol{ij}}\, e_{\boldsymbol{ij}}^{(n)} := \sum_{i_1, j_1, \ldots, i_n, j_n = \pm 1} a_{i_1, j_1, \ldots, i_n, j_n}\, e_{i_1 j_1}^{(1)} \,\hat{\otimes}\, \cdots \,\hat{\otimes}\, e_{i_n j_n}^{(1)},$$

where $a_{i_1, j_1, \ldots, i_n, j_n} \in \mathbb{C}$ are the matrix entries of $A^{(n)}$. In will be often convenient to write supermatrices in a nested form

$$A^{(n)} = \sum_{i,j=\pm 1} \left[ A^{(n)} \right]_{ij} \,\hat{\otimes}\, e_{ij}^{(1)}, \tag{2}$$

where $\left[ A^{(n)} \right]_{ij} \in \text{End}(\mathbb{C}^{n-1|n-1})$ are sub-supermatrices of $A^{(n)}$ given by

$$\left[ A^{(n)} \right]_{ij} = \sum_{i_1, j_1, \ldots, i_{n-1}, j_{n-1} = \pm 1} a_{i_1, j_1, \ldots, i_{n-1}, j_{n-1}, i, j}\, e_{i_1 j_1}^{(1)} \,\hat{\otimes}\, \cdots \,\hat{\otimes}\, e_{i_{n-1} j_{n-1}}^{(1)}.$$

We will sometimes adopt the notation

$$A^{(n-1)} := [A^{(n)}]_{-1,-1}, \qquad B^{(n-1)} := [A^{(n)}]_{-1,+1},$$
$$C^{(n-1)} := [A^{(n)}]_{+1,-1}, \qquad D^{(n-1)} := [A^{(n)}]_{+1,+1},$$

which will be used to denote the A, B, C, and D operators of the algebraic Bethe ansatz.

For any non-zero scalar $q$ we define a graded $q$-transposition $w$ on $\text{End}(\mathbb{C}^{n|n})$ via

$$(e_{\boldsymbol{ij}}^{(n)})^w := \theta_{\boldsymbol{ij}}\, q^{\vartheta_{\boldsymbol{i}} - \vartheta_{\boldsymbol{j}}}\, \overline{e_{-\boldsymbol{j},-\boldsymbol{i}}^{(n)}}, \tag{3}$$

where $\vartheta_{\boldsymbol{i}} = \sum_{p=1}^n (p - \frac{1}{2}) i_p$ and the overline means that the order of multiplying tensorands is reversed resulting in an overall sign; for instance,

$$\overline{e_{\boldsymbol{ij}}^{(2)}} = \overline{e_{i_1 j_1}^{(1)} \,\hat{\otimes}\, e_{i_2 j_2}^{(1)}} = (1 \,\hat{\otimes}\, e_{i_2 j_2}^{(1)})(e_{i_1 j_1}^{(1)} \,\hat{\otimes}\, 1) = (-1)^{\deg(e_{i_1 j_1}^{(1)})\deg(e_{i_2 j_2}^{(1)})} e_{\boldsymbol{ij}}^{(2)}.$$

The inverse of $w$ will be denoted by $\bar{w}$.

We define a linear map $\chi^{(n)} : \mathrm{End}(\mathbb{C}^{n|n}) \to (\mathbb{C}^{n|n})^* \otimes (\mathbb{C}^{n|n})^*$ via

$$\chi^{(n)}(e_{ij}^{(n)}) = c_{ij}\, \theta_{-i}\, q^{-\vartheta_i}\, e_{-i}^{(n)*} \otimes e_j^{(n)*}, \tag{4}$$

where $e_{-i}^{(n)*}$ and $e_j^{(n)*}$ are elementary supervectors in the dual superspaces and $c_{ij}$ is a grading factor defined recurrently via $c_{i_1\ldots i_n j_1\ldots j_n} = (-i_n)^n((-1)^{n-1} j_1 \cdots j_{n-1})^{\delta_{i_n,-j_n}} c_{i_1\ldots i_{n-1} j_1\ldots j_{n-1}}$ and $c_{i_1 j_1} = (-i_1)^1$. Then, given any $X, Y, Z \in \mathrm{End}(\mathbb{C}^{n|n})$, we have that

$$\chi^{(n)}(X^w Y Z) = \chi^{(n)}(Y)(\gamma(X) \otimes Z). \tag{5}$$

Let $V^{+(n-1)}$ and $V^{-(n-1)}$ denote the even- and odd-graded subspaces of $\mathbb{C}^{n|n}$, respectively. When $n = 2$, bbspace $V^{+(1)} \subset \mathbb{C}^{2|2}$ is spanned by vectors

$$e_{-1}^{(+)} := e_{-1}^{(1)} \hat{\otimes} e_{-1}^{(1)}, \qquad e_{+1}^{(+)} := e_{+1}^{(1)} \hat{\otimes} e_{+1}^{(1)},$$

and the odd-graded subspace $V^{-(1)} \subset \mathbb{C}^{2|2}$ is spanned by vectors

$$e_{-1}^{(-)} := e_{+1}^{(1)} \hat{\otimes} e_{-1}^{(1)}, \qquad e_{+1}^{(-)} := e_{-1}^{(1)} \hat{\otimes} e_{+1}^{(1)}.$$

When $n \geq 3$, the even-graded subspace $V^{+(n-1)} \subset \mathbb{C}^{n|n} \cong \mathbb{C}^{2|2} \hat{\otimes} (\mathbb{C}^{1|1})^{\hat{\otimes}(n-2)}$ is spanned by vectors

$$e_{i_1}^{(\pm)} \hat{\otimes} e_{i_2}^{(1)} \hat{\otimes} \cdots \hat{\otimes} e_{i_{n-1}}^{(1)},$$

with $i_1, \ldots, i_{n-1} = +1, -1$ such that $i_2 \cdots i_{n-1} = \pm(-1)^n$. Likewise, the odd-graded subspace $V^{-(n-1)} \subset \mathbb{C}^{n|n}$ is spanned by vectors of the same form except that $i_2 \cdots i_{n-1} = \mp(-1)^n$. Here $\pm$ and $\mp$ are linked with the plus-minus in $e_{i_1}^{(\pm)}$ stated in the formula above.

Define even- and odd-graded operators $e_{ij}^{(\pm)} \in \mathrm{End}(V^{\pm(1)})$ and $f_{ij}^{(\pm)} \in \mathrm{Hom}(V^{\pm(1)}, V^{\mp(1)})$ acting on vectors $e_i^{(\pm)}$ by

$$e_{ij}^{(\pm)} e_k^{(\pm)} = \delta_{jk} e_i^{(\pm)}, \qquad e_{ij}^{(\pm)} e_k^{(\mp)} = 0,$$
$$f_{ij}^{(\pm)} e_k^{(\pm)} = \delta_{jk} e_i^{(\mp)}, \qquad f_{ij}^{(\pm)} e_k^{(\mp)} = 0.$$

These operators allow us to write $A^{\pm(1)} \in \mathrm{End}(V^{\pm(1)})$ and $B^{\pm(1)} \in \mathrm{Hom}(V^{\pm(1)}, V^{\mp(1)})$ as

$$A^{\pm(1)} = \sum_{i,j=-1,+1} a_{ij}\, e_{ij}^{(\pm)}, \qquad B^{\pm(1)} = \sum_{i,j=-1,+1} b_{ij}\, f_{ij}^{(\pm)}.$$

We will write matrix operators $A^{\pm(n)} \in \mathrm{End}(V^{\pm(n)})$ and $B^{\pm(n)} \in \mathrm{Hom}(V^{\pm(n)}, V^{\mp(n)})$ when $n \geq 2$ as

$$A^{\pm(n)} = \sum_{i,j=+1,-1} [A^{\pm(n)}]_{ij} \,\hat{\otimes}\, e_{ij}^{(1)}, \qquad B^{\pm(n)} = \sum_{i,j=+1,-1} [B^{\pm(n)}]_{ij} \,\hat{\otimes}\, e_{ij}^{(1)},$$

where

$$[A^{\pm(n)}]_{-1,-1} \in \mathrm{End}(V^{\pm(n-1)}), \qquad [A^{\pm(n)}]_{-1,+1} \in \mathrm{Hom}(V^{\mp(n-1)}, V^{\pm(n-1)}),$$
$$[A^{\pm(n)}]_{+1,-1} \in \mathrm{Hom}(V^{\pm(n-1)}, V^{\mp(n-1)}), \qquad [A^{\pm(n)}]_{+1,+1} \in \mathrm{End}(V^{\mp(n-1)}),$$

and

$$[B^{\pm(n)}]_{-1,-1} \in \mathrm{Hom}(V^{\pm(n-1)}, V^{\mp(n-1)}), \qquad [B^{\pm(n)}]_{-1,+1} \in \mathrm{End}(V^{\mp(n-1)}),$$
$$[B^{\pm(n)}]_{+1,-1} \in \mathrm{End}(V^{\pm(n-1)}), \qquad [B^{\pm(n)}]_{+1,+1} \in \mathrm{Hom}(V^{\mp(n-1)}, V^{\pm(n-1)}).$$

We define a graded $q$-transposition $\mathit{uu}$ on $\mathrm{End}(V^{\pm(n)})$ and $\mathrm{Hom}(V^{\pm(n)}, V^{\mp(n)})$ via

$$(a_{ij}^{(\pm)} \,\hat{\otimes}\, e_{kl}^{(n-1)})^{\mathit{uu}} = (a_{ij}^{(\pm)})^{\mathit{uu}} \,\hat{\otimes}\, (e_{kl}^{(n-1)})^{\mathit{uu}}, \tag{6}$$

where $a \in \{e, f\}$ and

$$(e_{ij}^{(\pm)})^{\mathit{uu}} = ij\, q^{\frac{1}{2}(i-j)} e_{-j,-i}^{(\pm)}, \qquad (f_{ij}^{(\pm)})^{\mathit{uu}} = ij\, q^{\frac{1}{2}(i-j)} f_{-j,-i}^{(\mp)},$$
$$(e_{kl}^{(n-1)})^{\mathit{uu}} = \theta_{kl}\, q^{\xi_k - \xi_l}\, \overline{e_{-l,-k}^{(n-1)}}, \tag{7}$$

with $\xi_k = \sum_{p=1}^{n-1} \frac{1}{2}(p+1)k_p$. The inverse of $\mathit{uu}$ will be denoted by $\bar{\mathit{uu}}$.

We define a linear map $\chi^{\pm(n)} : \mathrm{Hom}(V^{\pm(n)}, V^{\mp(n)}) \to (V^{\pm(n)})^* \otimes (V^{\mp(n)})^*$ via

$$\chi^{\pm(n)}(a_{ij}^{(\pm)} \,\hat{\otimes}\, e_{kl}^{(n-1)}) = -i\, q^{-\frac{1}{2}i}\, c_{kl}^{\pm}\, \theta_{-k}\, q^{-\xi_k}\, e_{-k}^{(n)*} \,\hat{\otimes}\, a_{-i}^{(\pm)*} \otimes e_{j}^{(n)*} \,\hat{\otimes}\, b_{l}^{(\pm)*}, \tag{8}$$

where $a \in \{e, f\}$ and $b^{(\pm)} = e^{(\pm)}$ or $f^{(\mp)}$ if $a = e$ or $f$, respectively, and $c_{kl}^{\pm}$ is defined recurrently via $c_{k_1 \dots k_{n-1} l_1 \dots l_{n-1}}^{\pm} = \mp(-k_{n-1})^n (-k_1 \cdots k_{n-2} l_1 \cdots l_{n-2})^{\delta_{l_{n-1}, \mp 1}} c_{k_1 \dots k_{n-2} l_1 \dots l_{n-2}}^{-}$ with the base case $c_{k_1 l_1}^{\pm} = \mp(-k_1 l_1)^{\delta_{l_1, \mp 1}}$. Then, given any $Y^{\pm} \in \mathrm{Hom}(V^{\pm(n)}, V^{\mp(n)})$ and $X^{\pm}, Z^{\pm} \in \mathrm{End}(V^{\pm(n)})$, we have that

$$\chi^{\pm(n)}((X^{[\mp]})^{\mathit{uu}} Y^{\pm} Z^{\pm}) = \chi^{\pm(n)}(Y^{\pm})(X^{[\mp]} \otimes Z^{\pm}), \tag{9}$$

where $[\mp]$ is $\mp/\pm$ if $n$ is odd/even.

Lastly, for any matrix $X$ with entries $x_{ij}$ in an associative algebra $\mathcal{A}$ we write

$$X_s = \sum_{-n \le i,j \le n} I^{\otimes s-1} \otimes e_{ij} \otimes I^{\otimes m-s} \otimes x_{ij} \in \mathrm{End}(\mathbb{C}^N)^{\otimes m} \otimes \mathcal{A}, \tag{10}$$

where $I$ denotes the identity matrix and $m \in \mathbb{N}_{\ge 2}$ will always be clear from the context. Products of matrix operators will be ordered using the following rules:

$$\prod_{s=1}^{m} X_s = X_1 X_2 \cdots X_m \qquad \text{and} \qquad \prod_{s=m}^{1} X_s = X_m X_{m-1} \cdots X_1. \tag{11}$$

The standard multi-index ("multi-legged") generalisation of this notation will be used for both matrices and supermatrices.

## 2.2 Vector-vector $R$-matrix

Choose $q \in \mathbb{R}^{\times}$, not a root of unity, and set $\kappa = N/2 - 1$. Introduce a matrix-valued rational function, called the vector-vector $R$-matrix, by

$$R(u, v) := R_q + \frac{q - q^{-1}}{v/u - 1} P - \frac{q - q^{-1}}{q^{2\kappa} v/u - 1} Q_q, \tag{12}$$

where $R_q$, $P$ and $Q_q$ are matrix operators on $\mathbb{C}^N \otimes \mathbb{C}^N$ defined by

$$R_q := \sum_{-n \le i,j \le n} q^{\delta_{ij} - \delta_{i,-j}} e_{ii} \otimes e_{jj} + (q - q^{-1}) \sum_{-n \le i < j \le n} (e_{ij} \otimes e_{ji} - q^{\nu_i - \nu_j} e_{ij} \otimes e_{-i,-j}),$$
$$P := \sum_{-n \le i,j \le n} e_{ij} \otimes e_{ji}, \qquad Q_q := \sum_{-n \le i,j \le n} q^{\nu_i - \nu_j} e_{ij} \otimes e_{-i,-j}, \tag{13}$$

and the $N$-tuple $\nu$ is given by

$$(\nu_{-n}, \dots, \nu_n) := \begin{cases} (-n + \frac{1}{2}, -n + \frac{3}{2}, \dots, -\frac{1}{2}, 0, \frac{1}{2}, \dots, n - \frac{3}{2}, n - \frac{1}{2}) & \text{if } N = 2n+1, \\ (-n+1, -n+2, \dots, -1, 0, 0, 1, \dots, n-2, n-1) & \text{if } N = 2n. \end{cases} \tag{14}$$

The matrix $R(u,v)$, obtained by Jimbo in [11], is a solution of the quantum Yang-Baxter equation on $(\mathbb{C}^N)^{\otimes 3}$ with spectral parameters

$$R_{12}(u,v)R_{13}(u,w)R_{23}(v,w) = R_{23}(v,w)R_{13}(u,w)R_{12}(u,v), \tag{15}$$

where we have employed the multi-index extension of the notation (10).

## 2.3   Quantum loop algebra $U_q^{ex}(\mathfrak{Lso}_N)$

The vector-vector $R$-matrix can be used to define an extended quantum loop algebra of $\mathfrak{so}_N$ in the following way (see [12,13]). Introduce elements $\ell_{ij}^\pm[r]$ with $-n \le i,j \le n$ and $r \in \mathbb{Z}_{\ge 0}$, and combine them into formal series $\ell_{ij}^\pm(u) := \sum_{r \ge 0} \ell_{ij}^\pm[r]u^{\pm r}$, and collect into generating matrices

$$L^\pm(u) := \sum_{-n \le i,j \le n} e_{ij} \otimes \ell_{ij}^\pm(u). \tag{16}$$

The elements $\ell_{ii}^\pm[0]$ are invertible, and so are the $L^\pm(u)$. We will say that elements $\ell_{ij}^\pm[r]$ have degree $r$.

**Definition 2.1.** *The extended quantum loop algebra $U_q^{ex}(\mathfrak{Lso}_N)$ is the unital associative algebra with generators $\ell_{ij}^\pm[r]$ with $-n \le i,j \le n$ and $r \in \mathbb{Z}_{\ge 0}$, subject to the following relations:[1]*

$$\ell_{ii}^\pm[0]\ell_{ii}^\mp[0] = 1 \quad and \quad \ell_{ij}^-[0] = \ell_{ji}^+[0] = 0 \quad for \quad i < j \tag{17}$$

*and*

$$\begin{aligned} R(u,v)L_1^\pm(u)L_2^\pm(v) &= L_2^\pm(v)L_1^\pm(u)R(u,v), \\ R(u,v)L_1^+(u)L_2^-(v) &= L_2^-(v)L_1^+(u)R(u,v). \end{aligned} \tag{18}$$

*The Hopf algebra structure is given by*

$$\Delta : \ell_{ij}^\pm(u) \mapsto \sum_k \ell_{ik}^\pm(u) \otimes \ell_{kj}^\pm(u), \qquad S : L^\pm(u) \mapsto L^\pm(u)^{-1}, \qquad \epsilon : L^\pm(u) \mapsto I. \tag{19}$$

The degree zero elements $\ell_{ij}^\pm[0]$ generate the subalgebra $U_q^{ex}(\mathfrak{so}_N) \subset U_q^{ex}(\mathfrak{Lso}_N)$. In this work we focus on the spinor representation of $U_q^{ex}(\mathfrak{so}_N)$ which will be used to construct spinor-spinor and spinor-vector $R$-matrices. We will make use of the $q$-Clifford algebra realisation of $U_q^{ex}(\mathfrak{so}_N)$, see [14]. This realisation factors through the non-extended subalgebra $U_q(\mathfrak{so}_N) \subset U_q^{ex}(\mathfrak{so}_N)$ in which the symmetry relation $\ell_{-i,-i}^\pm[0]\ell_{ii}^\pm[0] = 1$ holds. We will denote generators of the latter subalgebra by $\ell_{ij}^\pm$ and the generating matrices by $L^\pm$.

**Definition 2.2.** *The $q$-Clifford algebra $\mathscr{C}_q^n$ is the unital associative algebra with generators $a_i$, $a_i^\dagger$, $\omega_i$, $\omega_i^{-1}$ with $1 \le i \le n$ satisfying*

$$\omega_i\omega_j = \omega_j\omega_i, \qquad \omega_i\omega_i^{-1} = \omega_i^{-1}\omega_i = 1, \tag{20}$$

$$\omega_i a_j \omega_i^{-1} = q^{\delta_{ij}}a_j, \qquad \omega_i a_j^\dagger \omega_i^{-1} = q^{-\delta_{ij}}a_j^\dagger, \tag{21}$$

$$a_i a_j + a_j a_i = 0, \qquad a_i^\dagger a_j^\dagger + a_j^\dagger a_i^\dagger = 0, \tag{22}$$

$$a_i a_j^\dagger + q^{\delta_{ij}} a_j^\dagger a_i = \delta_{ij}\omega_i^{-1}, \qquad a_i a_j^\dagger + q^{-\delta_{ij}} a_j^\dagger a_i = \delta_{ij}\omega_i. \tag{23}$$

Note that the relations (23), when $i = j$, are equivalent to

$$a_i^\dagger a_i = -\frac{\omega_i - \omega_i^{-1}}{q - q^{-1}}, \qquad a_i a_i^\dagger = \frac{q\,\omega_i - q^{-1}\omega_i^{-1}}{q - q^{-1}}. \tag{24}$$

---

[1] Our $U_q^{ex}(\mathfrak{Lso}_N)$ corresponds to $U(\overline{R})/\langle q^c = 1\rangle$ in [12].

The algebra $\mathscr{C}_q^n$ has a natural representation on the exterior algebra $\Lambda$ with generators $x_i$ with $1 \le i \le n$. For integers $\boldsymbol{m} = (m_1, \ldots, m_n) \in \mathbb{Z}^n$, we define an element $x(\boldsymbol{m})$ of $\Lambda$ as follows:

$$x(\boldsymbol{m}) = \begin{cases} x_1^{m_1} \wedge x_2^{m_2} \wedge \cdots \wedge x_n^{m_n} & \text{if } \boldsymbol{m} \in \{0,1\}^n, \\ 0 & \text{otherwise.} \end{cases}$$

The set $\{x(\boldsymbol{m}) : \boldsymbol{m} \in \{0,1\}^n\}$ is a basis of the vector space $\Lambda \cong \mathbb{C}^{n|n}$. Introduce elements $\boldsymbol{e}_i \in \mathbb{Z}_+^n$ defined by $\boldsymbol{e}_1 = (1, 0, \ldots, 0), \ldots, \boldsymbol{e}_n = (0, \ldots, 0, 1)$. The action of the algebra $\mathscr{C}_q^n$ on $\Lambda$ is given by

$$\begin{aligned} a_i(x(\boldsymbol{m})) &= (-1)^{m_1 + \ldots + m_{i-1}} x(\boldsymbol{m} - \boldsymbol{e}_i), \\ a_i^\dagger(x(\boldsymbol{m})) &= (-1)^{m_1 + \ldots + m_{i-1}} x(\boldsymbol{m} + \boldsymbol{e}_i), \\ \omega_i(x(\boldsymbol{m})) &= q^{-m_i} x(\boldsymbol{m}) \end{aligned} \tag{25}$$

for any $\boldsymbol{m} = (m_1, \ldots, m_n) \in \{0,1\}^n$. This turns $\Lambda$ into an irreducible $\mathscr{C}_q^n$-module.

Set $\deg(a_i) = \deg(a_i^\dagger) = 1$ and $\deg(\omega_i) = 0$, and extend this grading linearly on arbitrary monomials in $\mathscr{C}_q^n$. This defines a grading on $\mathscr{C}_q^n$. Denote by $\mathscr{C}_q^{n,+}$ the even-graded subalgebra of $\mathscr{C}_q^n$. Then the space $\Lambda$ splits into invariant subspaces, $\Lambda^+ = \{x(\boldsymbol{m}) : m_1 + \ldots + m_n \in 2\mathbb{Z}\}$ and $\Lambda^- = \{x(\boldsymbol{m}) : m_1 + \ldots + m_n + 1 \in 2\mathbb{Z}\}$, with respect to the action of $\mathscr{C}_q^{n,+}$.

**Proposition 2.3** ( [13]). *There exists an algebra homomorphism $\pi : U_q(\mathfrak{so}_N) \to \mathscr{C}_q^n$ defined by the following formulae:*

$$\begin{aligned} \ell_{00}^\pm &\mapsto 1, \qquad \ell_{i,i}^\pm \mapsto q^{\pm 1/2} \omega_i^{\pm 1}, \qquad \ell_{-i,-i}^\pm \mapsto q^{\mp 1/2} \omega_i^{\mp 1} && (i > 0), \\ \ell_{ij}^- &\mapsto (-1)^{i+j} q^{i-j-1/2}(q - q^{-1}) a_i^\dagger \omega_{i-1} \cdots \omega_{j+1} a_j \omega_j^{-1} && (i > j), \\ \ell_{ij}^+ &\mapsto -(-1)^{i+j} q^{i-j+3/2}(q - q^{-1}) \omega_i a_i^\dagger \omega_{i+1}^{-1} \cdots \omega_{j-1}^{-1} a_j && (i < j), \end{aligned}$$

*except $\ell_{ij}^\pm = 0$ if $i = -j \ne 0$, and we have assumed that*

$$\begin{aligned} \omega_0 &= q^{-1/2}, \qquad a_0 = (-1-q)^{-1/2}, \qquad a_0^\dagger = -q^{1/2}(-1-q)^{-1/2}, \\ \omega_{-i} &= q^{-1} \omega_i^{-1}, \qquad a_{-i} = q^{-1} a_i^\dagger, \qquad a_{-i}^\dagger = q\, a_i \qquad (i > 0). \end{aligned}$$

The mapping $\pi$ is a spinor representation of $U_q(\mathfrak{so}_N)$. In particular, the mapping $\pi$ turns $\Lambda$ into an irreducible $U_q(\mathfrak{so}_{2n+1})$-module with a highest vector $x(\boldsymbol{0})$ of weight

$$\lambda^\pm = (q^{\mp 1/2}, \ldots, q^{\mp 1/2}, 1, q^{\pm 1/2}, \ldots, q^{\pm 1/2}) \tag{26}$$

and $\Lambda^+$ (resp. $\Lambda^-$) into an irreducible $U_q(\mathfrak{so}_{2n})$-module with a highest vector $x(\boldsymbol{0})$ (resp. $x(\boldsymbol{e}_1)$) of weight

$$\lambda^\pm = (q^{\pm 1/2}, \ldots, q^{\pm 1/2}, q^{\mp 1/2}, \ldots, q^{\mp 1/2}), \tag{27}$$

$$\text{resp.} \quad \lambda^\pm = (q^{\pm 1/2}, \ldots, q^{\pm 1/2}, q^{\mp 1/2}, q^{\pm 1/2}, q^{\mp 1/2}, \ldots, q^{\mp 1/2}). \tag{28}$$

The spinor representation of $U_q(\mathfrak{so}_N)$ can be extended to a highest weight representation of the algebra $U_q^{ex}(\mathfrak{Lso}_N)$ by the rule

$$\pi_\rho : L^\pm(u) \mapsto \frac{\pi(q^{\pm 1/2} u^{\pm 1} L^\mp - q^{\mp 1/2} \rho^{\pm 1} L^\pm)}{u^{\pm 1} - \rho^{\pm 1}}, \tag{29}$$

for any $\rho \in \mathbb{C}^\times$, see [13].

## 2.4 Supermatrix representations of $\mathscr{C}_q^n$ and $\mathscr{C}_q^{n,+}$

We identify the space $\Lambda$ with $\mathbb{C}^{n|n}$ via the mapping

$$x(\boldsymbol{m}) \mapsto e_{2m_1-1}^{(1)} \hat{\otimes} \cdots \hat{\otimes} e_{2m_n-1}^{(1)}.$$

For instance, when $n = 2$, $\Lambda$ is identified with $\mathbb{C}^{2|2}$ via

$$\begin{aligned}
x(0,0) &\mapsto e_{-1}^{(1)} \hat{\otimes} e_{-1}^{(1)}, & x(0,1) &\mapsto e_{-1}^{(1)} \hat{\otimes} e_{+1}^{(1)}, \\
x(1,1) &\mapsto e_{+1}^{(1)} \hat{\otimes} e_{+1}^{(1)}, & x(1,0) &\mapsto e_{+1}^{(1)} \hat{\otimes} e_{-1}^{(1)}.
\end{aligned}$$

Let $\left(e_{ab}^{(1)}\right)_i$ denote the action of $e_{ab}^{(1)}$ on the $i$-th factor in the $n$-fold graded tensor product. Then it can be deduced from (25) that the mapping

$$\sigma \; : \; a_i \mapsto \left(e_{-1,+1}^{(1)}\right)_i, \quad a_i^\dagger \mapsto \left(e_{+1,-1}^{(1)}\right)_i, \quad \omega_i \mapsto \left(e_{-1,-1}^{(1)} + q^{-1}e_{+1,+1}^{(1)}\right)_i \tag{30}$$

defines a representation of $\mathscr{C}_q^n$ on $\mathbb{C}^{n|n}$.

When $n = 2$, we identify $\Lambda^+$ with the even-graded subspace $V^{+(1)} \subset \mathbb{C}^{2|2}$ via

$$x(0,0) \mapsto e_{-1}^{(+)}, \qquad x(1,1) \mapsto e_{+1}^{(+)},$$

and $\Lambda^+$ with the odd-graded subspace $V^{-(1)} \subset \mathbb{C}^{2|2}$ via

$$x(1,0) \mapsto e_{-1}^{(-)}, \qquad x(0,1) \mapsto e_{+1}^{(-)}.$$

When $n > 2$, we identify $\Lambda^+$ (resp. $\Lambda^-$) with the even- (resp. odd-) graded subspace $V^{\pm(n-1)} \subset \mathbb{C}^{n|n} \cong \mathbb{C}^{2|2} \hat{\otimes} (\mathbb{C}^{1|1})^{\hat{\otimes}\,n-2}$ via

$$x(\boldsymbol{m}) \mapsto \begin{cases} e_{2m_1-1}^{(+)} \hat{\otimes} e_{2m_3-1}^{(1)} \hat{\otimes} \cdots \hat{\otimes} e_{2m_n-1}^{(1)} & \text{if } m_1 = m_2, \\ e_{2m_2-1}^{(-)} \hat{\otimes} e_{2m_3-1}^{(1)} \hat{\otimes} \cdots \hat{\otimes} e_{2m_n-1}^{(1)} & \text{if } m_1 \neq m_2. \end{cases}$$

For instance, when $n = 3$, $\Lambda^+$ is identified with $V^{+(2)}$ via

$$\begin{aligned}
x(0,0,0) &\mapsto e_{-1}^{(+)} \hat{\otimes} e_{-1}^{(1)}, & x(1,0,1) &\mapsto e_{-1}^{(-)} \hat{\otimes} e_{+1}^{(1)}, \\
x(1,1,0) &\mapsto e_{+1}^{(+)} \hat{\otimes} e_{-1}^{(1)}, & x(0,1,1) &\mapsto e_{+1}^{(-)} \hat{\otimes} e_{+1}^{(1)},
\end{aligned}$$

and $\Lambda^-$ is identified with $V^{-(2)}$ via

$$\begin{aligned}
x(0,0,1) &\mapsto e_{-1}^{(+)} \hat{\otimes} e_{+1}^{(1)}, & x(1,0,0) &\mapsto e_{-1}^{(-)} \hat{\otimes} e_{-1}^{(1)}, \\
x(1,1,1) &\mapsto e_{+1}^{(+)} \hat{\otimes} e_{+1}^{(1)}, & x(0,1,0) &\mapsto e_{+1}^{(-)} \hat{\otimes} e_{-1}^{(1)}.
\end{aligned}$$

It follows from (25) that the mapping $\sigma^+ : \mathscr{C}_q^{n,+} \to \text{End}(V^{\pm(n-1)})$ given by

$$\begin{aligned}
a_1 a_2 &\mapsto -\left(e_{-1,+1}^{(+)}\right)_1, & a_1^\dagger a_2^\dagger &\mapsto \left(e_{+1,-1}^{(+)}\right)_1, & a_1 a_2^\dagger &\mapsto -\left(e_{+1,-1}^{(-)}\right)_1, & a_1^\dagger a_2 &\mapsto \left(e_{-1,+1}^{(-)}\right)_1, \\
a_i a_j &\mapsto \left(e_{-1,+1}^{(1)}\right)_{i-1}\left(e_{-1,+1}^{(1)}\right)_{j-1}, & a_i a_j^\dagger &\mapsto \left(e_{-1,+1}^{(1)}\right)_{i-1}\left(e_{+1,-1}^{(1)}\right)_{j-1}, \\
& & a_i^\dagger a_j^\dagger &\mapsto \left(e_{+1,-1}^{(1)}\right)_{i-1}\left(e_{+1,-1}^{(1)}\right)_{j-1}
\end{aligned}$$

and

$$a_1 a_j \mapsto \left(f^{(-)}_{-1,-1} + f^{(+)}_{+1,+1}\right)_1 \left(e^{(1)}_{-1,+1}\right)_{j-1}, \qquad a_1 a_j^\dagger \mapsto \left(f^{(-)}_{-1,-1} + f^{(+)}_{+1,+1}\right)_1 \left(e^{(1)}_{+1,-1}\right)_{j-1},$$

$$a_2 a_j \mapsto \left(f^{(-)}_{-1,+1} - f^{(+)}_{-1,+1}\right)_1 \left(e^{(1)}_{-1,+1}\right)_{j-1}, \qquad a_2 a_j^\dagger \mapsto \left(f^{(-)}_{-1,+1} - f^{(+)}_{-1,+1}\right)_1 \left(e^{(1)}_{+1,-1}\right)_{j-1},$$

$$a_1^\dagger a_j \mapsto \left(f^{(+)}_{-1,-1} + f^{(-)}_{+1,+1}\right)_1 \left(e^{(1)}_{-1,+1}\right)_{j-1}, \qquad a_1^\dagger a_j^\dagger \mapsto \left(f^{(+)}_{-1,-1} + f^{(-)}_{+1,+1}\right)_1 \left(e^{(1)}_{+1,-1}\right)_{j-1},$$

$$a_2^\dagger a_j \mapsto \left(f^{(+)}_{+1,-1} - f^{(-)}_{+1,-1}\right)_1 \left(e^{(1)}_{-1,+1}\right)_{j-1}, \qquad a_2^\dagger a_j^\dagger \mapsto \left(f^{(+)}_{+1,-1} - f^{(-)}_{+1,-1}\right)_1 \left(e^{(1)}_{+1,-1}\right)_{j-1},$$

and

$$\omega_1 \mapsto \left(e^{(+)}_{-1,-1} + q^{-1} e^{(+)}_{+1,+1} + q^{-1} e^{(-)}_{-1,-1} + e^{(-)}_{+1,+1}\right)_1,$$

$$\omega_2 \mapsto \left(e^{(+)}_{-1,-1} + q^{-1} e^{(+)}_{+1,+1} + e^{(-)}_{-1,-1} + q^{-1} e^{(-)}_{+1,+1}\right)_1,$$

$$\omega_i \mapsto \left(e^{(1)}_{-1,-1} + q^{-1} e^{(1)}_{+1,+1}\right)_{i-1},$$

for $3 \le i, j \le n$, defines a representation of $\mathscr{C}_q^{n,+}$ on $V^{\pm(n-1)}$.

## 2.5  Spinor-vector $R$-matrices

In the remaining parts of this paper we set the deformation parameter of $\mathfrak{so}_{2n+1}$ to $q^2$, that is, we will consider algebras $U^{ex}_{q^2}(\mathfrak{Lso}_{2n+1})$ and $U_{q^2}(\mathfrak{so}_{2n+1})$. This is to avoid having $\sqrt{q}$ in the spinor-spinor $R$-matrices (see Section 2.6) and the exchange relations (see Section 2.8).

We define the spinor-vector $R$-matrix of $U^{ex}_{q^2}(\mathfrak{Lso}_{2n+1})$ via the mapping $\pi_\rho$ composed with the representation $\sigma$ and a suitable transposition:

$$R^{(n)}(u,\rho) := \sum_{i,j} \left(\sigma \circ \pi_\rho(\ell^+_{-i,-j}(u))\right) \otimes e_{ij} = \sum_{i,j} \left(\sigma \circ \pi_\rho(\ell^-_{-i,-j}(u))\right) \otimes e_{ij}. \tag{31}$$

Our goal is to find a recurrence formula for $R^{(n)}(u,\rho)$. Introduce a rational function

$$f_q(v,u) := \frac{qv - q^{-1}u}{v - u}. \tag{32}$$

The Lemma below follows by directly evaluating (31).

**Lemma 2.4.** *The spinor-vector $R$-matrix of $U^{ex}_{q^2}(\mathfrak{Lso}_3)$ is an element of $\mathrm{End}(\mathbb{C}^{1|1} \otimes \mathbb{C}^3)$ given by*

$$R^{(1)}(u,\rho) = e^{(1)}_{-1,-1} \otimes \left(e_{-1,-1} + f_q(u,\rho) e_{00} + f_{q^2}(u,\rho) e_{11}\right)$$

$$+ \sqrt{-1}\sqrt{q + q^{-1}} \frac{q - q^{-1}}{u - \rho} \left(\sqrt{q}\, u\, e^{(1)}_{+1,-1} \otimes (e_{-1,0} - e_{01}) - \frac{\rho}{\sqrt{q}} e^{(1)}_{-1,+1} \otimes (e_{0,-1} - e_{10})\right)$$

$$+ e^{(1)}_{+1,+1} \otimes \left(f_{q^2}(u,\rho) e_{-1,-1} + f_q(u,\rho) e_{00} + e_{11}\right). \tag{33}$$

The Proposition below follows by an induction argument and lengthy but direct computations from (31). The base of induction is given by Lemma 2.4.

**Proposition 2.5.** *The spinor-vector $R$-matrix of $U^{ex}_{q^2}(\mathfrak{Lso}_{2n+1})$ for $n \ge 2$ is an element of the space $\mathrm{End}(\mathbb{C}^{n|n} \otimes \mathbb{C}^{2n+1})$ given by the following recurrence formula:*

$$R^{(n)}(u,\rho) = A^{(n-1)}(u,\rho)\,\hat{\otimes}\,e^{(1)}_{-1,-1} + B^{(n-1)}(u,\rho)\,\hat{\otimes}\,e^{(1)}_{-1,+1}$$

$$+ C^{(n-1)}(u,\rho)\,\hat{\otimes}\,e^{(1)}_{+1,-1} + D^{(n-1)}(u,\rho)\,\hat{\otimes}\,e^{(1)}_{+1,+1}, \tag{34}$$

*where*

$$A^{(n-1)}(u,\rho) = R^{(n-1)}(u,\rho) + I^{(n-1)} \otimes \left( e_{-n,-n} + f_{q^2}(u,\rho)\, e_{n,n} \right),$$

$$B^{(n-1)}(u,\rho) = q^{-\kappa}\rho\, \frac{q^2 - q^{-2}}{u - \rho} \sum_{ij} \sum_{k=0}^{n-1} \delta_{i_1,j_1}^{k,1} \cdots \delta_{i_{n-1},j_{n-1}}^{k,n-1} (-1)^{k+n+1} q^{i_k(k-3/2)} c_k$$
$$\times\, e_{i_1,j_1}^{(1)} \hat{\otimes} \cdots \hat{\otimes}\, e_{i_{n-1},j_{n-1}}^{(1)} \otimes \left( q^{-\sum_{l=k+1}^{n-1} i_l}\, e_{n,i_k k} - q^{\sum_{l=k+1}^{n-1} i_l}\, e_{-i_k k,-n} \right),$$

$$C^{(n-1)}(u,\rho) = q^{\kappa}u\, \frac{q^2 - q^{-2}}{u - \rho} \sum_{ij} \sum_{k=0}^{n-1} \delta_{i_1,j_1}^{k,1} \cdots \delta_{i_{n-1},j_{n-1}}^{k,n-1} (-1)^{k+n+1} q^{i_k(k-3/2)} c_k$$
$$\times\, e_{i_1,j_1}^{(1)} \hat{\otimes} \cdots \hat{\otimes}\, e_{i_{n-1},j_{n-1}}^{(1)} \otimes \left( q^{-\sum_{l=k+1}^{n-1} i_l}\, e_{-n,i_k k} - q^{\sum_{l=k+1}^{n-1} i_l}\, e_{-i_k k,n} \right),$$

$$D^{(n-1)}(u,\rho) = R^{(n-1)}(u,\rho) + I^{(n-1)} \otimes \left( f_{q^2}(u,\rho)\, e_{-n,-n} + e_{n,n} \right),$$

*with $\delta_{ij}^{kl} = (1 - \delta_{kl})\delta_{ij} + \delta_{kl}\delta_{i,-j}$, $i_0 = 1$, $c_0 = \frac{\sqrt{-1}\,q^{3/2}}{\sqrt{q+q^{-1}}}$ and $c_k = 1$ when $k \geq 1$. Here the End($\mathbb{C}^{2n+1}$)-valued leg of $R^{(n)}(u,\rho)$ is understood to be in the right-most space, that is,*

$$I^{(n-1)} \otimes \left( f_{q^2}(u,\rho)\, e_{-n,-n} + e_{n,n} \right) \hat{\otimes}\, e_{+1,+1}^{(1)} \equiv I^{(n-1)} \hat{\otimes}\, e_{+1,+1}^{(1)} \otimes \left( f_{q^2}(u,\rho)\, e_{-n,-n} + e_{n,n} \right).$$

The Lemma below follows directly from properties the $L$-operators $L^{\pm}(u)$ and (31).

**Lemma 2.6.** *The spinor-vector R-matrix of $U_{q^2}^{ex}(\mathfrak{Lso}_{2n+1})$ satisfies the equation*

$$R_{12}^{(n)}(u,\rho)R_{13}^{(n)}(v,\rho)R_{q^2,23}(v,u) = R_{q^2,23}(v,u)R_{13}^{(n)}(c,\rho)R_{12}^{(n)}(u,\rho),$$

*where $R_{q^2}(v,u)$ is obtained from (12) upon substituting $q \to q^2$.*

We define spinor-vector R-matrices of $U_q^{ex}(\mathfrak{Lso}_{2n+2})$ via the mapping $\pi_\rho$ composed with the representation $\sigma^+$ and a suitable transposition,

$$R^{\pm(n)}(u,\rho) := \sum_{i,j} \left( \sigma^+ \circ \pi_\rho(\ell_{-i,-j}^+(u)) \right)\Big|_{V^{\pm(n)}} \otimes e_{ij} = \sum_{i,j} \left( \sigma^+ \circ \pi_\rho(\ell_{-i,-j}^-(u)) \right)\Big|_{V^{\pm(n)}} \otimes e_{ij}, \quad (35)$$

where $|_{V^{\pm(n)}}$ denotes restriction to the corresponding $\mathscr{C}_q^{n+1,+}$-invariant subspace. The Lemma below follows by directly evaluating (35).

**Lemma 2.7.** *The spinor-vector R-matrices of $U_q^{ex}(\mathfrak{Lso}_4)$ are elements of $\text{End}(V^{\pm(1)} \otimes \mathbb{C}^4)$ given by*

$$R^{+(1)}(u,\rho) = e_{-1,-1}^{(+)} \otimes \left( e_{-2,-2} + e_{-1,-1} + f_q(u,\rho)(e_{11} + e_{22}) \right)$$
$$+ \frac{q - q^{-1}}{u - \rho}\left( q^{1/2} u\, e_{+1,-1}^{(+)} \otimes (e_{-2,1} - e_{-1,2}) + q^{-1/2}\rho\, e_{-1,+1}^{(+)} \otimes (e_{1,-2} - e_{2,-1}) \right)$$
$$+ e_{+1,+1}^{(+)} \otimes \left( f_q(u,\rho)(e_{-2,-2} + e_{-1,-1}) + e_{11} + e_{22} \right),$$

$$R^{-(1)}(u,\rho) = e_{-1,-1}^{(-)} \otimes \left( e_{-2,-2} + e_{11} + f_q(u,\rho)(e_{-1,-1} + e_{22}) \right)$$
$$- \frac{q - q^{-1}}{u - \rho}\left( q u\, e_{+1,-1}^{(-)} \otimes (e_{-2,-1} - e_{12}) + q^{-1}\rho\, e_{-1,+1}^{(-)} \otimes (e_{-1,-2} - e_{21}) \right)$$
$$+ e_{+1,+1}^{(-)} \otimes \left( f_q(u,\rho)(e_{-2,-2} + e_{11}) + e_{-1,-1} + e_{22} \right).$$

The Proposition below follows by an induction argument and lengthy but direct computations. The base of induction is given by Lemma 2.7.

**Proposition 2.8.** *The spinor-vector R-matrices of $U_q^{ex}(\mathfrak{Lso}_{2n+2})$ for $n \geq 2$ are elements of the space $\mathrm{End}(V^{\pm(n)} \otimes \mathbb{C}^{2n+2})$ given by following recurrence formulas:*

$$R^{\pm(n)}(u,\rho) = A^{\pm(n-1)}(u,\rho)\,\hat{\otimes}\,e^{(1)}_{-1,-1} + B^{\mp(n-1)}(u,\rho)\,\hat{\otimes}\,e^{(1)}_{-1,+1}$$
$$+ C^{\pm(n-1)}(u,\rho)\,\hat{\otimes}\,e^{(1)}_{+1,-1} + D^{\mp(n-1)}(u,\rho)\,\hat{\otimes}\,e^{(1)}_{+1,+1},$$

*where*

$$A^{\pm(n-1)}(u,\rho) = R^{\pm(n-1)}(u,\rho) + I^{\pm(n-1)} \otimes \left(e_{-n-1,-n-1} + f_q(u,\rho)\,e_{n+1,n+1}\right),$$

$$B^{\mp(n-1)}(u,\rho) = \varepsilon\, q^{-\frac{1}{4}(2\kappa+1)}\rho\,\frac{q-q^{-1}}{u-\rho}\left( \sum_i q^{\pm\frac{1}{4}\varepsilon\, i_1\cdots i_{n-1}}\, b_{i_1,i_1}\,\hat{\otimes}\,e^{(1)}_{i_2,i_2}\,\hat{\otimes}\cdots\hat{\otimes}\,e^{(1)}_{i_{n-1},i_{n-1}}\right.$$
$$\otimes\left(q^{-\frac{1}{2}\sum_{l=1}^{n-1}i_l}\,e_{n+1,\mp\varepsilon\, i_1\cdots i_{n-1}} - q^{\frac{1}{2}\sum_{l=1}^{n-1}i_l}\,e_{\pm\varepsilon\, i_1\cdots i_{n-1},-n-1}\right)$$
$$+ \sum_{ij}\sum_{k=1}^{n-1}\delta^{k,1}_{i_1,j_1}\cdots\delta^{k,n-1}_{i_{n-1},j_{n-1}}(i_1 j_1)^{\frac{1}{2}(1\mp1)}(\varepsilon\,\theta_i)^{\delta_{k1}}(-1)^k\, q^{\frac{1}{4}i_k(2k-1)}$$
$$\times b_{i_1,j_1}\,\hat{\otimes}\,e^{(1)}_{i_2,j_2}\,\hat{\otimes}\cdots\hat{\otimes}\,e^{(1)}_{i_{n-1},j_{n-1}}$$
$$\left.\otimes\left(q^{-\frac{1}{2}\sum_{l=k+1}^{n-1}i_l}\,e_{n+1,i_k(k+1)} - q^{\frac{1}{2}\sum_{l=k+1}^{n-1}i_l}\,e_{-i_k(k+1),-n-1}\right)\right),$$

$$C^{\pm(n-1)}(u,\rho) = \varepsilon\, q^{\frac{1}{4}(2\kappa+1)}u\,\frac{q-q^{-1}}{u-\rho}\left( \sum_i q^{\mp\frac{1}{4}\varepsilon\, i_1\cdots i_{n-1}}\, c_{i_1,i_1}\,\hat{\otimes}\,e^{(1)}_{i_2,i_2}\,\hat{\otimes}\cdots\hat{\otimes}\,e^{(1)}_{i_{n-1},i_{n-1}}\right.$$
$$\otimes\left(q^{-\frac{1}{2}\sum_{l=1}^{n-1}i_l}\,e_{-n-1,\pm\varepsilon\, i_1\cdots i_{n-1}} - q^{\frac{1}{2}\sum_{l=1}^{n-1}i_l}\,e_{\mp\varepsilon\, i_1\cdots i_{n-1},n+1}\right)$$
$$+ \sum_{ij}\sum_{k=1}^{n-1}\delta^{k,1}_{i_1,j_1}\cdots\delta^{k,n-1}_{i_{n-1},j_{n-1}}(i_1 j_1)^{\frac{1}{2}(1\pm1)}(\varepsilon\,\theta_i)^{\delta_{k1}}(-1)^k\, q^{\frac{1}{4}i_k(2k-1)}$$
$$\times c_{i_1,j_1}\,\hat{\otimes}\,e^{(1)}_{i_2,j_2}\,\hat{\otimes}\cdots\hat{\otimes}\,e^{(1)}_{i_{n-1},j_{n-1}}$$
$$\left.\otimes\left(q^{-\frac{1}{2}\sum_{l=k+1}^{n-1}i_l}\,e_{-n-1,i_k(k+1)} - q^{\frac{1}{2}\sum_{l=k+1}^{n-1}i_l}\,e_{-i_k(k+1),n+1}\right)\right),$$

$$D^{\mp(n-1)}(u,\rho) = R^{\mp(n-1)}(u,\rho) + I^{\mp(n-1)} \otimes \left(f_q(u,\rho)\,e_{-n-1,-n-1} + e_{n+1,n+1}\right),$$

*with $\delta^{kl}_{ij} = (1-\delta_{kl})\delta_{ij} + \delta_{kl}\delta_{i,-j}$ and $\varepsilon = (-1)^{n-1}$, and the type of opera-tors $b$ and $c$ is determined by requiring $B^{\mp(n-1)}(u,\rho) \in \mathrm{Hom}(V^{\mp(n-1)}, V^{\pm(n-1)})$ and $C^{\pm(n-1)}(u,\rho) \in \mathrm{Hom}(V^{\pm(n-1)}, V^{\mp(n-1)})$. For instance, when $n = 2$,*

$$B^{\mp(1)} = q^{-\frac{5}{4}}\rho\,\frac{q-q^{-1}}{u-\rho}\left(\pm q^{\pm\frac{1}{4}}f^{(\mp)}_{-1,-1}\otimes\left(q^{\frac{1}{2}}e_{3,\mp1} - q^{-\frac{1}{2}}e_{\pm1,-3}\right)\right.$$
$$\pm q^{-\frac{1}{4}}f^{(\mp)}_{-1,+1}\otimes\left(e_{3,-2} - e_{2,-3}\right)\mp q^{\frac{1}{4}}f^{(\mp)}_{+1,-1}\otimes\left(e_{32} - e_{-2,-3}\right)$$
$$\left.- q^{\mp\frac{1}{4}}f^{(\mp)}_{+1,+1}\otimes\left(q^{-\frac{1}{2}}e_{3,\pm1} - q^{\frac{1}{2}}e_{\mp1,-3}\right)\right),$$

$$C^{\pm(1)} = q^{\frac{5}{4}}u\,\frac{q-q^{-1}}{u-\rho}\left(-q^{\mp\frac{1}{4}}f^{(\pm)}_{-1,-1}\otimes\left(q^{\frac{1}{2}}e_{-3,\pm1} - q^{-\frac{1}{2}}e_{\mp1,3}\right)\right.$$
$$\mp q^{-\frac{1}{4}}f^{(\pm)}_{-1,+1}\otimes\left(e_{-3,-2} - e_{2,3}\right)\pm q^{\frac{1}{4}}f^{(\pm)}_{+1,-1}\otimes\left(e_{-3,2} - e_{-2,3}\right)$$
$$\left.- q^{\pm\frac{1}{4}}f^{(\pm)}_{+1,+1}\otimes\left(q^{-\frac{1}{2}}e_{-3,\mp1} - q^{\frac{1}{2}}e_{\pm1,3}\right)\right).$$

*Here the $\mathrm{End}(\mathbb{C}^{2n+2})$-valued leg of $R^{\pm(n)}(u,\rho)$ is understood to be in the right-most space.*

The Lemma below follows directly from properties of the $L$-operators $L^{\pm}(u)$ and (35).

**Lemma 2.9.** *The spinor-vector $R$-matrices of $U_q^{ex}(\mathfrak{Lso}_{2n+2})$ satisfy the equations*

$$R_{12}^{\pm(n)}(u,\rho)R_{13}^{\pm(n)}(v,\rho)R_{q^2,23}(v,u) = R_{q^2,23}(v,u)R_{13}^{\pm(n)}(v,\rho)R_{12}^{\pm(n)}(u,\rho).$$

## 2.6 Spinor-spinor $R$-matrices

We define the spinor-spinor $R$-matrix of $U_{q^2}^{ex}(\mathfrak{Lso}_{2n+1})$ as a $U_{q^2}^{ex}(\mathfrak{Lso}_{2n+1})$-equivariant map in the superspace $\mathrm{End}(\mathbb{C}^{n|n} \otimes \mathbb{C}^{n|n})$, i.e. it is a solution to the intertwining equation

$$\begin{aligned}
(\sigma \otimes \sigma) &\circ (\pi_v \otimes \pi_u)(\Delta'(\ell_{ij}^{\pm}(w)))R^{(n,n)}(u,v) \\
&= R^{(n,n)}(u,v)(\sigma \otimes \sigma) \circ (\pi_v \otimes \pi_u)(\Delta(\ell_{ij}^{\pm}(w))),
\end{aligned} \tag{36}$$

for all $-n \le i,j \le n$, where $\Delta'$ denotes the opposite coproduct. Our goal is to find a recurrence formula for $R^{(n,n)}(u,v)$. Introduce rational functions

$$\alpha(u,v) = \frac{v-u}{qv-q^{-1}u}, \qquad \beta(u,v) = \frac{q-q^{-1}}{qv-q^{-1}u}. \tag{37}$$

All the technical statements presented below are obtained using induction arguments and/or lengthy but direct computations. For instance, Lemma 2.10 follows by solving the intertwining equation (36) for $n = 1$. This Lemma then serves as the base of induction in verifying Proposition 2.12. We leave the technical details to an interested reader.

**Lemma 2.10.** *The spinor-spinor $R$-matrix of $U_{q^2}^{ex}(\mathfrak{Lso}_3)$ is an element of $\mathrm{End}(\mathbb{C}^{1|1} \otimes \mathbb{C}^{1|1})$ given by*

$$\begin{aligned}
R^{(1,1)}(u,v) = {}& e_{-1,-1}^{(1)} \otimes e_{-1,-1}^{(1)} + e_{11}^{(1)} \otimes e_{11}^{(1)} \\
&+ \alpha(u,v)(e_{-1,-1}^{(1)} \otimes e_{11}^{(1)} + e_{11}^{(1)} \otimes e_{-1,-1}^{(1)}) \\
&+ \beta(u,v)(v\, e_{-1,1}^{(1)} \otimes e_{1,-1}^{(1)} + u\, e_{1,-1}^{(1)} \otimes e_{-1,1}^{(1)}).
\end{aligned} \tag{38}$$

*Remark* 2.11. As an operator in $\mathscr{C}_{q^2}^1 \otimes \mathscr{C}_{q^2}^1$, the spinor-spinor $R$-matrix of $U_{q^2}(\mathfrak{Lso}_3)$ has the unique form

$$\begin{aligned}
\mathcal{R}^{(1,1)}(u,v) = {}& 1 - a_1^\dagger \omega_1 a_1 \otimes 1 - 1 \otimes a_1^\dagger \omega_1 a_1 + a_1^\dagger a_1 \otimes a_1^\dagger a_1 + a_1^\dagger \omega_1 a_1 \otimes a_1^\dagger \omega_1 a_1 \\
&+ \alpha(u,v)(a_1^\dagger \omega_1 a_1 \otimes \omega_1 + \omega_1 \otimes a_1^\dagger \omega_1 a_1 \\
&\qquad - q^{-2}a_1^\dagger a_1 \otimes a_1^\dagger \omega_1 a_1 - q^{-2}a_1^\dagger \omega_1 a_1 \otimes a_1^\dagger a_1) \\
&+ \beta(u,v)(v\, \omega_1 a_1 \otimes a_1^\dagger + u\, a_1^\dagger \omega_1 \otimes a_1).
\end{aligned}$$

When $n \ge 2$ the explicit form of $\mathcal{R}^{(n,n)}(u,v) \in \mathscr{C}_{q^2}^n \otimes \mathscr{C}_{q^2}^n$ is not unique, however the transition elements are unique in the sense that the image of $\mathcal{R}^{(n,n)}(u,v)$ in $\mathrm{End}(\mathbb{C}^{n|n} \otimes \mathbb{C}^{n|n})$ is unique.

**Proposition 2.12.** *The spinor-spinor $R$-matrix of $U_{q^2}^{ex}(\mathfrak{Lso}_{2n+1})$ when $n \ge 2$ in an element of the space $\mathrm{End}(\mathbb{C}^{n|n} \otimes \mathbb{C}^{n|n})$ given by the following recurrence formula:*

$$\begin{aligned}
R^{(n,n)}(u,v) = {}& R^{(n-1,n-1)}(u,v)\,\hat{\otimes}\,(e_{-1,-1}^{(1)} \otimes e_{-1,-1}^{(1)} + e_{11}^{(1)} \otimes e_{11}^{(1)}) \\
&+ \alpha(u,v)R^{(n-1,n-1)}(u,q^4v)\,\hat{\otimes}\,(e_{-1,-1}^{(1)} \otimes e_{11}^{(1)} + e_{11}^{(1)} \otimes e_{-1,-1}^{(1)}) \\
&+ \beta(u,v)U^{(n-1,n-1)}(u,q^4v)\,\hat{\otimes}\,(v\, e_{-1,1}^{(1)} \otimes e_{1,-1}^{(1)} + u\, e_{1,-1}^{(1)} \otimes e_{-1,1}^{(1)}),
\end{aligned} \tag{39}$$

where

$$U^{(n-1,n-1)}(u,v) := R^{(n-1,n-1)}(q^4,1)P'^{(n-1,n-1)}R^{(n-1,n-1)}(u,v) \tag{40}$$

and

$$P'^{(n-1,n-1)} := (\gamma \otimes id)(P^{(n-1,n-1)}) = (id \otimes \gamma)(P^{(n-1,n-1)}),$$

with $P^{(n-1,n-1)} := R^{(n-1,n-1)}(u,u)$, the permutation operator on $\mathbb{C}^{n|n} \otimes \mathbb{C}^{n|n}$.

**Lemma 2.13.** *The inverse of the spinor-spinor R-matrix of $U^{ex}_{q^2}(\mathfrak{L}\mathfrak{so}_{2n+1})$ is given by*

$$R^{(n,n)}_{q^{-1}}(u,v) = P^{(n,n)}R^{(n,n)}(v,u)P^{(n,n)} = (R^{(n,n)}(u,v))^{-1}. \tag{41}$$

*Moreover, the spinor-spinor R-matrix is crossing symmetric, that is*

$$(R^{(n,n)}(q^{4n-2}u,v))^{\bar{w}_1} = (R^{(n,n)}(q^{4n-2}u,v))^{w_2} = h^{(n)}(u,v)(R^{(n,n)}(u,v))^{-1}, \tag{42}$$

*with $h^{(n)}(u,v) := \prod_{j=1}^{n} \alpha(q^{4j-2}u,v)$ and the q-transposition w defined via (3).*

**Lemma 2.14.** *The spinor R-matrices of $U^{ex}_{q^2}(\mathfrak{L}\mathfrak{g}_{2n+1})$ satisfy the following quantum Yang-Baxter equations:*

$$R^{(n,n)}_{12}(u,v)R^{(n,n)}_{13}(u,w)R^{(n,n)}_{23}(v,w) = R^{(n,n)}_{23}(v,w)R^{(n,n)}_{13}(u,w)R^{(n,n)}_{12}(u,v), \tag{43}$$

$$R^{(n,n)}_{12}(u,v)R^{(n)}_{13}(u,\rho)R^{(n)}_{23}(v,\rho) = R^{(n)}_{23}(v,\rho)R^{(n)}_{12}(u,\rho)R^{(n,n)}_{12}(u,v). \tag{44}$$

We define the spinor-spinor R-matrices of $U^{ex}_q(\mathfrak{L}\mathfrak{so}_{2n+2})$ as $U^{ex}_q(\mathfrak{L}\mathfrak{so}_{2n+2})$-equivariant maps in the space $\mathrm{End}(V^{\epsilon_1(n)} \otimes V^{\epsilon_2(n)})$ with $\epsilon_1,\epsilon_2 = \pm$, i.e. they are solutions to the intertwining equation

$$(\sigma^+ \otimes \sigma^+) \circ (\pi_v \otimes \pi_u)(\Delta'(\ell^{\pm}_{ij}(w)))R^{\epsilon_1\epsilon_2(n,n)}(u,v)$$
$$= R^{\epsilon_1\epsilon_2(n,n)}(u,v)(\sigma^+ \otimes \sigma^+) \circ (\pi_v \otimes \pi_u)(\Delta(\ell^{\pm}_{ij}(w))) \tag{45}$$

for all $-n \le i, j \le n$.

**Lemma 2.15.** *The spinor-spinor R-matrices of $U^{ex}_q(\mathfrak{L}\mathfrak{so}_4)$ are elements of $\mathrm{End}(V^{\pm(1)} \otimes V^{\pm(1)})$ and $\mathrm{End}(V^{\pm(1)} \otimes V^{\mp(1)})$ given by*

$$R^{\pm\pm(1,1)}(u,v) = e^{(\pm)}_{-1,-1} \otimes e^{(\pm)}_{-1,-1} + e^{(\pm)}_{+1,+1} \otimes e^{(\pm)}_{+1,+1}$$
$$+ \alpha(u,v)(e^{(\pm)}_{-1,-1} \otimes e^{(\pm)}_{+1,+1} + e^{(\pm)}_{+1,+1} \otimes e^{(\pm)}_{-1,-1})$$
$$+ \beta(u,v)(v\, e^{(\pm)}_{-1,+1} \otimes e^{(\pm)}_{+1,-1} + u\, e^{(\pm)}_{+1,-1} \otimes e^{(\pm)}_{-1,+1}) \tag{46}$$

*and $R^{\pm\mp(1,1)}(u,v) = I^{\pm\mp(1,1)} := \sum_{i,j} e^{(\pm)}_{ii} \otimes e^{(\mp)}_{jj}$, the identity operator in $\mathrm{End}(V^{\pm(1)} \otimes V^{\mp(1)})$.*

**Lemma 2.16.** *The spinor-spinor R-matrices of $U^{ex}_q(\mathfrak{L}\mathfrak{so}_6)$ are elements of $\mathrm{End}(V^{\pm(2)} \otimes V^{\pm(2)})$ and $\mathrm{End}(V^{\pm(2)} \otimes V^{\mp(2)})$ given by*

$$R^{\pm\pm(2,2)}(u,v) = R^{\pm\pm(1,1)}(u,v)\,\hat{\otimes}\,(e^{(1)}_{-1,-1} \otimes e^{(1)}_{-1,-1}) + R^{\mp\mp(1,1)}(u,v)\,\hat{\otimes}\,(e^{(1)}_{+1,+1} \otimes e^{(1)}_{+1,+1})$$
$$+ \alpha(u,v)\Big(I^{\pm\mp(1,1)}\,\hat{\otimes}\,(e^{(1)}_{-1,-1} \otimes e^{(1)}_{+1,+1}) + I^{\mp\pm(1,1)}\,\hat{\otimes}\,(e^{(1)}_{+1,+1} \otimes e^{(1)}_{-1,-1})\Big)$$
$$- \beta(u,v)\Big(v\, F^{\mp\pm(1,1)}\,\hat{\otimes}\,(e^{(1)}_{-1,+1} \otimes e^{(1)}_{+1,-1}) + u\, F^{\pm\mp(1,1)}\,\hat{\otimes}\,(e^{(1)}_{+1,-1} \otimes e^{(1)}_{-1,+1})\Big), \tag{47}$$

$$R^{\pm\mp(2,2)}(u,v) = I^{\pm\mp(1,1)}\,\hat{\otimes}\,(e^{(1)}_{-1,-1} \otimes e^{(1)}_{-1,-1}) + I^{\mp\pm(1,1)}\,\hat{\otimes}\,(e^{(1)}_{+1,+1} \otimes e^{(1)}_{+1,+1})$$
$$+ R^{\pm\pm(1,1)}(u,q^2v)\,\hat{\otimes}\,(e^{(1)}_{-1,-1} \otimes e^{(1)}_{+1,+1}) + R^{\mp\mp(1,1)}(u,q^2v)\,\hat{\otimes}\,e^{(1)}_{+1,+1} \otimes e^{(1)}_{-1,-1})$$
$$- \frac{q-q^{-1}}{q^2v - q^{-2}u}\Big(v\, Q^{\mp\mp(1,1)}\,\hat{\otimes}\,(e^{(1)}_{-1,+1} \otimes e^{(1)}_{+1,-1}) + u\, Q^{\pm\pm(1,1)}\,\hat{\otimes}\,(e^{(1)}_{+1,-1} \otimes e^{(1)}_{-1,+1})\Big), \tag{48}$$

*where*

$$F^{\pm\mp(1,1)} := \sum_{i,j} f_{ij}^{(\pm)} \otimes f_{ji}^{(\mp)}, \qquad Q^{\pm\pm(1,1)} := \sum_{i,j} (ij) q^{j-i} f_{ij}^{(\pm)} \otimes f_{-i,-j}^{(\pm)}.$$

**Proposition 2.17.** *The spinor-spinor R-matrices of $U_q^{ex}(\mathfrak{Lso}_{2n+2})$ for $n > 2$ are elements of the spaces $\text{End}(V^{\pm(n)} \otimes V^{\pm(n)})$ and $\text{End}(V^{\pm(n)} \otimes V^{\mp(n)})$ given by the following recurrence formulas:*

$$
\begin{aligned}
R^{\pm\pm(n,n)}(u,v) = &\; R^{\pm\pm(n-1,n-1)}(u,v)\,\hat{\otimes}\,(e_{-1,-1}^{(1)} \otimes e_{-1,-1}^{(1)}) \\
&+ R^{\mp\mp(n-1,n-1)}(u,v)\,\hat{\otimes}\,(e_{+1,+1}^{(1)} \otimes e_{+1,+1}^{(1)}) \\
&+ \alpha(u,v)\Big( R^{\pm\mp(n-1,n-1)}(u,q^2v)\,\hat{\otimes}\,(e_{-1,-1}^{(1)} \otimes e_{+1,+1}^{(1)}) \\
&\qquad\quad + R^{\mp\pm(n-1,n-1)}(u,q^2v)\,\hat{\otimes}\,(e_{+1,+1}^{(1)} \otimes e_{-1,-1}^{(1)})\Big) \\
&- \beta(u,v)\Big( v\, U^{\mp\pm(n-1,n-1)}(u,q^2v)\,\hat{\otimes}\,(e_{-1,+1}^{(1)} \otimes e_{+1,-1}^{(1)}) \\
&\qquad\quad + u\, U^{\pm\mp(n-1,n-1)}(u,q^2v)\,\hat{\otimes}\,(e_{+1,-1}^{(1)} \otimes e_{-1,+1}^{(1)})\Big),
\end{aligned}
\tag{49}
$$

$$
\begin{aligned}
R^{\pm\mp(n,n)}(u,v) = &\; R^{\pm\mp(n-1,n-1)}(u,v)\,\hat{\otimes}\,(e_{-1,-1}^{(1)} \otimes e_{-1,-1}^{(1)}) \\
&+ R^{\mp\pm(n-1,n-1)}(u,v)\,\hat{\otimes}\,(e_{+1,+1}^{(1)} \otimes e_{+1,+1}^{(1)}) \\
&+ R^{\pm\pm(n-1,n-1)}(u,q^2v)\,\hat{\otimes}\,(e_{-1,-1}^{(1)} \otimes e_{+1,+1}^{(1)}) \\
&+ R^{\mp\mp(n-1,n-1)}(u,q^2v)\,\hat{\otimes}\,(e_{+1,+1}^{(1)} \otimes e_{-1,-1}^{(1)}) \\
&+ \frac{q-q^{-1}}{v-u}\Big( v\, U^{\mp\mp(n-1,n-1)}(u,q^2v)\,\hat{\otimes}\,(e_{-1,+1}^{(1)} \otimes e_{+1,-1}^{(1)}) \\
&\qquad\qquad + u\, U^{\pm\pm(n-1,n-1)}(u,q^2v)\,\hat{\otimes}\,(e_{+1,-1}^{(1)} \otimes e_{-1,+1}^{(1)})\Big),
\end{aligned}
\tag{50}
$$

*where*

$$U^{\pm\mp(n-1,n-1)}(u,v) := R^{\mp\pm(n-1,n-1)}(q^2,1)\, F^{\pm\mp(n-1,n-1)} R^{\pm\mp(n-1,n-1)}(u,v), \tag{51}$$

$$U^{\pm\pm(n-1,n-1)}(u,v) := Q^{\pm\pm(n-1,n-1)}\, P^{\pm\pm(n-1,n-1)} R^{\pm\pm(n-1,n-1)}(u,v), \tag{52}$$

*with $F^{\pm\mp(n-1,n-1)}$ and $Q^{\pm\pm(n-1,n-1)}$ defined by*

$$
\begin{aligned}
F^{\pm\mp(n,n)} := &\; F^{\pm\mp(n-1,n-1)}\,\hat{\otimes}\,(e_{-1,-1}^{(1)} \otimes e_{-1,-1}^{(1)}) + F^{\mp\pm(n-1,n-1)}\,\hat{\otimes}\,(e_{+1,+1}^{(1)} \otimes e_{+1,+1}^{(1)}) \\
&+ P^{\pm\pm(n-1,n-1)}\,\hat{\otimes}\,(e_{-1,+1}^{(1)} \otimes e_{+1,-1}^{(1)}) + P^{\mp\mp(n-1,n-1)}\,\hat{\otimes}\,(e_{+1,-1}^{(1)} \otimes e_{-1,+1}^{(1)}),
\end{aligned}
\tag{53}
$$

$$
\begin{aligned}
Q^{\pm\pm(n,n)} := &\; Q^{\pm\pm(n-1,n-1)}\,\hat{\otimes}\,(e_{-1,-1}^{(1)} \otimes e_{-1,-1}^{(1)}) + Q^{\mp\mp(n-1,n-1)}\,\hat{\otimes}\,(e_{+1,+1}^{(1)} \otimes e_{+1,+1}^{(1)}) \\
&+ F^{\pm\mp(n-1,n-1)}\,\hat{\otimes}\,(e_{-1,-1}^{(1)} \otimes e_{+1,+1}^{(1)}) + F^{\mp\pm(n-1,n-1)}\,\hat{\otimes}\,(e_{+1,+1}^{(1)} \otimes e_{-1,-1}^{(1)}) \\
&+ q^{-1} R^{\mp\pm(n-1,n-1)}(q^2,1)\,\hat{\otimes}\,(e_{-1,+1}^{(1)} \otimes e_{+1,-1}^{(1)}) \\
&+ q\, R^{\pm\mp(n-1,n-1)}(q^2,1)\,\hat{\otimes}\,(e_{+1,-1}^{(1)} \otimes e_{-1,+1}^{(1)})
\end{aligned}
\tag{54}
$$

*and $P^{\pm\pm(n,n)} := R^{\pm\pm(n,n)}(u,u)$.*

**Lemma 2.18.** *Let $\epsilon_1, \epsilon_2 = \pm$. The inverses of the spinor-spinor R-matrices of $U_q^{ex}(\mathfrak{Lso}_{2n+2})$ are given by*

$$R_{q^{-1}}^{\epsilon_1\epsilon_2(n,n)}(u,v) = P^{\epsilon_1\epsilon_2(n,n)} R^{\epsilon_1\epsilon_2(n,n)}(v,u) P^{\epsilon_1\epsilon_2(n,n)} = \big(R^{\epsilon_1\epsilon_2(n,n)}(u,v)\big)^{-1}. \tag{55}$$

*Moreover, the spinor-spinor R-matrices are crossing symmetric, that is*

$$(R^{\pm[\pm](n,n)}(q^{2n}u,v))^{\bar{w}_1} = (R^{\pm[\pm](n,n)}(q^{2n}u,v))^{w_2} = h^{+(n/2)}(u,v)(R^{\pm\pm(n,n)}(u,v))^{-1}, \qquad (56)$$

$$(R^{\pm[\mp](n,n)}(q^{2n}u,v))^{\bar{w}_1} = (R^{\pm[\mp](n,n)}(q^{2n}u,v))^{w_2} = h^{-(n/2)}(u,v)(R^{\pm\mp(n,n)}(u,v))^{-1}, \qquad (57)$$

*where* $[\pm] = \pm/\mp$ *if n is odd/even and similarly for* $[\mp]$ *and*

$$h^{+(n/2)}(u,v) := \prod_{j=1}^{\lceil n/2 \rceil} \alpha(q^{4j-2}u,v), \qquad h^{-(n/2)}(u,v) := \prod_{j=1}^{\lfloor n/2 \rfloor} \alpha(q^{4j}u,v) \qquad (58)$$

*and the q-transposition* $w$ *is defined via (6–7).*

**Lemma 2.19.** *Let* $\epsilon_1, \epsilon_2, \epsilon_3 = \pm$. *The spinor-spinor R-matrices of* $U_q^{ex}(\mathfrak{Lso}_{2n+2})$ *satisfy the following quantum Yang-Baxter equations:*

$$R_{12}^{\epsilon_1\epsilon_2(n,n)}(u,v)R_{13}^{\epsilon_1\epsilon_3(n,n)}(u,w)R_{23}^{\epsilon_2\epsilon_3(n,n)}(v,w) = R_{23}^{\epsilon_2\epsilon_3(n,n)}(v,w)R_{13}^{\epsilon_1\epsilon_3(n,n)}(u,w)R_{12}^{\epsilon_1\epsilon_2(n,n)}(u,v),$$

$$R_{12}^{\epsilon_1\epsilon_2(n,n)}(u,v)R_{13}^{\epsilon_1(n)}(u,\rho)R_{23}^{\epsilon_2(n)}(v,\rho) = R_{23}^{\epsilon_2(n)}(v,\rho)R_{13}^{\epsilon_1(n)}(u,\rho)R_{12}^{\epsilon_1\epsilon_2(n,n)}(u,v).$$

## 2.7 Fusion relations

We demonstrate fusion relations for spinor-spinor and spinor-vector *R*-matrices that may be viewed as *q*-analogues of relations (3.16) and (4.27) in [1]. We will make use of the usual check-notation, i.e. $\check{R}^{(n,n)} := P^{(n,n)}R^{(n,n)}$.

Consider the algebra $U_{q^2}(\mathfrak{so}_{2n+1})$ generated by the elements $\ell_{ij}^\pm$ with $-n \leq i,j \leq n$. Define a vector $\eta^{(n,n)} \in \mathbb{C}^{n|n} \otimes \mathbb{C}^{n|n}$ by

$$\eta^{(n,n)} := \left( \bigotimes_{i=1}^{n-1} \left( e_{-1}^{(1)} \otimes e_{+1}^{(1)} + (-1)^i q^{-2i+1} e_{+1}^{(1)} \otimes e_{-1}^{(1)} \right) \right) \hat{\otimes} \left( e_{-1}^{(1)} \otimes e_{-1}^{(1)} \right). \qquad (59)$$

Vector $\eta^{(n,n)}$ is a highest vector; it is a direct computation to verify that

$$\ell_{ij}^+ \cdot \eta^{(n,n)} = 0 \ \text{ for } \ i < j \ \text{ and}$$
$$\ell_{ii}^+ \cdot \eta^{(n,n)} = q^{2\delta_{in}-2\delta_{i,-n}} \eta^{(n,n)},$$

where the left $U_{q^2}(\mathfrak{so}_{2n+1})$-action is given by composing coproduct with the homomorphism $\pi \otimes \pi$ and representation $\sigma \otimes \sigma$. It follows that the subspace

$$W^{(n,n)} := U_{q^2}(\mathfrak{so}_{2n+1}) \cdot \eta^{(n,n)} \subset \mathbb{C}^{n|n} \otimes \mathbb{C}^{n|n}$$

is isomorphic to the first fundamental (vector) representation of $U_{q^2}(\mathfrak{so}_{2n+1})$, $W^{(n,n)} \cong \mathbb{C}^{2n+1}$.

**Lemma 2.20.** *Let* $\equiv$ *denote equality of operators in the space* $\mathbb{C}^{n|n} \otimes W^{(n,n)} \subset (\mathbb{C}^{n|n})^{\otimes 3}$. *Then, upon a suitable identification of* $W^{(n,n)}$ *and* $\mathbb{C}^{2n+1}$ *(which we label by the subscript* (23)*), we have that*

$$R_{13}^{(n,n)}(q^4 v, u) R_{12}^{(n,n)}(q^{4n-2}v, u) \equiv \frac{h^{(n)}(v,u)}{f_q(v,u)} R_{1(23)}^{(n)}(v,u). \qquad (60)$$

*Proof.* Define $\Pi^{(1,1)} := \check{R}^{(1,1)}(q^{-2},1)$ and $\Pi^{(n,n)} := \left( (1-q^{6-4n}v)\check{R}^{(n,n)}(v,1) \right)\big|_{v=q^{4n-6}}$ when $n \geq 2$. The operator $\Pi^{(n,n)}$ is a projector operator acting on $\eta^{(n,n)}$ by a scalar multiplication. In particular, it projects the space $\mathbb{C}^{n|n} \otimes \mathbb{C}^{n|n}$ to its subspace $W^{(n,n)}$. The Yang-Baxter equation (43) then implies that the l.h.s. of (60) acts stably on the space $\mathbb{C}^{n|n} \otimes W^{(n,n)}$. Therefore, thanks to the Schur's Lemma, it is sufficient to verify the equality (60) for a single vector, say $e_{-1}^{(1)} \otimes \eta^{(n,n)} \equiv e_{-1}^{(1)} \otimes e_{-n}$. $\qquad \square$

Next, for $n \geq 2$, consider the algebra $U_q(\mathfrak{so}_{2n+2})$ generated by the elements $\ell_{ij}^{\pm}$ with $-n-1 \leq i,j \leq n+1$. Introduce vectors

$$\psi^{\pm\pm(1,1)} := e_{+1}^{(\pm)} \otimes e_{-1}^{(\pm)} - q\, e_{-1}^{(\pm)} \otimes e_{+1}^{(\pm)} \in V^{\pm(1)} \otimes V^{\pm(1)}$$

satisfying

$$\ell_{ij}^{-} \cdot \psi^{\pm\pm(1,1)} = \ell_{ij}^{+} \cdot \psi^{\pm\pm(1,1)} = \delta_{ij} \psi^{\pm\pm(1,1)} \text{ for } -2 \leq i,j \leq 2\,.$$

Then, for $2 \leq k < n$, define recurrently vectors

$$\psi^{\mp\pm(k,k)} := \psi^{\pm\pm(k-1,k-1)} \,\hat{\otimes}\, (e_{+1}^{(1)} \otimes e_{-1}^{(1)}) + q^k \, \psi^{\mp\mp(k-1,k-1)} \,\hat{\otimes}\, (e_{-1}^{(1)} \otimes e_{+1}^{(1)}) \quad \text{if } k \text{ is even,}$$

$$\psi^{\pm\pm(k,k)} := \psi^{\pm\pm(k-2,k-2)} \,\hat{\otimes}\, \phi_{q^{2k-1}}^{++(2,2)} + q^{k-1} \, \psi^{\mp\mp(k-2,k-2)} \,\hat{\otimes}\, \phi_q^{--(2,2)} \quad \text{if } k \text{ is odd,}$$

where

$$\phi_q^{\pm\pm(2,2)} := \left(e_{\pm1}^{(1)} \otimes e_{\mp1}^{(1)}\right) \hat{\otimes} \left(e_{+1}^{(1)} \otimes e_{-1}^{(1)}\right) - q \left(e_{\mp1}^{(1)} \otimes e_{\pm1}^{(1)}\right) \hat{\otimes} \left(e_{-1}^{(1)} \otimes e_{+1}^{(1)}\right).$$

Finally set

$$\eta^{[\mp]\pm(n,n)} := \psi^{[\mp]\pm(n-1,n-1)} \,\hat{\otimes}\, (e_{-1}^{(1)} \otimes e_{-1}^{(1)}) \in V^{[\mp](n)} \otimes V^{\pm(n)}\,, \tag{61}$$

where $[\mp] = \mp/\pm$ if $n$ is odd/even. It is a highest vector; it is a direct computation to verify that

$$\ell_{ij}^{+} \cdot \eta^{[\mp]\pm(n,n)} = 0 \quad \text{for } i < j \text{ and}$$

$$\ell_{ii}^{+} \cdot \eta^{[\mp]\pm(n,n)} = q^{\delta_{i,n+1} - \delta_{-i,n+1}} \eta^{[\mp]\pm(n,n)}\,.$$

Thus the space

$$W^{[\mp]\pm(n,n)} := U_q(\mathfrak{so}_{2n+2}) \cdot \eta^{[\mp]\pm(n,n)} \subset V^{[\mp](n)} \otimes V^{\pm(n)}$$

is isomorphic to the first fundamental (vector) representation of $U_q(\mathfrak{so}_{2n+2})$, that is $W^{[\mp]\pm(n,n)} \cong \mathbb{C}^{2n+2}$.

**Lemma 2.21.** *Let $\equiv$ denote equality of operators in the space $V^{\epsilon(n)} \otimes W^{[\mp]\pm(n,n)}$. Then, upon a suitable identification of $W^{[\mp]\pm(n,n)}$ and $\mathbb{C}^{2n+2}$ (which we label by the subscript (23)), we have that*

$$R_{13}^{\mp\pm(n,n)}(q^2 v, u) R_{12}^{\mp[\mp](n,n)}(q^{2n} v, u) \equiv \frac{h^{+(n/2)}(v,u)}{f_q(v;u)} R_{1(23)}^{\mp(n)}(v,u)\,, \tag{62}$$

$$R_{13}^{\pm\pm(n,n)}(q^2 v, u) R_{12}^{\pm[\mp](n,n)}(q^{2n} v, u) \equiv h^{-(n/2)}(v,u) R_{1(23)}^{\pm(n)}(v,u)\,, \tag{63}$$

*where $h^{\pm(n/2)}(v,u)$ is given by (58) and $[\mp] = \mp/\pm$ when $n$ is odd/even.*

*Proof.* The proof is analogous to that of Lemma 2.20 except the projection operator is now defined by $\Pi^{[\mp]\pm(n,n)} := \left((1 - q^{2-2n} v) \check{R}^{[\mp]\pm(n,n)}(v,1)\right)\big|_{v=q^{2n-2}}$. $\qquad\square$

## 2.8 Exchange relations

The last ingredient that we will need are spinor-type Yang-Baxter exchange relations imposed by the spinor-spinor $R$-matrices. We will need "BB", "AB" and "DB" type relations only. For any $n \geq 0$ introduce a matrix $T^{(n+1)}(u)$ in $\mathrm{End}(\mathbb{C}^{n+1|n+1})$ with entries being operators in an associative algebra. Then write $T^{(n+1)}(u)$ in the nested form,

$$T^{(n+1)}(u) = A^{(n)}(u) \,\hat{\otimes}\, e_{-1,-1}^{(1)} + B^{(n)}(u) \,\hat{\otimes}\, e_{-1,+1}^{(1)} + C^{(n)}(u) \,\hat{\otimes}\, e_{+1,-1}^{(1)} + D^{(n)}(u) \,\hat{\otimes}\, e_{+1,+1}^{(1)}\,, \tag{64}$$

and require it to satisfy the equation

$$R_{12}^{(n+1,n+1)}(u,v)\, T_1^{(n+1)}(u)\, T_2^{(n+1)}(v) = T_2^{(n+1)}(v)\, T_1^{(n+1)}(u)\, R_{12}^{(n+1,n+1)}(u,v)\,, \tag{65}$$

so that the entries of $T^{(n+1)}(u)$ were operators in a Yang-Baxter algebra.

**Lemma 2.22.** *We have the following "BB", "AB" and "DB" exchange relations:*

$$R_{12}^{(n,n)}(v,u)B_1^{(n)}(v)B_2^{(n)}(u) = B_2^{(n)}(u)B_1^{(n)}(v)R_{12}^{(n,n)}(v,u), \tag{66}$$

$$A_1^{(n)}(v)B_2^{(n)}(u) = f_q(v,u)R_{21}^{(n,n)}(u,v)B_2^{(n)}(u)A_1^{(n)}(v)R_{12}'^{(n,n)}(q^4v,u)$$
$$- \frac{v/u}{v-u}\operatorname*{Res}_{w\to u}\left(f_q(w,u)R_{21}^{(n,n)}(u,w)B_2^{(n)}(v)A_1^{(n)}(w)R_{12}'^{(n,n)}(q^4w,u)\right), \tag{67}$$

$$D_1^{(n)}(v)B_2^{(n)}(u) = f_{q^{-1}}(v,u)R_{21}^{(n,n)}(q^4u,v)B_2^{(n)}(u)D_1^{(n)}(v)R_{12}'^{(n,n)}(v,u)$$
$$- \frac{v/u}{v-u}\operatorname*{Res}_{w\to u}\left(f_{q^{-1}}(w,u)R_{21}^{(n,n)}(q^4u,w)B_2^{(n)}(v)D_1^{(n)}(w)R_{12}'^{(n,n)}(w,u)\right), \tag{68}$$

*where $R^{(0,0)}(u,v) = 1$ and $R'^{(n,n)} := (\gamma \otimes id)(R^{(n,n)}) = (id \otimes \gamma)(R^{(n,n)})$.*

*Proof.* These relations are obtained by substituting (64) into (65). For (67) and (68) one also needs to use (41), $R^{(n,n)}(u,u) = P^{(n,n)}$, and

$$P_{12}'^{(n,n)}R_{12}^{(n,n)}(u,v)P_{12}'^{(n,n)} = R_{21}^{(n,n)}(u,v), \qquad P_{12}'^{(n,n)}X_1'^{(n)}P_{12}'^{(n,n)} = X_2^{(n)},$$

for any $X^{(n)} \in \operatorname{End}(\mathbb{C}^{n|n})$ and $X'^{(n)} = \gamma(X^{(n)})$ with $\gamma(e_{ij}^{(n)}) = \theta_{ij} e_{ij}^{(n)}$. □

Next, introduce a matrix $T^{\pm(n+1)}(u)$ in $\operatorname{End}(V^{\pm(n+1)})$ with entries being operators in an associative algebra. Then write $T^{\pm(n+1)}(u)$ as

$$T^{\pm(n+1)}(u) = A^{\pm(n)}(u)\hat{\otimes}e_{-1,-1}^{(1)} + B^{\mp(n)}(u)\hat{\otimes}e_{-1,+1}^{(1)}$$
$$+ C^{\pm(n)}(u)\hat{\otimes}e_{+1,-1}^{(1)} + D^{\mp(n)}(u)\hat{\otimes}e_{+1,+1}^{(1)} \tag{69}$$

and require it to satisfy the equation

$$R_{12}^{\epsilon_1\epsilon_2(n+1,n+1)}(u,v)\,T_1^{\epsilon_1(n+1)}(u)\,T_2^{\epsilon_2(n+1)}(v) = T_2^{\epsilon_2(n+1)}(v)\,T_1^{\epsilon_1(n+1)}(u)\,R_{12}^{\epsilon_1\epsilon_2(n+1,n+1)}(u,v), \tag{70}$$

where $\epsilon_1, \epsilon_2 = \pm$.

**Lemma 2.23.** *We have the following "BB", "AB" and "DB" exchange relations:*

$$R_{12}^{-\epsilon_1-\epsilon_2(n,n)}(v,u)B_1^{\epsilon_1(n)}(v)B_2^{\epsilon_2(n)}(u) = B_2^{\epsilon_2(n)}(u)B_1^{\epsilon_1(n)}(v)R_{12}^{\epsilon_1\epsilon_2(n,n)}(v,u), \tag{71}$$

$$A_1^{\pm(n)}(v)B_2^{\mp(n)}(u) = f_q(v,u)R_{21}^{\pm\pm(n,n)}(u,v)B_2^{\mp(n)}(u)A_1^{\pm(n)}(v)R_{12}^{\pm\mp(n,n)}(q^2v,u)$$
$$- \frac{v/u}{v-u}\operatorname*{Res}_{w\to u}\Big(f_q(w,u)R_{21}^{\pm\pm(n,n)}(u,w)$$
$$\times B_2^{\mp(n)}(v)A_1^{\pm(n)}(w)R_{12}^{\pm\mp(n,n)}(q^2w,u)\Big), \tag{72}$$

$$D_1^{\mp(n)}(v)B_2^{\mp(n)}(u) = f_{q^{-1}}(v,u)R_{21}^{\pm\mp(n,n)}(q^2u,v)B_2^{\mp(n)}(u)D_1^{\mp(n)}(v)R_{12}^{\mp\mp(n,n)}(v,u)$$
$$- \frac{v/u}{v-u}\operatorname*{Res}_{w\to u}\Big(f_{q^{-1}}(w,u)R_{21}^{\pm\mp(n,n)}(q^2u,w)$$
$$\times B_2^{\mp(n)}(v)D_1^{\mp(n)}(w)R_{12}^{\mp\mp(n,n)}(w,u)\Big), \tag{73}$$

$$A_1^{\pm(n)}(v)B_2^{\pm(n)}(u) = R_{21}^{\mp\pm(n,n)}(u,v)B_2^{\pm(n)}(u)A_1^{\pm(n)}(v)R_{12}^{\pm\pm(n,n)}(q^2v,u)$$
$$- v\frac{q-q^{-1}}{v-u}B_1^{\mp(n)}(v)A_2^{\mp(n)}(u)$$
$$\times U_{21}^{\pm\pm(n,n)}(u,q^2v)R_{12}^{\pm\pm(n,n)}(q^2v,u), \tag{74}$$

$$D_1^{\mp(n)}(v)B_2^{\pm(n)}(u) = R_{21}^{\mp\mp(n,n)}(q^2 u, v)B_2^{\pm(n)}(u)D_1^{\mp(n)}(v)R_{12}^{\mp\pm(n,n)}(v,u)$$

$$-u\,\frac{q-q^{-1}}{u-v}\,R_{21}^{\mp\mp(n,n)}(q^2 u, v)$$

$$\times\, U_{21}^{\pm\pm(n,n)}(v, q^2 u)B_1^{\mp(n)}(v)D_2^{\pm(n)}(u), \qquad (75)$$

where $U_{21}^{\pm\pm(1,1)}(u, q^2 v) := \dfrac{v-u}{q^2 v - q^{-2}u}\,Q_{21}^{\pm\pm(1,1)}$.

*Proof.* The proof is analogous to that of Lemma 2.22. The exchange relations are obtained by substituting (69) into (70). For (72) and (73) one also needs to use (55) and $R^{\pm\pm(n,n)}(u,u) = P^{\pm\pm(n,n)}$.

$\square$

# 3  Algebraic Bethe Ansatz for $U_{q^2}(\mathfrak{so}_{2n+1})$-symmetric spin chains

In this section we study spectrum of $U_{q^2}(\mathfrak{so}_{2n+1})$-symmetric chains with the *full quantum space* given by

$$L^{(n)} = L^V := (\mathbb{C}^{2n+1})^{\otimes\ell} \quad\text{or}\quad L^{(n)} = L^S := (\mathbb{C}^{n|n})^{\otimes\ell}, \qquad (76)$$

where $\ell \in \mathbb{N}$ is the length of the chain. We will say that $L^{(n)}$ is the *level-n quantum space*. For each individual quantum space we assign a non-zero complex parameter $\rho_i$, called an *inhomogeneity* or a *marked point*. Their collection will be denoted by $\boldsymbol{\rho} = (\rho_1, \dots, \rho_\ell) \in (\mathbb{C}^\times)^\ell$. We will assume that all $\rho_i$ are distinct.

## 3.1  Quantum spaces and monodromy matrices

Choose $m_1, m_2, \dots, m_n \in \mathbb{Z}_{\geq 0}$, the excitation, or magnon, numbers. For each $m_k$ assign an $m_k$-tuple $\boldsymbol{u}^{(k)} := (u_1^{(k)}, \dots, u_{m_k}^{(k)})$ of non-zero complex parameters that will accommodate Bethe roots, and, when $k \geq 2$, three $m_k$-tuples of labels, $\dot{\boldsymbol{a}}^k := (\dot{a}_1^k, \dots, \dot{a}_{m_k}^k)$, $\ddot{\boldsymbol{a}}^k := (\ddot{a}_1^k, \dots, \ddot{a}_{m_k}^k)$, and $\boldsymbol{a}^k := (a_1^k, \dots, a_{m_k}^k)$. These labels will be used to enumerate *nested quantum spaces*. In particular, for each $\dot{a}_i^k$ and each $\ddot{a}_i^k$ we associate a copy of $\mathbb{C}^{k-1|k-1}$ denoted by $V_{\dot{a}_i^k}^{(k-1)}$ and $V_{\ddot{a}_i^k}^{(k-1)}$, respectively. We then identify subspaces $W_{a_i^k} \subset V_{\dot{a}_i^k}^{(k-1)} \otimes V_{\ddot{a}_i^k}^{(k-1)}$, isomorphic to $\mathbb{C}^{2k-1}$, in the following way. Let $\eta_{a_i^k} \in V_{\dot{a}_i^k}^{(k-1)} \otimes V_{\ddot{a}_i^k}^{(k-1)}$ be a highest vector as per (59). Then $W_{a_i^k} \cong U_{q^2}(\mathfrak{so}_{2k-1}) \cdot \eta_{a_i^k}$, as a vector space.

For each $1 \leq k < n$ we recurrently define the *nested level-k quantum space* $L^{(k)}$ by

$$L^{(k)} := (L^{(k+1)})^0 \otimes W_{a_1^{k+1}} \otimes \cdots \otimes W_{a_{m_{k+1}}^{k+1}},$$

where $(L^{(k+1)})^0 \subset L^{(k+1)}$ is the *level-(k+1) vacuum subspace* such that each individual tensorand is a $U_{q^2}(\mathfrak{so}_{2k+1}) \subset U_{q^2}(\mathfrak{so}_{2k+3})$ stable subspace annihilated by $\ell_{i,k+1}^+$ with $-k-1 \leq i \leq k$. In particular, $(L^{(k+1)})^0 \cong \mathbb{C}$ or $(\mathbb{C}^{k|k})^{\otimes\ell}$ when $L^{(n)} = L^V$ or $L^S$, respectively.

We will make use of the following shorthand notation:

$$\alpha(v; \boldsymbol{u}^{(k)}) := \prod_{i=1}^{m_k} \alpha(v, u_i^{(k)}), \qquad f_q(v; \boldsymbol{u}^{(k)}) := \prod_{i=1}^{m_k} f_q(v, u_i^{(k)}).$$

For any $k < l$ we set $\boldsymbol{u}^{(k\dots l)} := (\boldsymbol{u}^{(k)}, \dots, \boldsymbol{u}^{(l)})$ and $\boldsymbol{u}^{(l\dots k)} := \emptyset$. We will also assume that $\boldsymbol{u}^{(n+1)} = \boldsymbol{\rho}$.

Having set up all the necessary quantum spaces and the shorthand notation we are ready to introduce the relevant monodromy matrices of the spin chain. With this goal in mind we introduce a diagonal "twist" matrix

$$\mathcal{E}^{(n)} := \sum_{\boldsymbol{i}} \varepsilon_{i_1}^{(1)} \cdots \varepsilon_{i_n}^{(n)} e_{i_1 i_1}^{(1)} \hat{\otimes} \cdots \hat{\otimes} e_{i_n i_n}^{(1)} \in \mathrm{End}(\mathbb{C}^{n|n})$$

and set $\varepsilon^{(k)} := \varepsilon_{+1}^{(k)}/\varepsilon_{-1}^{(k)}$ for all $k$. This matrix satisfies the Yang-Baxter relation (65) and will play the role of the twisted diagonal periodic boundary conditions. (Note the factorisation relation: $\mathcal{E}^{(n)} = \mathcal{E}^{(n-1)} \hat{\otimes} (\varepsilon_{-1}^{(n)} e_{-1,-1}^{(1)} + \varepsilon_{+1}^{(n)} e_{+1,+1}^{(1)})$ with $\mathcal{E}^{(n-1)} \in \mathrm{End}(\mathbb{C}^{n-1|n-1})$.) Let $V_a^{(k)}$ and $V_b^{(k)}$ denote copies of $\mathbb{C}^{k|k}$, called *auxiliary spaces*. We define the *level-$n$ monodromy matrix* with entries acting on the level-$n$ quantum space $L^{(n)}$ by

$$T_a^{(n)}(v) := \mathcal{E}_a^{(n)} T_{a1}^{(n)}(v, \rho_1) \cdots T_{a\ell}^{(n)}(v, \rho_\ell), \tag{77}$$

where $T_{ai}^{(n)}(v, \rho_i) = R_{ai}^{(n)}(v, \rho_i)$ or $R_{ai}^{(n,n)}(q^2 v, \rho_i)$ when $L^{(n)} = L^V$ or $L^S$, respectively. (The $q^2$ in $R_{ai}^{(n,n)}(q^2 v, \rho_i)$ helps the final expressions to be more elegant.) Then, for each $1 \le k < n$, we recurrently define the *nested level-$k$ monodromy matrices* with entries acting on the nested level-$k$ quantum space $L^{(k)}$ by

$$\begin{aligned}
T_a^{(k)}(v; \boldsymbol{u}^{(k+1\ldots n)}) :=& \frac{f_q(v; \boldsymbol{u}^{(k+1)})}{h^{(k)}(v; \boldsymbol{u}^{(k+1)})} A_a^{(k)}(v; \boldsymbol{u}^{(k+2\ldots n)}) \\
& \times \prod_{i=1}^{m_{k+1}} R_{a\ddot{a}_i^{k+1}}'^{(k,k)}(q^4 v, u_i^{(k+1)}) R_{a\dot{a}_i^{k+1}}'^{(k,k)}(q^{4k-2} v, u_i^{(k+1)}) \\
\equiv& A_a^{(k)}(v; \boldsymbol{u}^{(k+2\ldots n)}) \prod_{i=1}^{m_{k+1}} R_{aa_i^{k+1}}^{(k)}(v, u_i^{(k+1)}),
\end{aligned} \tag{78}$$

$$\begin{aligned}
\widetilde{T}_a^{(k)}(v; \boldsymbol{u}^{(k+1\ldots n)}) :=& \frac{f_q(q^{-4} v; \boldsymbol{u}^{(k+1)})}{h^{(k)}(q^{-4} v; \boldsymbol{u}^{(k+1)})} D_a^{(k)}(v; \boldsymbol{u}^{(k+2\ldots n)}) \\
& \times \prod_{i=1}^{m_{k+1}} R_{a\ddot{a}_i^{k+1}}'^{(k,k)}(v, u_i^{(k+1)}) R_{a\dot{a}_i^{k+1}}'^{(k,k)}(q^{4k-6} v, u_i^{(k+1)}) \\
\equiv& D_a^{(k)}(v; \boldsymbol{u}^{(k+2\ldots n)}) \prod_{i=1}^{m_{k+1}} R_{aa_i^{k+1}}^{(k)}(v, q^4 u_i^{(k+1)}),
\end{aligned} \tag{79}$$

where

$$A_a^{(k)}(v; \boldsymbol{u}^{(k+2\ldots n)}) = \left[ T_a^{(k+1)}(v; \boldsymbol{u}^{(k+2\ldots n)}) \right]_{-1,-1}, \tag{80}$$

$$D_a^{(k)}(v; \boldsymbol{u}^{(k+2\ldots n)}) = \left[ T_a^{(k+1)}(v; \boldsymbol{u}^{(k+2\ldots n)}) \right]_{+1,+1}, \tag{81}$$

and $\equiv$ denotes equality of operators in the space $L^{(k)}$ subject to a suitable identification of the spaces $W_{a_i^{k+1}} \subset V_{\dot{a}_i^{k+1}}^{(k)} \otimes V_{\ddot{a}_i^{k+1}}^{(k)}$ and copies of $\mathbb{C}^{2k+1}$, as per Lemma 2.20.

The nested monodromy matrices span the following nesting tree:

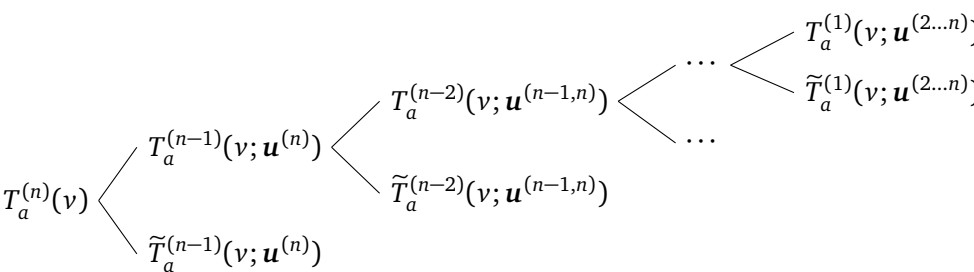

It will be sufficient to focus on the non-tilded monodromy matrices at each level of nesting. Indeed, it follows from the explicit form of the spinor $R$-matrices given by (34) and (39) and definitions of the nested monodromy matrices in (78) and (79) that we have the following equalities of operators (80) and (81) in the spaces $L^{(n-1)}$ and $L^{(k)}$ with $1 \le k < n-1$, subject to the choice of the full quantum space $L^{(n)}$:

| | $L^V$ | $L^S$ |
|---|---|---|
| $A_a^{(n-1)}(v)/\varepsilon_{-1}^{(n)}$ | $\mathcal{E}_a^{(n-1)}$ | $T_a^{(n-1)}(v)$ |
| $D_a^{(n-1)}(v)/\varepsilon_{+1}^{(n)}$ | $f_{q^2}(v;\boldsymbol{\rho})\,\mathcal{E}_a^{(n-1)}$ | $f_q(v;\boldsymbol{\rho})\,T_a^{(n-1)}(q^{-4}v)$ |
| $A_a^{(k)}(v;\boldsymbol{u}^{(k+2\ldots n)})/\tilde{\varepsilon}_{-1}^{(k+1)}$ | $\mathcal{E}_a^{(k)}$ | $T_a^{(k)}(v)$ |
| $D_a^{(k)}(v;\boldsymbol{u}^{(k+2\ldots n)})/\tilde{\varepsilon}_{+1}^{(k+1)}$ | $f_{q^2}(v;\boldsymbol{u}^{(k+2)})\,\mathcal{E}_a^{(k)}$ | $f_q(v;\boldsymbol{\rho})\,f_{q^2}(v;\boldsymbol{u}^{(k+2)})\,T_a^{(k)}(q^{-4}v)$ |

Here $\tilde{\varepsilon}_{\mp 1}^{(k+1)} = \varepsilon_{-1}^{(n)} \cdots \varepsilon_{-1}^{(k+2)} \varepsilon_{\mp 1}^{(k+1)}$ and the operators $T_a^{(n-1)}(v)$ and $T_a^{(k)}(v)$ are defined in the same way as $T_a^{(n)}(v)$, viz. (77). This table states that, for instance, $A_a^{(n-1)}(v) \equiv \varepsilon_{-1}^{(n)} \mathcal{E}_a^{(n-1)}$ or $\varepsilon_{-1}^{(n)} T_a^{(n-1)}(v)$ in the space $L^{(n-1)}$ when $L^{(n)} = L^V$ or $L^S$, respectively. It is now easy to deduce that

$$R_{ab}^{(k,k)}(v,w)\,T_a^{(k)}(v;\boldsymbol{u}^{(k+1\ldots n)})\,T_b^{(k)}(w;\boldsymbol{u}^{(k+1\ldots n)})$$
$$\equiv T_b^{(k)}(w;\boldsymbol{u}^{(k+1\ldots n)})\,T_a^{(k)}(v;\boldsymbol{u}^{(k+1\ldots n)})\,R_{ab}^{(k,k)}(v,w), \tag{82}$$

for $1 \le k < n$. Therefore the entries of $T_a^{(k)}(v;\boldsymbol{u}^{(k+1\ldots n)})$ in the space $L^{(k)}$ satisfy exchange relations given by Lemma 2.22. In other words, $T_a^{(k)}(v;\boldsymbol{u}^{(k+1\ldots n)})$ is a monodromy matrix for a nested $U_{q^2}(\mathfrak{so}_{2k+1})$-symmetric spin chain with the full quantum space $L^{(k)}$.

## 3.2 Creation operators and Bethe vectors

For each level of nesting we need to introduce $m_k$-magnon creation operators that will help us to define Bethe vectors. We will make use of the following notation:

$$\mathcal{b}(v;\boldsymbol{u}^{(2\ldots n)}) := \left[T_a^{(1)}(v;\boldsymbol{u}^{(2\ldots n)})\right]_{-1,+1},$$
$$B_a^{(k-1)}(v;\boldsymbol{u}^{(k+1\ldots n)}) := \left[T_a^{(k)}(v;\boldsymbol{u}^{(k+1\ldots n)})\right]_{-1,+1},$$

where $2 \le k \le n$. Note that $\mathcal{b}$ is an operator acting on $L^{(1)}$, and $B_a^{(k-1)}$ is a matrix in $\mathrm{End}(V_a^{(k-1)})$ with entries acting on $L^{(k)}$.

We define the *level-1 creation operator* by

$$\mathcal{B}^{(0)}(\boldsymbol{u}^{(1)};\boldsymbol{u}^{(2\ldots n)}) := \prod_{i=m_1}^{1} \mathcal{b}(u_i^{(1)};\boldsymbol{u}^{(2\ldots n)}). \tag{83}$$

For each $2 \le k \le n$ we define the *level-k creation operator* by

$$\mathcal{B}^{(k-1)}(\boldsymbol{u}^{(k)};\boldsymbol{u}^{(k+1\ldots n)}) := \prod_{i=m_k}^{1} \mathcal{b}_{\dot{a}_i^k \ddot{a}_i^k}^{(k-1,k-1)}(u_i^{(k)};\boldsymbol{u}^{(k+1\ldots n)}), \tag{84}$$

where

$$\mathcal{b}_{\dot{a}_i^k \ddot{a}_i^k}^{(k-1,k-1)}(u_i^{(k)};\boldsymbol{u}^{(k+1\ldots n)}) := \chi_{\dot{a}_i^k \ddot{a}_i^k}^{(k-1)}\left(B_a^{(k-1)}(u_i^{(k)};\boldsymbol{u}^{(k+1\ldots n)})\right), \tag{85}$$

with $\chi_{\dot{a}_i^k \ddot{a}_i^k}^{(k-1)} : \mathrm{End}(V_a^{(k-1)}) \to (V_{\dot{a}_i^k}^{(k-1)})^* \otimes (V_{\ddot{a}_i^k}^{(k-1)})^*$ defined via (4).

Bethe vectors will be constructed by acting with creation operators on a suitably chosen highest vector $\eta \in L^{(1)}$, the *nested vacuum vector*, defined by

$$\eta := \eta_1 \otimes \cdots \otimes \eta_\ell \otimes \eta_{a_1^n} \otimes \cdots \otimes \eta_{a_{m_n}^n} \otimes \cdots \otimes \eta_{a_1^2} \otimes \cdots \otimes \eta_{a_{m_2}^2}. \tag{86}$$

Here $\eta_1, \ldots, \eta_\ell$ are highest vectors of the initial quantum spaces, viz. (76), and $\eta_{a_1^n}, \ldots, \eta_{a_{m_2}^2}$ are highest vectors of the nested quantum spaces $W_{a_1^n}, \ldots, W_{a_{m_2}^2}$. For each $1 \le k \le n$ we define the *level-$k$ Bethe vector* by

$$\Phi^{(k)}(\boldsymbol{u}^{(1\ldots k)}; \boldsymbol{u}^{(k+1\ldots n)}) := \left( \prod_{i=k}^{1} \mathscr{B}^{(i-1)}(\boldsymbol{u}^{(i)}; \boldsymbol{u}^{(i+1\ldots n)}) \right) \cdot \eta. \tag{87}$$

The Bethe vector $\Phi^{(k)}(\boldsymbol{u}^{(1\ldots k)}; \boldsymbol{u}^{(k+1\ldots n)})$ is an element of the level-$k$ quantum space $L^{(k)}$ and has $\boldsymbol{u}^{(k+1\ldots n)}$ and $\boldsymbol{\rho}$ as its free parameters. Furthermore, it is invariant under an interchange of any two of its non-free parameters of the same level, i.e. $u_i^{(l)}$ and $u_j^{(l)}$ for any $1 \le l \le k$ and any admissible $i$ and $j$. Indeed, set $\mathfrak{S}_{\boldsymbol{m}_{1\ldots k}} := \mathfrak{S}_{m_1} \times \cdots \times \mathfrak{S}_{m_k}$ where each $\mathfrak{S}_{m_l}$ is the symmetric group on $m_l$ letters. Then, given any $\sigma^{(l)} \in \mathfrak{S}_{m_l}$, define the action of $\mathfrak{S}_{\boldsymbol{m}_{1\ldots k}}$ on $\Phi^{(k)}(\boldsymbol{u}^{(1\ldots k)}; \boldsymbol{u}^{(k+1\ldots n)})$ by

$$\sigma^{(l)} : \boldsymbol{u}^{(1\ldots k)} \mapsto \boldsymbol{u}_{\sigma^{(l)}}^{(1\ldots k)} := (\boldsymbol{u}^{(1)}, \ldots, \boldsymbol{u}_{\sigma^{(l)}}^{(l)}, \ldots, \boldsymbol{u}^{(k)}\} \quad \text{where} \quad \boldsymbol{u}_{\sigma^{(l)}}^{(l)} := (u_{\sigma^{(l)}(1)}^{(l)}, \ldots, u_{\sigma^{(l)}(m_l)}^{(l)}).$$

For further convenience we set $\sigma_j^{(l)} \in \mathfrak{S}_{m_l}$ to be the $j$-cycle such that

$$\boldsymbol{u}_{\sigma_j^{(l)}}^{(l)} = (u_j^{(l)}, u_{j+1}^{(l)}, \ldots, u_{m_l}^{(l)}, u_1^{(l)}, \ldots, u_{j-1}^{(l)}). \tag{88}$$

We will also make use of the notation

$$\boldsymbol{u}_{\sigma_j^{(l)}, u_j^{(l)} \to v}^{(l)} := \boldsymbol{u}_{\sigma_j^{(l)}}^{(l)} \Big|_{u_j^{(l)} \to v} = (v, u_{j+1}^{(l)}, \ldots, u_{m_l}^{(l)}, u_1^{(l)}, \ldots, u_{j-1}^{(l)}). \tag{89}$$

**Lemma 3.1.** *The Bethe vector* $\Phi^{(k)}(\boldsymbol{u}^{(1\ldots k)}; \boldsymbol{u}^{(k+1\ldots n)})$ *is invariant under the action of* $\mathfrak{S}_{\boldsymbol{m}_{1\ldots k}}$.

*Proof.* We rewrite the "BB" exchange relation (66) in terms of the creation operators (85),

$$\mathcal{b}_{\dot{a}_i^k \ddot{a}_i^k}^{(k-1,k-1)}(u_i^{(k)}; \boldsymbol{u}^{(k+1\ldots n)}) \mathcal{b}_{\dot{a}_{i+1}^k \ddot{a}_{i+1}^k}^{(k-1,k-1)}(u_{i+1}^{(k)}; \boldsymbol{u}^{(k+1\ldots n)})$$
$$= \mathcal{b}_{\dot{a}_i^k \ddot{a}_i^k}^{(k-1,k-1)}(u_{i+1}^{(k)}; \boldsymbol{u}^{(k+1\ldots n)}) \mathcal{b}_{\dot{a}_{i+1}^k \ddot{a}_{i+1}^k}^{(k-1,k-1)}(u_i^{(k)}; \boldsymbol{u}^{(k+1\ldots n)})$$
$$\times \hat{R}_{\dot{a}_i^k \dot{a}_{i+1}^k}^{(k-1,k-1)}(u_{i+1}^{(k)}, u_i^{(k)}) \breve{R}_{\ddot{a}_i^k \ddot{a}_{i+1}^k}^{(k-1,k-1)}(u_i^{(k)}, u_{i+1}^{(k)}),$$

where $\hat{R}^{(k,k)} := R^{(k,k)} P^{(k,k)}$ and $\breve{R}^{(k,k)} := P^{(k,k)} R^{(k,k)}$. Then one can verify that

$$\hat{R}_{\dot{a}_i^k \dot{a}_{i+1}^k}^{(k-1,k-1)}(u_{i+1}^{(k)}, u_i^{(k)}) \breve{R}_{\ddot{a}_i^k \ddot{a}_{i+1}^k}^{(k-1,k-1)}(u_i^{(k)}, u_{i+1}^{(k)}) \cdot \eta = \eta.$$

This implies that $\Phi^{(k)}(\boldsymbol{u}^{(1\ldots k)}; \boldsymbol{u}^{(k+1\ldots n)})$ is invariant under the interchange of $u_i^{(k)}$ and $u_{i+1}^{(k)}$. Analogous arguments also imply that $\Phi^{(k)}(\boldsymbol{u}^{(1\ldots k)}; \boldsymbol{u}^{(k+1\ldots n)})$ is invariant under the interchange of $u_i^{(l)}$ and $u_{i+1}^{(l)}$ for any $1 \le l \le k$ and any admissible $i$, thus implying the claim. $\square$

### 3.3 Transfer matrices, their eigenvalues, and Bethe equations

We are now in position to define transfer matrices and study their spectrum. We begin from the simplest case, the $U_{q^2}(\mathfrak{so}_3)$-symmetric spin chain. This chain is a special case of the XXZ spin chain with spin-$\frac{1}{2}$ transfer matrix and spin-1 or spin-$\frac{1}{2}$ quantum spaces when $L^{(1)} = L^V$ or $L^S$, respectively. It will serve us as a warm-up exercise. We define the *level-1 transfer matrix* by

$$\tau^{(1)}(v) := \operatorname{tr}_a T_a^{(1)}(v).$$

**Theorem 3.2.** *The Bethe vector* $\Phi^{(1)}(\boldsymbol{u}^{(1)})$ *is an eigenvector of* $\tau^{(1)}(v)$ *with the eigenvalue*

$$\Lambda^{(1)}(v; \boldsymbol{u}^{(1)}) := \varepsilon_{-1}^{(1)} f_q(v; \boldsymbol{u}^{(1)}) + \varepsilon_{+1}^{(1)} f_{q^{-1}}(v; \boldsymbol{u}^{(1)}) f_{q^\mu}(v; \boldsymbol{\rho}), \tag{90}$$

*where* $\mu = 2$ *or* 1 *when* $L^{(1)} = L^V$ *or* $L^S$, *respectively, provided*

$$\operatorname*{Res}_{v \to u_j^{(1)}} \Lambda^{(1)}(v; \boldsymbol{u}^{(1)}) = 0 \quad \text{for} \quad 1 \le j \le m_1. \tag{91}$$

The explicit form of the Bethe equations (91) is

$$\prod_{i=1}^{m_1} \frac{q u_j^{(1)} - q^{-1} u_i^{(1)}}{q^{-1} u_j^{(1)} - q u_i^{(1)}} = -\varepsilon^{(1)} \prod_{i=1}^{\ell} \frac{q^\mu v - q^{-\mu} \rho_i}{v - \rho_i}.$$

*Proof of Theorem 3.2.* This is a standard result, see e.g. [15]. Write $T^{(1)}(u)$ as

$$T^{(1)}(u) = a(u) e_{-1,-1}^{(1)} + b(u) e_{-1,+1}^{(1)} + c(u) e_{+1,-1}^{(1)} + d(u) e_{+1,+1}^{(1)}.$$

Lemma 2.22 then implies that

$$b(v) b(u) = b(u) b(v),$$

$$a(v) b(u) = f_q(v, u) b(u) a(v) - \frac{v/u}{v - u} \operatorname*{Res}_{w \to u} \big( f_q(w, u) b(v) a(w) \big),$$

$$d(v) b(u) = f_{q^{-1}}(v, u) b(u) d(v) - \frac{v/u}{v - u} \operatorname*{Res}_{w \to u} \big( f_{q^{-1}}(w, u) b(v) d(w) \big).$$

Using the relations above and the standard symmetry arguments, cf. Lemma 3.1, we obtain

$$\begin{aligned}
\tau^{(1)}(v) \Phi^{(1)}(\boldsymbol{u}^{(1)}) &= \big( a(v) + d(v) \big) \mathscr{B}^{(0)}(\boldsymbol{u}^{(1)}) \cdot \eta \\
&= \mathscr{B}^{(0)}(\boldsymbol{u}^{(1)}) \big( f_q(v; \boldsymbol{u}^{(1)}) a(v) + f_{q^{-1}}(v) d(v) \big) \cdot \eta \\
&\quad - \sum_{j=1}^{m_1} \frac{v/u_j^{(1)}}{v - u_j^{(1)}} \mathscr{B}^{(0)}(\boldsymbol{u}_{u_j^{(1)} \to v}^{(1)}) \\
&\quad \times \operatorname*{Res}_{w \to u_j^{(1)}} \big( f_q(w; \boldsymbol{u}^{(1)}) a(w) + f_{q^{-1}}(w; \boldsymbol{u}^{(1)}) d(w) \big) \cdot \eta,
\end{aligned}$$

which, upon evaluation, yields the wanted result. $\qquad\square$

We now turn to the $U_{q^2}(\mathfrak{so}_{2n+1})$-symmetric spin chains with $n \ge 2$. We define the *level-n transfer matrix* by

$$\tau^{(n)}(v) := \operatorname{tr}_a T_a^{(n)}(v).$$

Moreover, for each $1 \le k \le n-1$, we define the *nested level-$k$ transfer matrices* by

$$\tau^{(k)}(v; \boldsymbol{u}^{(k+1\dots n)}) := \text{tr}_a \, T_a^{(k)}(v; \boldsymbol{u}^{(k+1\dots n)}),$$

$$\widetilde{\tau}^{(k)}(v; \boldsymbol{u}^{(k+1\dots n)}) := \text{tr}_a \, \widetilde{T}_a^{(k)}(v; \boldsymbol{u}^{(k+1\dots n)}).$$

Let $\equiv$ denote equality of operators in the nested space $L^{(k)}$ and set $\boldsymbol{u}^{(n+1)} := \boldsymbol{\rho}$. It follows from the results of Section 3.1 that

$$\widetilde{\tau}^{(k)}(v; \boldsymbol{u}^{(k+1\dots n)}) \equiv \varepsilon^{(k+1)} \mu^{(k)}(v; \boldsymbol{u}^{(k+2)}) \tau^{(k)}(q^{-4}v; \boldsymbol{u}^{(k+1\dots n)}), \tag{92}$$

where $\mu^{(k)}(v; \boldsymbol{u}^{(k+2)})$ is given by

|  | $L^V$ | $L^S$ |
|---|---|---|
| $\mu^{(n-1)}(v; \boldsymbol{u}^{(n+1)})$ | $f_{q^2}(v; \boldsymbol{\rho})$ | $f_q(v; \boldsymbol{\rho})$ |
| $\mu^{(k)}(v; \boldsymbol{u}^{(k+2)})$ | $f_{q^2}(v; \boldsymbol{u}^{(k+2)})$ | $f_q(v; \boldsymbol{\rho}) f_{q^2}(v; \boldsymbol{u}^{(k+2)})$ |

We extend the prescription above to include the $k = 0$ case. The Theorem below is the main result of this section.

**Theorem 3.3.** *The Bethe vector $\Phi^{(n)}(\boldsymbol{u}^{(1\dots n)})$ with $n \ge 2$ is an eigenvector of $\tau^{(n)}(v)$ with the eigenvalue*

$$\Lambda^{(n)}(v; \boldsymbol{u}^{(1\dots n)}) := \sum_{\boldsymbol{i}} f_q(q^{p_0(\boldsymbol{i})}v; \boldsymbol{u}^{(1)})$$

$$\times \prod_{j=1}^n \varepsilon_{i_j}^{(j)} \Big( \mu^{(j-1)}(q^{p_j(\boldsymbol{i})}v; \boldsymbol{u}^{(j+1)}) f_{q^{-2}}(q^{p_j(\boldsymbol{i})}v; \boldsymbol{u}^{(j)}) \Big)^{\frac{1}{2}(1+i_j)}, \tag{93}$$

*where $p_j(\boldsymbol{i}) = -2\sum_{k=j+1}^n (1 + i_k)$ provided*

$$\operatorname*{Res}_{v \to u_j^{(k)}} \Lambda^{(n)}(v; \boldsymbol{u}^{(1\dots n)}) = 0 \quad \text{for} \quad 1 \le j \le m_k, \, 1 \le k \le n. \tag{94}$$

The explicit form of the Bethe equations (94) is

$$\prod_{i=1}^{m_1} \frac{q u_j^{(1)} - q^{-1} u_i^{(1)}}{q^{-1} u_j^{(1)} - q u_i^{(1)}} \prod_{i=1}^{m_2} \frac{u_j^{(1)} - u_i^{(2)}}{q^2 u_j^{(1)} - q^{-2} u_i^{(2)}} = -\varepsilon^{(1)} \lambda_1(u_j^{(1)}), \tag{95}$$

$$\prod_{i=1}^{m_{k-1}} \frac{q^{-2} u_j^{(k)} - q^2 u_i^{(k-1)}}{u_j^{(k)} - u_i^{(k-1)}} \prod_{i=1}^{m_k} \frac{q^2 u_j^{(k)} - q^{-2} u_i^{(k)}}{q^{-2} u_j^{(k)} - q^2 u_i^{(k)}} \prod_{i=1}^{m_{k+1}} \frac{u_j^{(k)} - u_i^{(k+1)}}{q^2 u_j^{(k)} - q^{-2} u_i^{(k+1)}} = -\frac{\varepsilon^{(k)}}{\varepsilon^{(k-1)}}, \tag{96}$$

$$\prod_{i=1}^{m_{n-1}} \frac{q^{-2} u_j^{(n)} - q^2 u_i^{(n-1)}}{u_j^{(n)} - u_i^{(n-1)}} \prod_{i=1}^{m_n} \frac{q^2 u_j^{(n)} - q^{-2} u_i^{(n)}}{q^{-2} u_j^{(n)} - q^2 u_i^{(n)}} = -\frac{\varepsilon^{(n)}}{\varepsilon^{(n-1)}} \lambda_n(u_j^{(n)}), \tag{97}$$

where $\lambda_1(v) = 1$ or $f_q(v; \boldsymbol{\rho})$ and $\lambda_n(v) = f_{q^2}(v; \boldsymbol{\rho})$ or $1$ when $L^{(n)} = L^V$ or $L^S$, respectively.

*Proof of Theorem 3.3.* We begin by rewriting the "AB" and "DB" exchange relations, (67) and (68), in a more convenient form. Lemma 2.13 implies that

$$R_{21}^{(n-1,n-1)}(u, v) = \frac{(R_{12}^{(n-1,n-1)}(q^{4n-6}v, u))^{w_2}}{h^{(n-1)}(v, u)}.$$

Combining this identity with (5), (67) and (85) yields the wanted form of the "AB" exchange relation,

$$
\begin{aligned}
A_a^{(n-1)}&(v)\,b_{\dot{a}_i^n\ddot{a}_i^n}^{(n-1,n-1)}(u_i^{(n)}) \\
&= b_{\dot{a}_i^n\ddot{a}_i^n}^{(n-1,n-1)}(u_i^{(n)})\left(\frac{f_q(v,u_i^{(n)})}{h^{(n-1)}(v,u_i^{(n)})}\right. \\
&\qquad\qquad\qquad \times R_{a\dot{a}_i^n}^{\prime(n-1,n-1)}(q^{4n-6}v,u_i^{(n)})A_a^{(n-1)}(v)R_{a\ddot{a}_i^n}^{\prime(n-1,n-1)}(q^4v,u_i^{(n)})\bigg) \\
&\quad -\frac{v/u_i^{(n)}}{v-u_i^{(n)}}\,b_{\dot{a}_i^n\ddot{a}_i^n}^{(n-1,n-1)}(v)\,\operatorname*{Res}_{w\to u_i^{(n)}}\left(\frac{f_q(w,u_i^{(n)})}{h^{(n-1)}(w,u_i^{(n)})}\right. \\
&\qquad\qquad\qquad \times R_{a\dot{a}_i^n}^{\prime(n-1,n-1)}(q^{4n-6}w,u_i^{(n)})A_a^{(n-1)}(w)R_{a\ddot{a}_i^n}^{\prime(n-1,n-1)}(q^4w,u_i^{(n)})\bigg). \qquad (98)
\end{aligned}
$$

Applying the same arguments and the identity

$$
f_{q^{-1}}(v,u_i^{(k+1)}) = f_{q^{-2}}(v,u_i^{(k+1)})f_q(q^{-4}v,u_i^{(k+1)})
$$

to (68) we find the wanted form of the "DB" exchange relation,

$$
\begin{aligned}
D_a^{(n-1)}&(v)\,b_{\dot{a}_i^n\ddot{a}_i^n}^{(n-1)}(u_i^{(n)}) \\
&= b_{\dot{a}_i^n\ddot{a}_i^n}^{(n-1,n-1)}(u_i^{(n)})\left(f_{q^{-2}}(v,u_i^{(k+1)})\frac{f_q(q^{-4}v,u_i^{(n)})}{h^{(n-1)}(q^{-4}v,u_i^{(n)})}\right. \\
&\qquad\qquad\qquad \times R_{a\dot{a}_i^n}^{\prime(n-1,n-1)}(q^{4n-10}v,u_i^{(n)})D_a^{(n-1)}(v)R_{a\ddot{a}_i^n}^{\prime(n-1,n-1)}(v,u_i^{(n)})\bigg) \\
&\quad -\frac{v/u_i^{(n)}}{v-u_i^{(n)}}\,b_{\dot{a}_i^n\ddot{a}_i^n}^{(n-1,n-1)}(v)\,\operatorname*{Res}_{w\to u_i^{(n)}}\left(f_{q^{-2}}(w,u_i^{(k+1)})\frac{f_q(q^{-4}w,u_i^{(n)})}{h^{(n-1)}(q^{-4}w,u_i^{(n)})}\right. \\
&\qquad\qquad\qquad \times R_{a\dot{a}_i^n}^{\prime(n-1,n-1)}(q^{4n-10}w,u_i^{(n)})D_a^{(n-1)}(w)R_{a\ddot{a}_i^n}^{\prime(n-1,n-1)}(w,u_i^{(n)})\bigg). \qquad (99)
\end{aligned}
$$

Inspired by the exchange relations above we define a barred transfer matrix

$$
\begin{aligned}
\overline{\tau}^{(n-1)}(v;\boldsymbol{u}^{(n)}) := \frac{f_q(v;\boldsymbol{u}^{(n)})}{h^{(n-1)}(v;\boldsymbol{u}^{(n)})}\operatorname{tr}_a\left(A_a^{(n-1)}(v)\prod_{i=1}^{m_n}R_{a\ddot{a}_i^n}^{\prime(n-1,n-1)}(q^4v,u_i^{(n)})\right. \\
\left. \times \prod_{i=m_n}^{1}R_{a\dot{a}_i^n}^{\prime(n-1,n-1)}(q^{4n-6}v,u_i^{(n)})\right),
\end{aligned}
$$

which differs from $\tau^{(n-1)}(v;\boldsymbol{u}^{(n)})$ in (78) by the ordering of the $R$-matrices only. The ordering can be amended with the help of operator $X^{(n-1)}:=\prod_{i=1}^{m_n-1}X_i^{(n-1)}$ where

$$
X_i^{(n-1)} := \prod_{j=i+1}^{m_n}R_{\dot{a}_j^n\dot{a}_i^n}^{(n-1,n-1)}(u_j^{(n)},u_i^{(n)})\prod_{j=m_n}^{i+1}R_{\ddot{a}_j^n\dot{a}_i^n}^{(n-1,n-1)}(q^{4n-10}u_j^{(n)},u_i^{(n)}).
$$

In particular, $\overline{\tau}^{(n-1)}(v;\boldsymbol{u}^{(n-1)}) = X^{(n-1)}\tau^{(n-1)}(v;\boldsymbol{u}^{(n-1)})(X^{(n-1)})^{-1}$. Moreover, each $X_i^{(n-1)}$ acts as a scalar operator on $\Phi^{(n-1)}(\boldsymbol{u}^{(1...n-1)};\boldsymbol{u}^{(n)})$. Then, using the wanted exchange relations

above, Lemma 3.1, the standard symmetry arguments, the equality (92), and recalling that $\tau^{(n)}(v) = \mathrm{tr}_a(A_a^{(n-1)}(v) + D_a^{(n-1)}(v))$, we obtain

$$\tau^{(n)}(v)\Phi^{(n)}(\boldsymbol{u}^{(1\ldots n)}) = \mathscr{B}^{(n-1)}(\boldsymbol{u}^{(n)})X^{(n-1)}\tau^{(n)}(v;\boldsymbol{u}^{(n)})(X^{(n-1)})^{-1}\Phi^{(n-1)}(\boldsymbol{u}^{(1\ldots n-1)};\boldsymbol{u}^{(n)})$$

$$-\sum_{j=1}^{m_n}\frac{v/u_j^{(n)}}{v-u_j^{(n)}}\mathscr{B}^{(n-1)}(\boldsymbol{u}_{\sigma_j^{(n)},u_j^{(n)}\to v}^{(n)})X^{(n-1)}$$

$$\times \underset{w\to u_j^{(n)}}{\mathrm{Res}}\,\tau^{(n)}(w;\boldsymbol{u}_{\sigma_j^{(n)}}^{(n)})(X^{(n-1)})^{-1}\Phi^{(n-1)}(\boldsymbol{u}^{(1\ldots n-1)};\boldsymbol{u}_{\sigma_j^{(n)}}^{(n)}),$$

where

$$\tau^{(n)}(v;\boldsymbol{u}^{(n)}) := \tau^{(n-1)}(v;\boldsymbol{u}^{(n)}) + \varepsilon^{(n)}f_{q^{-2}}(v;\boldsymbol{u}^{(n)})\mu^{(n-1)}(v;\boldsymbol{u}^{(n+1)})\tau^{(n-1)}(q^{-4}v;\boldsymbol{u}^{(n)}).$$

Since $(X^{(n-1)})^{-1}$ acts as a scalar operator, we are only left to determine the action of $\tau^{(n)}(v;\boldsymbol{u}^{(n)})$ on $\Phi^{(n-1)}(\boldsymbol{u}^{(1\ldots n-1)};\boldsymbol{u}^{(n)})$. But $\Phi^{(n-1)}(\boldsymbol{u}^{(1\ldots n-1)};\boldsymbol{u}^{(n)}) \in L^{(n-1)}$ and thus we can use (82) and repeat the same arguments as above down the nesting. This gives a recurrence relation for the eigenvalue $\Lambda^{(n)}(v;\boldsymbol{u}^{(1\ldots n)})$:

$$\Lambda^{(k)}(v;\boldsymbol{u}^{(1\ldots k)};\boldsymbol{u}^{(k+1\ldots n)}) := \varepsilon_{-1}^{(k)}\Lambda^{(k-1)}(v;\boldsymbol{u}^{(1\ldots k-1)};\boldsymbol{u}^{(k\ldots n)})$$

$$+ \varepsilon_{+1}^{(k)}f_{q^{-2}}(v,\boldsymbol{u}^{(k)})\mu^{(k-1)}(v;\boldsymbol{u}^{(k+1)})\Lambda^{(k-1)}(q^{-4}v;\boldsymbol{u}^{(1\ldots k-1)};\boldsymbol{u}^{(k\ldots n)}),$$

where $\Lambda^{(1)}(v;\boldsymbol{u}^{(1)};\boldsymbol{u}^{(2\ldots n)}) := \varepsilon_{-1}^{(1)}f_q(v;\boldsymbol{u}^{(1)}) + \varepsilon_{+1}^{(1)}f_{q^{-1}}(v,\boldsymbol{u}^{(1)})\mu^{(0)}(v;\boldsymbol{u}^{(2)})$. Solving this recurrence relation yields the wanted result. $\qquad\square$

## 4 Algebraic Bethe Ansatz for $U_q(\mathfrak{so}_{2n+2})$-symmetric spin chains

In this section we study spectrum of $U_q(\mathfrak{so}_{2n+2})$-symmetric spin chains with the *full quantum space* given by

$$L^{(n)} = L^V := (\mathbb{C}^{2n+2})^{\otimes\ell} \quad \text{or} \quad L^{(n)} = L^{\pm S} := (V^{\pm(n)})^{\otimes\ell}. \tag{100}$$

Our approach will be very similar to that in Section 3, thus most of the notation will carry through with minor adjustments only.

### 4.1 Quantum spaces and monodromy matrices

Choose $m_\pm, m_2, \ldots, m_n \in \mathbb{Z}_{\geq 0}$, the excitation, or magnon, numbers. For each $m_k$ assign an $m_k$-tuple $\boldsymbol{u}^{(k)} := (u_1^{(k)}, \ldots, u_{m_k}^{(k)})$ of non-zero complex parameters, that will accommodate Bethe roots. We will write $u_i^\pm = u_i^{(\pm)}$ and say that $u_i^\pm$ are level-1 parameters. Accordingly, we set $m_1 := m_+ + m_-$ to be the number of level-1 excitations. Then, for each $2 \leq k < n$, we introduce three $m_k$-tuples of labels, $\dot{\boldsymbol{a}} = (\dot{a}_1^k, \ldots, \dot{a}_{m_k}^k)$, $\ddot{\boldsymbol{a}} = (\ddot{a}_1^k, \ldots, \ddot{a}_{m_k}^k)$, and $\boldsymbol{a} = (a_1^k, \ldots, a_{m_k}^k)$. For each label $\dot{a}_i^k$ and $\ddot{a}_i^k$ we associate a copy of $V^{[+](k-1)}$ and $V^{-(k-1)}$, respectively, where $[+] = +/-$ if $k-1$ is odd/even. We then identify subspaces $W_{a_i^k} \subset V_{\dot{a}_i^k}^{[+](k-1)} \otimes V_{\ddot{a}_i^k}^{-(k-1)}$, isomorphic to $\mathbb{C}^{2k}$, in the following way. Let $\eta_{a_i^k} \in V_{\dot{a}_i^k}^{[+](k-1)} \otimes V_{\ddot{a}_i^k}^{-(k-1)}$ be a highest vector as per (61). Then $W_{a_i^k} \cong U_q(\mathfrak{so}_{2k}) \cdot \eta_{a_i^k}$, as a vector space.

For each $2 \leq k < n$ we recurrently define the *nested level-k quantum space $L^{(k)}$* in a similar way as we did in Section 3.1, that is

$$L^{(k)} := (L^{(k+1)})^0 \otimes W_{a_1^{k+1}} \otimes \cdots \otimes W_{a_{m_{k+1}}^{k+1}},$$

where $(L^{(k+1)})^0 \subset L^{(k+1)}$ is the *level-*$(k+1)$ *vacuum subspace* such that each individual tensorand is a $U_q(\mathfrak{so}_{2k}) \subset U_q(\mathfrak{so}_{2k+2})$ stable subspace annihilated by $\ell^+_{i,k+2}$ with $-k-2 \leq i \leq k+1$. In particular, $(L^{(k+1)})^0 \cong \mathbb{C}$ or $(V^{\pm(k)})^{\otimes \ell}$ when $L^{(n)} = L^V$ or $L^{\pm S}$, respectively. Finally, we define the *nested level-*1 *quantum space* to be

$$L^{(1)} := (L^{(2)})^0 \otimes V^{+(1)}_{\ddot{a}^{(2)}_1} \otimes V^{-(1)}_{\ddot{a}^{(2)}_1} \otimes \cdots \otimes V^{+(1)}_{\ddot{a}^{(2)}_{m_2}} \otimes V^{-(1)}_{\ddot{a}^{(2)}_{m_2}}. \tag{101}$$

We are now ready to introduce monodromy matrices. The diagonal "twist" matrix that we will need is

$$\mathcal{E}^{\pm(n)} := \sum_i \varepsilon^{(1)}_{i_1} \cdots \varepsilon^{(n)}_{i_n} e^{(\epsilon)}_{i_1 i_1} \hat{\otimes} e^{(1)}_{i_2 i_2} \hat{\otimes} \cdots \hat{\otimes} e^{(1)}_{i_n i_n} \in \operatorname{End}(V^{\pm(n)}),$$

where $\epsilon = \pm/\mp$ if $(-1)^{n-1} i_2 \cdots i_n = +1/-1$. We define the even and odd *level-n monodromy matrices* with entries acting on the level-$n$ quantum space $L^{(n)}$ by

$$T^{\pm(n)}_a(v) := \mathcal{E}^{\pm(n)} T^{\pm(n)}_{a1}(v) \cdots T^{\pm(n)}_{a\ell}(v), \tag{102}$$

where $T^{\pm(n)}_{ai}(v) = R^{\pm(n)}_{ai}(v, \rho_i)$ or $R^{\pm+(n,n)}_{ai}(q^2 v, \rho_i)$ or $R^{\pm-(n,n)}_{ai}(q^2 v, \rho_i)$ when $L^{(n)} = L^V$ or $L^{+S}$ or $L^{-S}$, respectively. Then, for each $1 \leq k < n$, we recurrently define the even and odd *nested level-k monodromy matrices* with entries acting on the level-$k$ quantum space $L^{(k)}$ by

$$T^{\pm(k)}_a(v; \boldsymbol{u}^{(k+1\ldots n)}) := \frac{(f_q(v; \boldsymbol{u}^{(k+1)}))^{\frac{1 \pm 1}{2}}}{h^{\pm(k/2)}(v; \boldsymbol{u}^{(k+1)})} A^{\pm(k)}_a(v; \boldsymbol{u}^{(k+2\ldots n)})$$

$$\times \prod_{i=1}^{m_{k+1}} R^{\pm-(k,k)}_{a\ddot{a}^{k+1}_i}(q^2 v, u^{(k+1)}_i) R^{\pm[+](k,k)}_{a\dot{a}^{k+1}_i}(q^{2k} v, u^{(k+1)}_i)$$

$$\overset{k \geq 2}{\equiv} A^{\pm(k)}_a(v; \boldsymbol{u}^{(k+2\ldots n)}) \prod_{i=1}^{m_{k+1}} R^{\pm(k)}_{aa^{k+1}_i}(v, u^{(k+1)}_i), \tag{103}$$

$$\widetilde{T}^{\mp(k)}_a(v; \boldsymbol{u}^{(k+1\ldots n)}) := \frac{(f_q(q^{-2} v; \boldsymbol{u}^{(k+1)}))^{\frac{1 \mp 1}{2}}}{h^{\mp(k/2)}(q^{-2} v; \boldsymbol{u}^{(k+1)})} D^{\mp(k)}_a(v; \boldsymbol{u}^{(k+2\ldots n)})$$

$$\times \prod_{i=1}^{m_{k+1}} R^{\mp-(k,k)}_{a\ddot{a}^{k+1}_i}(v, u^{(k+1)}_i) R^{\mp[+](k,k)}_{a\dot{a}^{k+1}_i}(q^{2k-2} v, u^{(k+1)}_i)$$

$$\overset{k \geq 2}{\equiv} D^{\mp(k)}_a(v; \boldsymbol{u}^{(k+2\ldots n)}) \prod_{i=1}^{m_{k+1}} R^{\mp(k)}_{aa^{k+1}_i}(q^{-2} v, u^{(k+1)}_i), \tag{104}$$

where $[+] = +/-$ if $k$ is odd/even, and

$$A^{\pm(k)}_a(v; \boldsymbol{u}^{(k+2\ldots n)}) = \left[ T^{\pm(k+1)}_a(v; \boldsymbol{u}^{(k+2\ldots n)}) \right]_{-1,-1}, \tag{105}$$

$$D^{\mp(k)}_a(v; \boldsymbol{u}^{(k+2\ldots n)}) = \left[ T^{\pm(k+1)}_a(v; \boldsymbol{u}^{(k+2\ldots n)}) \right]_{+1,+1}, \tag{106}$$

and $\overset{k \geq 2}{\equiv}$ denotes equality of operators in the space $L^{(k)}$ when $k \geq 2$, subject to a suitable identification of the subspaces $W_{a^{k+1}_i} \subset V^{[+](k)}_{\dot{a}^{k+1}_i} \otimes V^{-(k)}_{\ddot{a}^{k+1}_i}$ and copies of $\mathbb{C}^{2k+2}$, as per Lemma 2.21. When $k = 1$, the expressions above simplify to those in (111–114) stated below because of the identity $R^{\pm\mp(1,1)}(u, v) = I^{\pm\mp(1,1)}$.

The nested monodromy matrices span the following nesting tree:

$$T_a^{\pm(n)}(v) \Bigg\langle \begin{array}{c} T_a^{\pm(n-1)}(v;\boldsymbol{u}^{(n)}) \Big\langle \begin{array}{c} T_a^{\pm(n-2)}(v;\boldsymbol{u}^{(n-1,n)}) \Big\langle \begin{array}{c} \cdots \Big\langle \begin{array}{c} T_a^{\pm(1)}(v;\boldsymbol{u}^{(2\ldots n)}) \\ \widetilde{T}_a^{\mp(1)}(v;\boldsymbol{u}^{(2\ldots n)}) \end{array} \\ \cdots \end{array} \\ \widetilde{T}_a^{\mp(n-2)}(v;\boldsymbol{u}^{(n-1,n)}) \end{array} \\ \widetilde{T}_a^{\mp(n-1)}(v;\boldsymbol{u}^{(n)}) \end{array}$$

By the same arguments as in the previous case, it will be sufficient to focus on the non-tilded monodromy matrices at each level of nesting. In particular, we have the following equalities of operators (105) and (106) in the spaces $L^{(n-1)}$ and $L^{(k)}$ with $1 \le k < n-1$, subject to the choice of the full quantum space $L^{(n)}$:

|  | $L^V$ | $L^{+S}$ | $L^{-S}$ |
|---|---|---|---|
| $A_a^{\pm(n-1)}(v)/\varepsilon_{-1}^{(n)}$ | $\mathcal{E}_a^{\pm(n-1)}$ | $T_a^{\pm(n-1)}(v)$ | $T_a^{\pm(n-1)}(v)$ |
| $D_a^{-(n-1)}(v)/\varepsilon_{+1}^{(n)}$ | $f_q(v;\boldsymbol{\rho})\mathcal{E}_a^{-(n-1)}$ | $f_q(v;\boldsymbol{\rho})T_a^{-(n-1)}(\tilde{v})$ | $T_a^{-(n-1)}(\tilde{v})$ |
| $D_a^{+(n-1)}(v)/\varepsilon_{+1}^{(n)}$ | $f_q(v;\boldsymbol{\rho})\mathcal{E}_a^{+(n-1)}$ | $T_a^{-(n-1)}(\tilde{v})$ | $f_q(v;\boldsymbol{\rho})T_a^{-(n-1)}(\tilde{v})$ |
| $A_a^{\pm(k)}(v;\boldsymbol{u}^{(k+2\ldots n)})/\tilde{\varepsilon}_{-1}^{(k+1)}$ | $\mathcal{E}_a^{\pm(k)}$ | $T_a^{\pm(k)}(v)$ | $T_a^{\pm(k)}(v)$ |
| $D_a^{-(k)}(v;\boldsymbol{u}^{(k+2\ldots n)})/\tilde{\varepsilon}_{+1}^{(k+1)}$ | $f_q(v;\boldsymbol{u}^{(k+2)})\mathcal{E}_a^{-(k)}$ | $g_q(v)T_a^{-(k)}(\tilde{v})$ | $f_q(v;\boldsymbol{u}^{(k+2)})T_a^{-(k)}(\tilde{v})$ |
| $D_a^{+(k)}(v;\boldsymbol{u}^{(k+2\ldots n)})/\tilde{\varepsilon}_{+1}^{(k+1)}$ | $f_q(v;\boldsymbol{u}^{(k+2)})\mathcal{E}_a^{+(k)}$ | $f_q(v;\boldsymbol{u}^{(k+2)})T_a^{+(k)}(\tilde{v})$ | $g_q(v)T_a^{+(k)}(\tilde{v})$ |

Here $\tilde{\varepsilon}_{\mp1}^{(k+1)} = \varepsilon_{-1}^{(n)}\cdots\varepsilon_{-1}^{(k+2)}\varepsilon_{\mp1}^{(k+1)}$, $g_q(v) = f_q(v;\boldsymbol{\rho})f_q(v;\boldsymbol{u}^{(k+2)})$, $\tilde{v} = q^{-2}v$, and the operators $T_a^{\pm(n-1)}(v)$ and $T_a^{\pm(k)}(v)$ are defined in the same way as $T_a^{\pm(n)}(v)$, viz. (102). It is now easy to see that, for $\epsilon_a, \epsilon_b = \pm$,

$$R_{ab}^{\epsilon_a\epsilon_b(k,k)}(v,w)\,T_a^{\epsilon_a(k)}(v;\boldsymbol{u}^{(k+1\ldots n)})\,T_b^{\epsilon_b(k)}(w;\boldsymbol{u}^{(k+1\ldots n)})$$
$$\equiv T_b^{\epsilon_b(k)}(w;\boldsymbol{u}^{(k+1\ldots n)})\,T_a^{\epsilon_a(k)}(v;\boldsymbol{u}^{(k+1\ldots n)})\,R_{ab}^{\epsilon_a\epsilon_b(k,k)}(v,w)\,. \tag{107}$$

Thus entries of $T_a^{\pm(k)}(v;\boldsymbol{u}^{(k+1\ldots n)})$ in the space $L^{(k)}$ satisfy the exchange relations given by Lemma 2.23. In other words, operators $T_a^{+(k)}(v;\boldsymbol{u}^{(k+1\ldots n)})$ and $T_a^{-(k)}(v;\boldsymbol{u}^{(k+1\ldots n)})$ are even and odd monodromy matrices for a nested $U_q(\mathfrak{so}_{2k+2})$-symmetric spin chain with the full quantum space $L^{(k)}$.

## 4.2 Creation operators and Bethe vectors

We now introduce $m_k$-magnon creation operators. We will make use of the following notation:

$$b^{\mp}(v;\boldsymbol{u}^{(2\ldots n)}) := \left[T_a^{\pm(1)}(v;\boldsymbol{u}^{(2\ldots n)})\right]_{-1,+1},$$
$$B_a^{\mp(k-1)}(v;\boldsymbol{u}^{(k+1\ldots n)}) := \left[T_a^{\pm(k)}(v;\boldsymbol{u}^{(k+1\ldots n)})\right]_{-1,+1}.$$

We define the *level-1 creation operator* by

$$\mathscr{B}^{(0)}(\boldsymbol{u}^{(1)};\boldsymbol{u}^{(2\ldots n)}) := \prod_{i=m_+}^{1} b^+(u_i^+;\boldsymbol{u}^{(2\ldots n)}) \prod_{i=m_-}^{1} b^-(u_i^-;\boldsymbol{u}^{(2\ldots n)}).$$

For each $2 \leq k \leq n$ we define the *level-k creation operator* by

$$\mathscr{B}^{(k-1)}(\boldsymbol{u}^{(k)};\boldsymbol{u}^{(k+1\ldots n)}) := \prod_{i=m_k}^{1} \mathcal{b}_{\dot{a}_i^k \ddot{a}_i^k}^{[+]-(k-1,k-1)}(u_i^{(k)};\boldsymbol{u}^{(k+1\ldots n)}),$$

where

$$\mathcal{b}_{\dot{a}_i^k \ddot{a}_i^k}^{[+]-(k-1,k-1)}(u_i^{(k)};\boldsymbol{u}^{(k+1\ldots n)}) := \chi_{\dot{a}_i^k \ddot{a}_i^k}^{-(k-1)}\left(B_{a_i^k}^{-(k-1)}(u_i^{(k)};\boldsymbol{u}^{(k+1\ldots n)})\right), \tag{108}$$

with $\chi_{\dot{a}_i^k \ddot{a}_i^k}^{-(k-1)} : \mathrm{Hom}(V_{a_i^k}^{-(n-1)}, V_{a_i^k}^{+(n-1)}) \to (V_{\dot{a}_i^k}^{[+](k-1)})^* \otimes (V_{\ddot{a}_i^k}^{-(k-1)})^*$ defined via (8).

We define the nested vacuum vector $\eta$ and the Bethe vectors $\Phi^{(k)}(\boldsymbol{u}^{(1\ldots k)};\boldsymbol{u}^{(k+1\ldots n)})$ with $1 \leq k \leq n$ in the same way as before, that is, by (86)–(87), except that $\eta_{a_i^k}$ with $2 < k \leq n$ are now given by (61) and $\eta_{a_i^2} = e_{-1}^{(+)} \otimes e_{-1}^{(-)}$. We set $\mathfrak{S}_{\boldsymbol{m}_{1\ldots k}} := \mathfrak{S}_{m_+} \times \mathfrak{S}_{m_-} \times \mathfrak{S}_{m_2} \times \cdots \times \mathfrak{S}_{m_k}$ and define its action on $\Phi^{(k)}(\boldsymbol{u}^{(1\ldots k)};\boldsymbol{u}^{(k+1\ldots n)})$ in the same way as we did before. The proof of the Lemma below is analogous to that of Lemma 3.1.

**Lemma 4.1.** *The Bethe vector $\Phi^{(k)}(\boldsymbol{u}^{(1\ldots k)};\boldsymbol{u}^{(k+1\ldots n)})$ is invariant under the action of $\mathfrak{S}_{\boldsymbol{m}_{1\ldots k}}$.*

## 4.3 Transfer matrices, their eigenvalues, and Bethe equations

We begin with the first non-trivial case, the $U_q(\mathfrak{so}_4)$-symmetric spin chain. In this case the monodromy matrices $T_a^{+(1)}(v)$ and $T_a^{-(1)}(w)$ commute for any values of $v$ and $w$. Thus the spin chain effectively factorises into two XXZ spin chains with the even and odd transfer matrices given by

$$\tau^{\pm(1)}(v) := \mathrm{tr}_a\, T_a^{\pm(1)}(v).$$

When $L^{(1)} = L^V$, the vacuum vector is $\eta = e_{-2} \otimes \cdots \otimes e_{-2}$. It is a unique joined highest vector of both $T_a^{+(1)}(v)$ and $T_a^{-(1)}(v)$. The operator $\mathscr{B}^{(0)}(\boldsymbol{u}^{(1)})$ acting on $\eta$ creates $m_+$ even and $m_-$ odd excitations. When $L^{(1)} = L^{+S}$, the vacuum vector is $\eta = e_{-1}^{(+)} \otimes \cdots \otimes e_{-1}^{(+)}$. It is now a highest vector of $T_a^{+(1)}(v)$ and a singular vector of $T_a^{-(1)}(v)$, i.e. $\eta$ is annihilated by the off-diagonal matrix entries of $T_a^{-(1)}(v)$. Thus the operator $\mathscr{B}^{(0)}(\boldsymbol{u}^{(1)})$ now creates $m_+$ even excitations only. Lastly, when $L^{(1)} = L^{-S}$, the vacuum vector is $\eta = e_{-1}^{(-)} \otimes \cdots \otimes e_{-1}^{(-)}$. It is a highest vector of $T_a^{-(1)}(v)$ and a singular vector of $T_a^{+(1)}(v)$. Thus the operator $\mathscr{B}^{(0)}(\boldsymbol{u}^{(1)})$ creates $m_-$ odd excitations only.

The Theorem below follows by the same arguments as Theorem 3.2.

**Theorem 4.2.** *The Bethe vector $\Phi^{(1)}(\boldsymbol{u}^{(1)})$ is an eigenvector of $\tau^{\pm(1)}(v)$ with the eigenvalue*

$$\Lambda^{\pm(1)}(v;\boldsymbol{u}^{\pm}) := \varepsilon_{-1}^{(1)} f_q(v;\boldsymbol{u}^{\pm}) + \varepsilon_{+1}^{(1)} f_{q^{-1}}(v;\boldsymbol{u}^{\pm}) f_{q^{-1}}(v;\boldsymbol{\rho}), \tag{109}$$

*provided*

$$\operatorname*{Res}_{v \to u_j^{\pm}} \Lambda^{\pm(1)}(v;\boldsymbol{u}^{\pm}) = 0 \quad for \quad 1 \leq j \leq m_{\pm}. \tag{110}$$

The explicit form of the Bethe equations (110) is

$$\prod_{i=1}^{m_{\pm}} \frac{qu_j^{\pm} - q^{-1}u_i^{\pm}}{q^{-1}u_j^{\pm} - qu_i^{\pm}} = -\varepsilon^{(1)} \prod_{i=1}^{\ell} \frac{qu_j^{\pm} - q^{-1}\rho_i}{u_j^{\pm} - \rho_i}.$$

We note that these are two independent sets of Bethe equations, for $\boldsymbol{u}^+$ and for $\boldsymbol{u}^-$, and the excitation numbers $m_+$ and $m_-$ depend on the choice of $L^{(1)}$.

We now turn our focus to the $U_q(\mathfrak{so}_6)$-symmetric spin chain. This chain can be viewed as a generalised ($U_q(\mathfrak{gl}_4)$-symmetric) XXZ spin chain. We begin by addressing the corresponding

nested $U_q(\mathfrak{so}_4)$-symmetric spin chain. The nested level-1 quantum space is given by (101). The nested vacuum vector takes the form

$$\eta = \eta_1 \otimes \cdots \otimes \eta_\ell \otimes e_{-1}^{(+)} \otimes e_{-1}^{(-)} \otimes \cdots \otimes e_{-1}^{(+)} \otimes e_{-1}^{(-)}.$$

The nested level-1 monodromy matrices that we will need are (cf. (103) and (104)):

$$T_a^{+(1)}(v; \boldsymbol{u}^{(2)}) = A_a^{+(1)}(v) \prod_{i=1}^{m_2} R_{a\dot{a}_i^2}^{++(1,1)}(q^2 v, u_i^{(2)}), \tag{111}$$

$$T_a^{-(1)}(v; \boldsymbol{u}^{(2)}) = A_a^{-(1)}(v) \prod_{i=1}^{m_2} R_{a\ddot{a}_i^2}^{--(1,1)}(q^2 v, u_i^{(2)}), \tag{112}$$

$$\widetilde{T}_a^{+(1)}(v; \boldsymbol{u}^{(2)}) = D_a^{+(1)}(v) \prod_{i=1}^{m_2} R_{a\dot{a}_i^2}^{++(1,1)}(v, u_i^{(2)}), \tag{113}$$

$$\widetilde{T}_a^{-(1)}(v; \boldsymbol{u}^{(2)}) = D_a^{-(1)}(v) \prod_{i=1}^{m_2} R_{a\ddot{a}_i^2}^{--(1,1)}(v, u_i^{(2)}), \tag{114}$$

where $A_a^{\pm(1)}(v) = [T_a^{\pm(2)}(v)]_{-1,-1}$ and $D_a^{\mp(1)}(v) = [T_a^{\pm(2)}(v)]_{+1,+1}$. The corresponding nested transfer matrices are

$$\tau^{\pm(1)}(v; \boldsymbol{u}^{(2)}) = \mathrm{tr}_a \, T_a^{\pm(1)}(v; \boldsymbol{u}^{(2)}), \qquad \widetilde{\tau}^{\pm(1)}(v; \boldsymbol{u}^{(2)}) = \mathrm{tr}_a \, \widetilde{T}_a^{\pm(1)}(v; \boldsymbol{u}^{(2)}).$$

Let $\equiv$ denote equality of operators in the nested space $L^{(1)}$. Then

$$\widetilde{\tau}^{\pm(1)}(v; \boldsymbol{u}^{(2)}) \equiv \varepsilon^{(1)} \mu^{\pm(1)}(v) \tau^{\pm(1)}(q^{-2}v; \boldsymbol{u}^{(2)}). \tag{115}$$

We also have that

$$a^{\pm}(v; \boldsymbol{u}^{(2)}) \cdot \eta = \eta, \qquad d^{\pm}(v; \boldsymbol{u}^{(2)}) \cdot \eta = f_q(v; \boldsymbol{u}^{(2)}) \lambda_{\pm}(v) \eta.$$

Here $\mu^{\pm(1)}(v)$ and $\lambda_{\pm}(v)$ are given by

|  | $L^V$ | $L^{+S}$ | $L^{-S}$ |
|---|---|---|---|
| $\mu^{+(1)}(v)$ | $f_q(v; \boldsymbol{\rho})$ | $1$ | $f_q(v; \boldsymbol{\rho})$ |
| $\mu^{-(1)}(v)$ | $f_q(v; \boldsymbol{\rho})$ | $f_q(v; \boldsymbol{\rho})$ | $1$ |
| $\lambda_+(v)$ | $1$ | $f_q(v; \boldsymbol{\rho})$ | $1$ |
| $\lambda_-(v)$ | $1$ | $1$ | $f_q(v; \boldsymbol{\rho})$ |

The Proposition below follows by the standard arguments.

**Proposition 4.3.** *The nested Bethe vector $\Phi^{(1)}(\boldsymbol{u}^{\pm}; \boldsymbol{u}^{(2)})$ is an eigenvector of $\tau^{\pm(1)}(v; \boldsymbol{u}^{(2)})$ with the eigenvalue*

$$\Lambda^{\pm(1)}(v; \boldsymbol{u}^{\pm}; \boldsymbol{u}^{(2)}) := \varepsilon_{-1}^{(1)} f_q(v; \boldsymbol{u}^{\pm}) + \varepsilon_{+1}^{(1)} f_{q^{-1}}(v; \boldsymbol{u}^{\pm}) f_q(v; \boldsymbol{u}^{(2)}) \lambda_{\pm}(v), \tag{116}$$

*provided*

$$\operatorname*{Res}_{v \to u_j^{\pm}} \Lambda^{\pm(1)}(v; \boldsymbol{u}^{\pm}; \boldsymbol{u}^{(2)}) = 0 \quad \textit{for} \quad 1 \le j \le m^{\pm}. \tag{117}$$

We are now ready to address the full $U_q(\mathfrak{so}_6)$-symmetric spin chain. We define its transfer matrices by

$$\tau^{\pm(2)}(v) := \mathrm{tr}_a \, T_a^{\pm(2)}(v).$$

The Theorem below is the first main result of this section.

**Theorem 4.4.** *The Bethe vector $\Phi^{(2)}(\boldsymbol{u}^{(1,2)})$ is an eigenvector of $\tau^{\pm(2)}(v)$ with the eigenvalue*

$$
\begin{aligned}
\Lambda^{\pm(2)}(v;\boldsymbol{u}^{(1,2)}) :=\; & \varepsilon_{-1}^{(2)}\Big(\varepsilon_{-1}^{(1)} f_q(v;\boldsymbol{u}^{\pm}) + \varepsilon_{+1}^{(1)} f_q(v;\boldsymbol{u}^{(2)}) f_{q^{-1}}(v;\boldsymbol{u}^{\pm})\lambda_{\pm}(v)\Big) \\
& + \varepsilon_{+1}^{(2)}\mu^{\mp(1)}(v)\Big(\varepsilon_{-1}^{(1)} f_{q^{-1}}(v;\boldsymbol{u}^{(2)}) f_q(q^{-2}v;\boldsymbol{u}^{\mp}) \\
& \hspace{4cm} + \varepsilon_{+1}^{(1)} f_{q^{-1}}(q^{-2}v;\boldsymbol{u}^{\mp})\lambda_{\mp}(q^{-2}v)\Big),
\end{aligned}
\tag{118}
$$

*provided*

$$
\operatorname*{Res}_{v\to u_j^{(k)}} \Lambda^{\pm(2)}(v;\boldsymbol{u}^{(1,2)}) = 0 \quad for \quad 1\le j\le m_k,\; k=1,2.
\tag{119}
$$

The explicit form of the Bethe equations (119) is

$$
\prod_{i=1}^{m_{\pm}} \frac{qu_j^{\pm}-q^{-1}u_i^{\pm}}{q^{-1}u_j^{\pm}-qu_i^{\pm}} \prod_{i=1}^{m_2} \frac{u_j^{\pm}-u_i^{(2)}}{qu_j^{\pm}-q^{-1}u_i^{(2)}} = -\varepsilon^{(1)}\lambda_{\pm}(u_j^{\pm}),
\tag{120}
$$

$$
\prod_{i=1}^{m_+} \frac{q^{-1}u_j^{(2)}-qu_i^+}{u_j^{(2)}-u_i^+} \prod_{i=1}^{m_-} \frac{q^{-1}u_j^{(2)}-qu_i^-}{u_j^{(2)}-u_i^-} \prod_{i=1}^{m_2} \frac{qu_j^{(2)}-q^{-1}u_i^{(2)}}{q^{-1}u_j^{(2)}-qu_i^{(2)}} = -\frac{\varepsilon^{(2)}}{\varepsilon^{(1)}}\lambda_2(u_j^{(2)}),
\tag{121}
$$

where $\lambda_2$ is given by $\lambda_2(v)=f_q(v;\boldsymbol{\rho})$ or 1 when $L^{(2)}=L^V$ or $L^{\pm S}$, respectively.

*Proof of Theorem 4.4.* We start by rewriting the "AB" and "DB" exchange relations, (72) and (73), in a more convenient form. First, using Lemma 2.18, we deduce that

$$
R_{21}^{\pm\pm(1,1)}(u,v) = \frac{(R_{12}^{\pm\pm(1,1)}(q^2v,u))^{w_2}}{f_q(v,u)}.
$$

Then, repeating the same arguments as in the Proof of Theorem 3.3, we find the wanted exchange relations for $A_a^{+(1)}(v)$ and $D_a^{-(1)}(v)$ to be

$$
\begin{aligned}
& A_a^{+(1)}(v)\,\mathit{6}_{\acute{a}_i^2\ddot{a}_i^2}^{+-(1,1)}(u_i^{(2)}) \\
& = \mathit{6}_{\acute{a}_i^2\ddot{a}_i^2}^{+-(1,1)}(u_i^{(2)})\Big(R_{a\acute{a}_i^2}^{++(1,1)}(q^2v,u_i^{(2)})A_a^{+(1)}(v)\Big) \\
& \quad - \frac{v/u_i^{(2)}}{v-u_i^{(2)}}\,\mathit{6}_{\acute{a}_i^2\ddot{a}_i^2}^{+-(1,1)}(v)\operatorname*{Res}_{w\to u_i^{(2)}}\Big(R_{a\acute{a}_i^2}^{++(1,1)}(q^2w,u_i^{(2)})A_a^{+(1)}(w)\Big),
\end{aligned}
$$

$$
\begin{aligned}
& D_a^{-(1)}(v)\,\mathit{6}_{\acute{a}_i^2\ddot{a}_i^2}^{+-(1,1)}(u_i^{(2)}) \\
& = \mathit{6}_{\acute{a}_i^2\ddot{a}_i^2}^{+-(1,1)}(u_i^{(2)})\Big(f_{q^{-1}}(v,u_i^{(2)})D_a^{-(1)}(v)R_{a\ddot{a}_i^2}^{--(1,1)}(v,u_i^{(2)})\Big) \\
& \quad - \frac{v/u_i^{(2)}}{v-u_i^{(2)}}\,\mathit{6}_{\acute{a}_i^2\ddot{a}_i^2}^{+-(1,1)}(v)\operatorname*{Res}_{w\to u_i^{(2)}}\Big(f_{q^{-1}}(w,u_i^{(2)})D_a^{-(1)}(w)R_{a\ddot{a}_i^2}^{--(1,1)}(w,u_i^{(2)})\Big).
\end{aligned}
$$

Consequently, using Lemma 4.1, relation (115), and the standard symmetry arguments, we

find

$$\tau^{+(2)}(v)\Phi^{(2)}(\boldsymbol{u}^{(1,2)}) = \Big(\mathrm{tr}_a A_a^{+(1)}(v) + \mathrm{tr}_a D_a^{-(1)}(v)\Big)\mathscr{B}^{(1)}(\boldsymbol{u}^{(2)})\Phi^{(1)}(\boldsymbol{u}^{(1)})$$

$$= \mathscr{B}^{(1)}(\boldsymbol{u}^{(2)})\Big(\tau^{+(1)}(v;\boldsymbol{u}^{(2)})$$

$$+ f_{q^{-1}}(v;\boldsymbol{u}^{(2)})\mu^{-(1)}(v)\,\tau^{-(1)}(q^{-2}v;\boldsymbol{u}^{(2)})\Big)\Phi^{(1)}(\boldsymbol{u}^{(1)})$$

$$- \sum_{j=1}^{m_2}\frac{v/u_j^{(2)}}{v-u_j^{(2)}}\mathscr{B}^{(1)}(\boldsymbol{u}_{\sigma_j^{(2)},u_j^{(2)}\to v}^{(2)})\operatorname*{Res}_{w\to u_j^{(2)}}\Big(\tau^{+(1)}(w;\boldsymbol{u}_{\sigma_j^{(2)}}^{(2)})$$

$$+ f_{q^{-1}}(w;\boldsymbol{u}^{(2)})\mu^{-(1)}(w)\,\tau^{-(1)}(q^{-2}w;\boldsymbol{u}_{\sigma_j^{(2)}}^{(2)})\Big)\Phi^{(1)}(\boldsymbol{u}^{(1)}),$$

which, combined with Proposition 4.3, implies the claim for $\tau^{+(2)}(v)$.

We now repeat the same analysis for $\tau^{-(2)}(v)$. This time we focus on the "wanted" terms only. The exchange relations for $A_a^{-(1)}(v)$ and $D_a^{+(1)}(v)$ take the form

$$A_a^{-(1)}(v)\mathscr{b}_{\ddot{a}_i^2\ddot{a}_i^2}^{+-(1,1)}(u_i^{(2)}) = \mathscr{b}_{\ddot{a}_i^2\ddot{a}_i^2}^{+-(1,1)}(u_i^{(2)})\Big(A_a^{-(1)}(v)R_{a\ddot{a}_i^2}^{--(1,1)}(q^2v,u_i^{(2)})\Big) + UWT\,,$$

$$D_a^{+(1)}(v)\mathscr{b}_{\ddot{a}_i^2\ddot{a}_i^2}^{+-(1,1)}(u_i^{(2)}) = \mathscr{b}_{\ddot{a}_i^2\ddot{a}_i^2}^{+-(1,1)}(u_i^{(2)})\Big(f_{q^{-1}}(v,u_i^{(2)})R_{a\ddot{a}_i^2}^{++(1,1)}(v,u_i^{(2)})D_a^{+(1)}(v)\Big) + UWT\,,$$

where $UWT$ denote the remaining "unwanted" terms. Then, repeating the same steps as before, we find

$$\tau^{-(2)}(v)\Phi^{(2)}(\boldsymbol{u}^{(1,2)}) = \Big(\mathrm{tr}_a A_a^{-(1)}(v) + \mathrm{tr}_a D_a^{+(1)}(v)\Big)\mathscr{B}^{(1)}(\boldsymbol{u}^{(2)})\Phi^{(1)}(\boldsymbol{u}^{(1)})$$

$$= \mathscr{B}^{(1)}(\boldsymbol{u}^{(2)})\Big(\tau^{-(1)}(v;\boldsymbol{u}^{(2)})$$

$$+ f_{q^{-1}}(v;\boldsymbol{u}^{(2)})\mu^{+(1)}(v)\,\tau^{+(1)}(q^{-2}v;\boldsymbol{u}^{(2)})\Big)\Phi^{(1)}(\boldsymbol{u}^{(1)})$$

$$+ UWT\,.$$

Since $\tau^{-(2)}(v)$ and $\tau^{+(2)}(w)$ commute for any values of $v$ and $w$, we do not need to consider the unwanted terms. Proposition 4.3 then yields the eigenvalue of $\tau^{-(2)}(v)$. $\qquad\square$

We are finally ready to consider the $U_q(\mathfrak{so}_{2n+2})$-symmetric spin chains with $n \geq 3$. We define the level-$n$ transfer matrices in the usual way,

$$\tau^{\pm(n)}(v) := \mathrm{tr}_a\,T_a^{\pm(n)}(v)\,.$$

Then for each $1 \leq k \leq n-1$ we define the nested level-$k$ transfer matrices by

$$\tau^{\pm(k)}(v;\boldsymbol{u}^{(k+1\ldots n)}) := \mathrm{tr}_a\,T_a^{\pm(k)}(v;\boldsymbol{u}^{(k+1\ldots n)})\,,$$

$$\widetilde{\tau}^{\mp(k)}(v;\boldsymbol{u}^{(k+1\ldots n)}) := \mathrm{tr}_a\,\widetilde{T}_a^{\mp(k)}(v;\boldsymbol{u}^{(k+1\ldots n)})\,.$$

Let $\equiv$ denote equality of operators in the nested space $L^{(k)}$. Then we have that

$$\widetilde{\tau}^{\pm(k)}(v;\boldsymbol{u}^{(k+1\ldots n)}) \equiv \varepsilon^{(k+1)}\mu^{\pm(k)}(v;\boldsymbol{u}^{(k+2)})\,\tau^{\pm(k)}(q^{-2}v;\boldsymbol{u}^{(k+1\ldots n)})\,,$$

where $\mu^{\pm(k)}(v;\boldsymbol{u}^{(k+2)})$ is given by

|  | $L^V$ | $L^{+S}$ | $L^{-S}$ |
|---|---|---|---|
| $\mu^{+(n-1)}(v;\boldsymbol{u}^{(n+1)})$ | $f_q(v;\boldsymbol{\rho})$ | $1$ | $f_q(v;\boldsymbol{\rho})$ |
| $\mu^{-(n-1)}(v;\boldsymbol{u}^{(n+1)})$ | $f_q(v;\boldsymbol{\rho})$ | $f_q(v;\boldsymbol{\rho})$ | $1$ |
| $\mu^{+(k)}(v;\boldsymbol{u}^{(k+2)})$ | $f_q(v;\boldsymbol{u}^{(k+2)})$ | $f_q(v;\boldsymbol{u}^{(k+2)})$ | $f_q(v;\boldsymbol{\rho})f_q(v;\boldsymbol{u}^{(k+2)})$ |
| $\mu^{-(k)}(v;\boldsymbol{u}^{(k+2)})$ | $f_q(v;\boldsymbol{u}^{(k+2)})$ | $f_q(v;\boldsymbol{\rho})f_q(v;\boldsymbol{u}^{(k+2)})$ | $f_q(v;\boldsymbol{u}^{(k+2)})$ |

We extend the definition above to include the $k = 0$ case. The Theorem below is the second main result of this section.

**Theorem 4.5.** *The Bethe vector $\Phi^{(n)}(\boldsymbol{u}^{(1...n)})$ with $n \geq 3$ is an eigenvector of $\tau^{\pm(n)}(v)$ with the eigenvalue*

$$\Lambda^{\pm(n)}(v; \boldsymbol{u}^{(1...n)}) := \sum_{\boldsymbol{i}} f_q(q^{p_0(\boldsymbol{i})}v; \boldsymbol{u}^{(\mp s_0(\boldsymbol{i}))})$$

$$\times \prod_{j=1}^{n} \varepsilon_{i_j}^{(j)}\Big(\mu^{\pm s_j(\boldsymbol{i})(j-1)}(q^{p_j(\boldsymbol{i})}v; \boldsymbol{u}^{(j+1)}) f_{q^{-1}}(q^{p_j(\boldsymbol{i})}v; \boldsymbol{u}^{(j)})\Big)^{\frac{1}{2}(1+i_j)}, \quad (122)$$

*where $p_j(\boldsymbol{i}) = -\sum_{k=j+1}^{n}(1 + i_k)$ and $s_j(\boldsymbol{i}) = \text{sign}\big((-1)^{n-j-1}\prod_{k=j+1}^{n} i_k\big)$ provided*

$$\underset{v \to u_j^{(k)}}{\text{Res}} \Lambda^{\pm(n)}(v; \boldsymbol{u}^{(1...n)}) = 0 \quad \text{for} \quad 1 \leq k \leq n, \, 1 \leq j \leq m_k. \quad (123)$$

The explicit form of the Bethe equations of (123) with $n \geq 3$ is

$$\prod_{i=1}^{m_{\pm}} \frac{qu_j^{\pm} - q^{-1}u_i^{\pm}}{q^{-1}u_j^{\pm} - qu_i^{\pm}} \prod_{i=1}^{m_2} \frac{u_j^{\pm} - u_i^{(2)}}{qu_j^{\pm} - q^{-1}u_i^{(2)}} = -\varepsilon^{(1)}\lambda_{\pm}(u_j^{\pm}), \quad (124)$$

$$\prod_{i=1}^{m_+} \frac{q^{-1}u_j^{(2)} - qu_i^+}{u_j^{(2)} - u_i^+} \prod_{i=1}^{m_-} \frac{q^{-1}u_j^{(2)} - qu_i^-}{u_j^{(2)} - u_i^-} \prod_{i=1}^{m_2} \frac{qu_j^{(2)} - q^{-1}u_i^{(2)}}{q^{-1}u_j^{(2)} - qu_i^{(2)}} \prod_{i=1}^{m_3} \frac{u_j^{(2)} - u_i^{(3)}}{qu_j^{(2)} - q^{-1}u_i^{(3)}} = -\frac{\varepsilon^{(2)}}{\varepsilon^{(1)}}, \quad (125)$$

$$\prod_{i=1}^{m_{k-1}} \frac{q^{-1}u_j^{(k)} - qu_i^{(k-1)}}{u_j^{(k)} - u_i^{(k-1)}} \prod_{i=1}^{m_k} \frac{qu_j^{(k)} - q^{-1}u_i^{(k)}}{q^{-1}u_j^{(k)} - qu_i^{(k)}} \prod_{i=1}^{m_{k+1}} \frac{u_j^{(k)} - u_i^{(k+1)}}{qu_j^{(k)} - q^{-1}u_i^{(k+1)}} = -\frac{\varepsilon^{(k)}}{\varepsilon^{(k-1)}}, \quad (126)$$

$$\prod_{i=1}^{m_{n-1}} \frac{q^{-1}u_j^{(n)} - qu_i^{(n-1)}}{u_j^{(n)} - u_i^{(n-1)}} \prod_{i=1}^{m_n} \frac{qu_j^{(n)} - q^{-1}u_i^{(n)}}{q^{-1}u_j^{(n)} - qu_i^{(n)}} = -\frac{\varepsilon^{(n)}}{\varepsilon^{(n-1)}}\lambda_n(u_j^{(n)}), \quad (127)$$

where $\lambda_n$ is given by $\lambda_n(v) = f_q(v; \boldsymbol{\rho})$ or 1 when $L^{(n)} = L^V$ or $L^{\pm S}$, respectively.

*Proof of Theorem 4.5.* The proof is very similar to that of Theorem 3.3. We begin by focusing on $\tau^{+(n)}(v)$ and rewriting the corresponding "AB" and "DB" exchange relations in a more convenient form. From Lemma 2.18 we deduce that

$$R_{21}^{\pm+(n-1,n-1)}(u, v) = \frac{(R_{12}^{\pm[+](n-1,n-1)}(q^{2n-2}v, u))^{u_2}}{h^{\pm((n-1)/2)}(v, u)},$$

where $[+] = +/-$ if $n-1$ is odd/even. Combining these identities with (9), (72), (73) and (108) yields the wanted "AB" and "DB" exchange relations:

$$A_a^{+(n-1)}(v)\,\mathscr{b}_{\dot{a}_i^n \ddot{a}_i^n}^{[+]-(n-1,n-1)}(u_i^{(n)})$$

$$= \mathscr{b}_{\dot{a}_i^n \ddot{a}_i^n}^{[+]-(n-1,n-1)}(u_i^{(n)})\Bigg(\frac{f_q(v, u_i^{(n)})}{h^{+((n-1)/2)}(v, u_i^{(n)})}$$

$$\times R_{a\dot{a}_i^n}^{+[+](n-1,n-1)}(q^{2n-2}v, u_i^{(n)}) A_a^{+(n-1)}(v) R_{a\ddot{a}_i^n}^{+-(n-1,n-1)}(q^2 v, u_i^{(n)})\Bigg)$$

$$- \frac{v/u_i^{(n)}}{v - u_i^{(n)}} \mathscr{b}_{\dot{a}_i^n \ddot{a}_i^n}^{[+]-(n-1,n-1)}(v) \underset{w \to u_i^{(n)}}{\text{Res}} \Bigg(\frac{f_q(w, u_i^{(n)})}{h^{+((n-1)/2)}(w, u_i^{(n)})}$$

$$\times R_{a\dot{a}_i^n}^{+[+](n-1,n-1)}(q^{2n-2}w, u_i^{(n)}) A_a^{+(n-1)}(w) R_{a\ddot{a}_i^n}^{+-(n-1,n-1)}(q^2 w, u_i^{(n)})\Bigg), \quad (128)$$

$$D_a^{-(n-1)}(v)\,\ell_{\dot{a}_i^n\ddot{a}_i^n}^{[+]-(n-1)}(u_i^{(n)})$$

$$= \ell_{\dot{a}_i^n\ddot{a}_i^n}^{[+]-(n-1,n-1)}(u_i^{(n)})\left(\frac{f_{q^{-1}}(v,u_i^{(n)})}{h^{-((n-1)/2)}(q^{-2}v,u_i^{(n)})}\right.$$

$$\times R_{a\dot{a}_i^n}^{-[+](n-1,n-1)}(q^{2n-4}v,u_i^{(n)})\,D_a^{-(n-1)}(v)\,R_{a\ddot{a}_i^n}^{--(n-1,n-1)}(v,u_i^{(n)})\bigg)$$

$$-\frac{v/u_i^{(n)}}{v-u_i^{(n)}}\,\ell_{\dot{a}_i^n\ddot{a}_i^n}^{[+]-(n-1,n-1)}(v)\,\operatorname*{Res}_{w\to u_i^{(n)}}\left(\frac{f_{q^{-1}}(w,u_i^{(n)})}{h^{-((n-1)/2)}(q^{-2}w,u_i^{(n)})}\right.$$

$$\times R_{a\dot{a}_i^n}^{-[+](n-1,n-1)}(q^{2n-4}w,u_i^{(n)})\,D_a^{-(n-1)}(w)\,R_{a\ddot{a}_i^n}^{--(n-1,n-1)}(w,u_i^{(n)})\bigg). \qquad (129)$$

Inspired by the exchange relations above we define barred transfer matrices

$$\overline{\tau}^{+(n-1)}(v;\boldsymbol{u}^{(n)}) := \frac{f_q(v;\boldsymbol{u}^{(n)})}{h^{+((n-1)/2)}(v;\boldsymbol{u}^{(n)})}\,\mathrm{tr}_a\bigg(A_a^{+(n-1)}(v)\prod_{i=1}^{m_n}R_{a\ddot{a}_i^n}^{+-(n-1,n-1)}(q^2v,u_i^{(n)})$$

$$\times \prod_{i=m_n}^{1}R_{a\dot{a}_i^n}^{+[+](n-1,n-1)}(q^{2n-2}v,u_i^{(n)})\bigg),$$

$$\overline{\tau}^{-(n-1)}(v;\boldsymbol{u}^{(n)}) := \frac{f_{q^{-1}}(v;\boldsymbol{u}^{(n)})}{h^{-((n-1)/2)}(q^{-2}v;\boldsymbol{u}^{(n)})}\,\mathrm{tr}_a\bigg(D_a^{-(n-1)}(v)\prod_{i=1}^{m_n}R_{a\ddot{a}_i^n}^{--(n-1,n-1)}(v,u_i^{(n)})$$

$$\times \prod_{i=m_n}^{1}R_{a\dot{a}_i^n}^{-[+](n-1,n-1)}(q^{2n-4}v,u_i^{(n)})\bigg),$$

which differ from $\tau^{\pm(n-1)}(v;\boldsymbol{u}^{(n)})$ in (103) and (104) by the ordering of $R$-matrices. The ordering can be amended with the help of operator $X^{(n-1)} := \prod_{i-1}^{m_n-1}X_i^{(n-1)}$ where

$$X_i^{(n-1)} := \prod_{j=i+1}^{m_n}R_{\dot{a}_j^n\dot{a}_i^n}^{[+,+](n-1,n-1)}(u_j^{(n)},u_i^{(n)})\prod_{j=m_n}^{i+1}R_{\ddot{a}_j^n\dot{a}_i^n}^{-[+](n-1,n-1)}(q^{2n-4}u_j^{(n)},u_i^{(n)}).$$

In particular, $\overline{\tau}^{\pm(n-1)}(v;\boldsymbol{u}^{(n)}) = X^{(n-1)}\tau^{\pm(n-1)}(v;\boldsymbol{u}^{(n)})(X^{(n-1)})^{-1}$ and each $X_i^{(n-1)}$ acts as a scalar operator on $\Phi^{(n-1)}(\boldsymbol{u}^{(1\ldots n-1)};\boldsymbol{u}^{(n)})$. Therefore

$$\tau^{+(n)}(v)\Phi^{(n)}(\boldsymbol{u}^{(1\ldots n)}) = \mathcal{B}^{(n-1)}(\boldsymbol{u}^{(n)})X^{(n-1)}\,\tau^{+(n)}(v;\boldsymbol{u}^{(n)})(X^{(n-1)})^{-1}\Phi^{(n-1)}(\boldsymbol{u}^{(1\ldots n-1)};\boldsymbol{u}^{(n)})$$

$$-\sum_{j=1}^{m_n}\frac{v/u_j^{(n)}}{v-u_j^{(n)}}\,\mathcal{B}^{(n-1)}(\boldsymbol{u}_{\sigma_j^{(n)},u_j^{(n)}\to v}^{(n)})X^{(n-1)}$$

$$\times \operatorname*{Res}_{w\to u_j^{(n)}}\tau^{+(n)}(w;\boldsymbol{u}_{\sigma_j^{(n)}}^{(n)})(X^{(n-1)})^{-1}\Phi^{(n-1)}(\boldsymbol{u}^{(1\ldots n-1)};\boldsymbol{u}_{\sigma_j^{(n)}}^{(n)}),$$

where

$$\tau^{+(n)}(v;\boldsymbol{u}^{(n)}) := \tau^{+(n-1)}(v;\boldsymbol{u}^{(n)}) + \varepsilon^{(n)}f_{q^{-1}}(v,\boldsymbol{u}^{(n)})\mu^{-(n-1)}(v;\boldsymbol{\rho})\,\tau^{-(n-1)}(q^{-2}v;\boldsymbol{u}^{(n)}).$$

We now repeat the same analysis for $\tau^{-(n)}(v)$. This time we focus on the "wanted" terms only.

The relevant exchange relations are now

$$A_a^{-(n-1)}(v)\,\mathscr{b}_{\dot{a}_i^n\ddot{a}_i^n}^{[+]-(n-1)}(u_i^{(n)})$$

$$= \mathscr{b}_{\dot{a}_i^n\ddot{a}_i^n}^{[+]-(n-1)}(u_i^{(n)})\Bigg(\frac{1}{h^{-((n-1)/2)}(v,u_i^{(n)})}$$

$$\times R_{a\dot{a}_i^n}^{-[+](n-1,n-1)}(q^{2n-2}v,u_i^{(n)})A_a^{-(n-1)}(v)R_{a\ddot{a}_i^n}^{--(n-1,n-1)}(q^2v,u_i^{(n)})\Bigg)+UWT\,,$$

$$D_a^{+(n-1)}(v)\,\mathscr{b}_{\dot{a}_i^n\ddot{a}_i^n}^{[+]-(n-1)}(u_i^{(n)})$$

$$= \mathscr{b}_{\dot{a}_i^n\ddot{a}_i^n}^{[+]-(n-1)}(u_i^{(n)})\Bigg(\frac{1}{h^{+((n-1)/2)}(q^{-2}v,u_i^{(n)})}$$

$$\times R_{a\dot{a}_i^n}^{+[+](n-1,n-1)}(q^{2n-4}v,u_i^{(n)})D_a^{+(n-1)}(v)R_{a\ddot{a}_i^n}^{+-(n-1,n-1)}(v,u_i^{(n)})\Bigg)+UWT\,.$$

Repeating the same steps as above we obtain

$$\tau^{-(n)}(v)\Phi^{(n)}(\boldsymbol{u}^{(1\ldots n)}) = \mathscr{B}^{(n-1)}(\boldsymbol{u}^{(n)})X^{(n-1)}\,\tau^{-(n)}(v;\boldsymbol{u}^{(n)})(X^{(n-1)})^{-1}$$
$$\times\Phi^{(n-1)}(\boldsymbol{u}^{(1\ldots n-1)};\boldsymbol{u}^{(n)})+UWT\,, \qquad (130)$$

where

$$\tau^{-(n)}(v;\boldsymbol{u}^{(n)}) := \tau^{-(n-1)}(v;\boldsymbol{u}^{(n)})+\varepsilon^{(n)}f_{q^{-1}}(v,\boldsymbol{u}^{(n)})\mu^{+(n-1)}(v;\boldsymbol{\rho})\,\tau^{+(n-1)}(q^{-2}v;\boldsymbol{u}^{(n)})\,.$$

Since $\tau^{-(n)}(v)$ and $\tau^{+(n)}(w)$ commute for any values of $v$ and $w$, we do not need to consider the unwanted terms in (130). It remains to repeat the same analysis down the nesting by taking into account (107) together with the fact that $\Phi^{(k)}(\boldsymbol{u}^{(1\ldots k)};\boldsymbol{u}^{(k+1\ldots n)})\in L^{(k)}$, and use Proposition 4.3 (with slight amendments). This gives a recurrence relation, for $2\le k\le n$,

$$\Lambda^{\pm(k)}(v;\boldsymbol{u}^{(1\ldots k)};\boldsymbol{u}^{(k+1\ldots n)}) := \varepsilon_{-1}^{(k)}\Lambda^{\pm(k-1)}(v;\boldsymbol{u}^{(1\ldots k-1)};\boldsymbol{u}^{(k\ldots n)})$$
$$+\varepsilon_{+1}^{(k)}f_{q^{-1}}(v,\boldsymbol{u}^{(k)})\mu^{\mp(k-1)}(v;\boldsymbol{u}^{(k+1)})$$
$$\times\Lambda^{\mp(k-1)}(q^{-2}v;\boldsymbol{u}^{(1\ldots k-1)};\boldsymbol{u}^{(k\ldots n)})\,,$$

with $\Lambda^{\pm(1)}$ given by (116). Upon solving this recurrence relation we recover the claim of the Theorem. $\qquad\square$

*Remark* 4.6. Let $a_{ij}$ denote matrix entries of a connected Dynkin diagram of type $B_n$ or $D_n$ and let $I$ denote the set of its nodes. Then put $d_\pm = d_2 = \ldots = d_n = 1$ for $D_{n+1}$ and $2d_1 = d_2 = \ldots = d_n = 2$ for $B_n$. Upon substituting $u_j^{(k)}\to q^{\tilde{d}_k}z_j^{(k)}$, where $\tilde{d}_k = \sum_{i=1}^k d_i$ with $d_1 = d_\pm$ for $D_{n+1}$, Bethe equations (95)–(97) and (124)–(127) can be written as

$$\prod_{l\in I}\prod_{i=1}^{m_l}\frac{q^{d_k a_{kl}}z_j^{(k)}-z_i^{(l)}}{z_j^{(k)}-q^{d_k a_{kl}}z_i^{(l)}} = -\frac{\varepsilon^{(k)}}{\varepsilon^{(k-1)}}\,\lambda_k(q^{\tilde{d}_k}z_j^{(k)})\,,$$

for all $k\in I$ and all allowed $j$. Here $\varepsilon^{(0)} = 1$ and $\lambda_k(q^{\tilde{d}_k}z_j^{(k)}) = 1$ when $k\notin\{\pm,1,n\}$.

## 5 Conclusions and Outlook

The results of this paper are two-fold. First, we proposed a new construction of $q$-deformed $\mathfrak{so}_{2n+1}$- and $\mathfrak{so}_{2n}$-invariant spinor-vector and spinor-spinor $R$-matrices in terms of supermatrices and found explicit recurrence relations. We believe these results will be of interest on their own right. For instance, this opens a door to study spectral properties of open spin chains with spinor-type transfer matrices thus complementing the results obtained by Artz, Mezincescu and Nepomechie in [16]. Second, we solved the long-standing problem of diagonalizing transfer matrices that obey quadratic relations defined by the aforementioned $q$-deformed spinor-spinor $R$-matrices. The corresponding Bethe ansatz equations were already known since they can be determined from the Cartan datum only. The constructed Bethe vectors and the corresponding eigenvalues are new results. A natural next step is to find recursion relations for these Bethe vectors and investigate scalar products in the spirit of the approach put forward by Hutsalyuk et. al. in [17]. Moreover, it would be interesting to construct $q$-deformed spinor-oscillator $R$-matrices and investigate the spinor-type QQ-system following the steps of Ferrando, Frassek and Kazakov in [8]. Lastly, we believe this work might help to better undertstand the Bethe ansazt for fishnets and fishchains emerging in the AdS/CFT integrability framework, see [18–21] and references therein.

## Acknowledgements

The author thanks Rouven Frassek, Allan Gerrard and Eric Ragoucy for useful discussions and comments.

**Funding information.**  This research was supported by the European Social Fund under Grant No. 09.3.3-LMT-K-712-01-0051.

## A  The semi-classical limit

### A.1  $U_{q^2}(\mathfrak{so}_{2n+1})$-symmetric spin chains

The semi-classical limit is obtained by setting $v = \exp(2y\hbar)$, $u_j^{(k)} = \exp(2x_j^{(k)}\hbar)$, $q = \exp(\hbar/2)$, and carefully taking the $\hbar \to 0$ limit. Introduce a rational function

$$f_k(y,x) = \frac{y - x + k}{y - x}.$$

The eigenvalue (93) then becomes

$$\Lambda^{(n)}(y; \boldsymbol{x}^{(1\ldots n)}) := \sum_{\boldsymbol{i}} f_{1/2}(y + p_0(\boldsymbol{i}); \boldsymbol{x}^{(1)})$$

$$\times \prod_{j=1}^{n} \varepsilon_{i_j}^{(j)} \left( \mu^{(j-1)}(y + p_j(\boldsymbol{i}); \boldsymbol{x}^{(j+1)}) f_{-1}(y + p_j(\boldsymbol{i}); \boldsymbol{x}^{(j)}) \right)^{\frac{1}{2}(1+i_j)},$$

where $p_j(\boldsymbol{i}) = -\sum_{k=j+1}^{n}(1 + i_k)$ and $\mu^{(k)}(y; \boldsymbol{x}^{(k+2)})$ are given by

| | $L^V$ | $L^S$ |
|---|---|---|
| $\mu^{(n-1)}(y; \boldsymbol{x}^{(n+1)})$ | $f_1(y; \boldsymbol{\rho})$ | $f_{1/2}(y; \boldsymbol{\rho})$ |
| $\mu^{(k)}(y; \boldsymbol{x}^{(k+2)})$ | $f_1(y; \boldsymbol{x}^{(k+2)})$ | $f_{1/2}(y; \boldsymbol{\rho}) f_1(y; \boldsymbol{x}^{(k+2)})$ |

The Bethe equations (95)–(97) become

$$\prod_{i=1}^{m_1} \frac{x_j^{(1)} - x_i^{(1)} + \frac{1}{2}}{x_j^{(1)} - x_i^{(1)} - \frac{1}{2}} \prod_{i=1}^{m_2} \frac{x_j^{(1)} - x_i^{(2)}}{x_j^{(1)} - x_i^{(2)} + 1} = -\varepsilon^{(1)} \lambda_1(x_j^{(1)}),$$

$$\prod_{i=1}^{m_{k-1}} \frac{x_j^{(k)} - x_i^{(k-1)} - 1}{x_j^{(k)} - x_i^{(k-1)}} \prod_{i=1}^{m_k} \frac{x_j^{(k)} - x_i^{(k)} + 1}{x_j^{(k)} - x_i^{(k)} - 1} \prod_{i=1}^{m_{k+1}} \frac{x_j^{(k)} - x_i^{(k+1)}}{x_j^{(k)} - x_i^{(k+1)} + 1} = -\frac{\varepsilon^{(k)}}{\varepsilon^{(k-1)}},$$

$$\prod_{i=1}^{m_{n-1}} \frac{x_j^{(n)} - x_i^{(n-1)} - 1}{x_j^{(n)} - x_i^{(n-1)}} \prod_{i=1}^{m_n} \frac{x_j^{(n)} - x_i^{(n)} + 1}{x_j^{(n)} - x_i^{(n)} - 1} = -\frac{\varepsilon^{(n)}}{\varepsilon^{(n-1)}} \lambda_n(x_j^{(n)}),$$

where $\lambda_1(y) = 1$ or $f_{1/2}(y; \boldsymbol{\rho})$ and $\lambda_n(y) = f_1(y; \boldsymbol{\rho})$ or $1$ when $L^{(n)} = L^V$ or $L^S$, respectively.

## A.2 $U_q(\mathfrak{so}_6)$-symmetric spin chain

The semi-classical limit is obtained in the same way as before, except that we set $q = \exp(\hbar)$. The eigenvalue (118) becomes

$$\Lambda^{\pm(2)}(y; \boldsymbol{x}^{(1,2)}) := \varepsilon_{-1}^{(2)}\Big(\varepsilon_{-1}^{(1)} f_1(y; \boldsymbol{x}^\pm) + \varepsilon_{+1}^{(1)} f_1(y; \boldsymbol{x}^{(2)}) f_{-1}(y; \boldsymbol{x}^\pm) \lambda_\pm(y)\Big)$$
$$+ \varepsilon_{+1}^{(2)} \mu^{\mp(1)}(y)\Big(\varepsilon_{-1}^{(1)} f_{-1}(y; \boldsymbol{x}^{(2)}) f_1(y-1; \boldsymbol{x}^\mp)$$
$$+ \varepsilon_{+1}^{(1)} f_{-1}(y-1; \boldsymbol{x}^\mp) \lambda_\mp(y-1)\Big),$$

where $\mu^{\pm(1)}(y)$ and $\lambda_\pm(y)$ are given by

|  | $L^V$ | $L^{+S}$ | $L^{-S}$ |
|---|---|---|---|
| $\mu^{+(1)}(y)$ | $f_1(y; \boldsymbol{\rho})$ | $1$ | $f_1(y; \boldsymbol{\rho})$ |
| $\mu^{-(1)}(y)$ | $f_1(y; \boldsymbol{\rho})$ | $f_1(y; \boldsymbol{\rho})$ | $1$ |
| $\lambda_+(y)$ | $1$ | $f_1(y; \boldsymbol{\rho})$ | $1$ |
| $\lambda_-(y)$ | $1$ | $1$ | $f_1(y; \boldsymbol{\rho})$ |

The Bethe equations (120)–(121) become

$$\prod_{i=1}^{m_\pm} \frac{x_j^\pm - x_i^\pm + 1}{x_j^\pm - x_i^\pm - 1} \prod_{i=1}^{m_2} \frac{x_j^\pm - x_i^{(2)}}{x_j^\pm - x_i^{(2)} + 1} = -\varepsilon^{(1)} \lambda_\pm(x_j^\pm),$$

$$\prod_{i=1}^{m_+} \frac{x_j^{(2)} - x_i^+ - 1}{x_j^{(2)} - x_i^+} \prod_{i=1}^{m_-} \frac{x_j^{(2)} - x_i^- - 1}{x_j^{(2)} - x_i^-} \prod_{i=1}^{m_2} \frac{x_j^{(2)} - x_i^{(2)} + 1}{x_j^{(2)} - x_i^{(2)} - 1} = -\frac{\varepsilon^{(2)}}{\varepsilon^{(1)}} \lambda_2(x_j^{(2)}),$$

where $\lambda_2$ is given by $\lambda_2(y) = f_1(y; \boldsymbol{\rho})$ or $1$ when $L^{(2)} = L^V$ or $L^{\pm S}$, respectively.

## A.3 $U_q(\mathfrak{so}_{2n+2})$-symmetric spin chains

By the same arguments as above, the eigenvalue (122) becomes

$$\Lambda^{\pm(n)}(y; \boldsymbol{x}^{(1\ldots n)}) := \sum_{\boldsymbol{i}} f_1(y + p_0(\boldsymbol{i}); \boldsymbol{u}^{(\mp s_0(\boldsymbol{i}))})$$

$$\times \prod_{j=1}^{n} \varepsilon_{i_j}^{(j)} \Big(\mu^{\pm s_j(\boldsymbol{i})(j-1)}(y + p_j(\boldsymbol{i}); \boldsymbol{u}^{(j+1)}) f_{-1}(y + p_j(\boldsymbol{i}); \boldsymbol{u}^{(j)})\Big)^{\frac{1}{2}(1+i_j)},$$

where $p_j(\boldsymbol{i}) = -\frac{1}{2}\sum_{k=j+1}^{n}(1+i_k)$, $s_j(\boldsymbol{i}) = \mathrm{sign}\left((-1)^{n-j-1}\prod_{k=j+1}^{n}i_k\right)$ and $\mu^{\pm(k)}(y;\boldsymbol{x}^{(k+2)})$ are given by

| | $L^V$ | $L^{+S}$ | $L^{-S}$ |
|---|---|---|---|
| $\mu^{+(n-1)}(y;\boldsymbol{\rho})$ | $f_1(y;\boldsymbol{\rho})$ | $1$ | $f_1(y;\boldsymbol{\rho})$ |
| $\mu^{-(n-1)}(y;\boldsymbol{\rho})$ | $f_1(y;\boldsymbol{\rho})$ | $f_1(y;\boldsymbol{\rho})$ | $1$ |
| $\mu^{+(k)}(y;\boldsymbol{x}^{(k+2)})$ | $f_1(y;\boldsymbol{x}^{(k+2)})$ | $f_1(y;\boldsymbol{x}^{(k+2)})$ | $f_1(y;\boldsymbol{\rho})f_1(y;\boldsymbol{x}^{(k+2)})$ |
| $\mu^{-(k)}(y;\boldsymbol{x}^{(k+2)})$ | $f_1(y;\boldsymbol{u}^{(k+2)})$ | $f_1(y;\boldsymbol{\rho})f_1(y;\boldsymbol{x}^{(k+2)})$ | $f_1(y;\boldsymbol{x}^{(k+2)})$ |

The Bethe equations (124)–(127) become

$$\prod_{i=1}^{m_\pm}\frac{x_j^\pm - x_i^\pm + 1}{x_j^\pm - x_i^\pm - 1}\prod_{i=1}^{m_2}\frac{x_j^\pm - x_i^{(2)}}{x_j^\pm - x_i^{(2)} + 1} = -\varepsilon^{(1)}\lambda_\pm(x_j^\pm),$$

$$\prod_{i=1}^{m_+}\frac{x_j^{(2)} - x_i^+ - 1}{x_j^{(2)} - x_i^+}\prod_{i=1}^{m_-}\frac{x_j^{(2)} - x_i^- - 1}{x_j^{(2)} - x_i^-}\prod_{i=1}^{m_2}\frac{x_j^{(2)} - x_i^{(2)} + 1}{x_j^{(2)} - x_i^{(2)} - 1}\prod_{i=1}^{m_3}\frac{x_j^{(2)} - x_i^{(3)}}{x_j^{(2)} - x_i^{(3)} + 1} = -\frac{\varepsilon^{(2)}}{\varepsilon^{(1)}},$$

$$\prod_{i=1}^{m_{k-1}}\frac{x_j^{(k)} - x_i^{(k-1)} - 1}{x_j^{(k)} - x_i^{(k-1)}}\prod_{i=1}^{m_k}\frac{x_j^{(k)} - x_i^{(k)} + 1}{x_j^{(k)} - x_i^{(k)} - 1}\prod_{i=1}^{m_{k+1}}\frac{x_j^{(k)} - x_i^{(k+1)}}{x_j^{(k)} - x_i^{(k+1)} + 1} = -\frac{\varepsilon^{(k)}}{\varepsilon^{(k-1)}},$$

$$\prod_{i=1}^{m_{n-1}}\frac{x_j^{(n)} - x_i^{(n-1)} - 1}{x_j^{(n)} - x_i^{(n-1)}}\prod_{i=1}^{m_n}\frac{x_j^{(n)} - x_i^{(n)} + 1}{x_j^{(n)} - x_i^{(n)} - 1} = -\frac{\varepsilon^{(n)}}{\varepsilon^{(n-1)}}\lambda_n(x_j^{(n)}),$$

where $\lambda_n$ is given by $\lambda_n(v) = f_1(y;\boldsymbol{\rho})$ or $1$ when $L^{(n)} = L^V$ or $L^{\pm S}$, respectively.

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
