# Peer review of "Algebraic Bethe Ansatz for spinor R-matrices"

_SciPost Physics, doi:SciPost Phys. 12, 067 (2022)_

## Round 1 · Referee Report · Anonymous (Referee 1) · 2021-9-5

Report

I think that the paper "Algebraic Bethe Ansatz for spinor R-matrices" meets the requirements of the SciPost Physics Journal.

Attachment

  • validity: high
  • significance: good
  • originality: high
  • clarity: good
  • formatting: good
  • grammar: good

Author:  Vidas Regelskis  on 2021-11-01  [id 1898]

(in reply to Report 1 on 2021-09-05)
Category:
answer to question

I thank the referee for the point they have raised. I have provided an answer the attached pdf.

Attachment:

referee_1.pdf

---

## Round 1 · Referee Report · Anonymous (Referee 2) · 2021-10-12

Strengths

1- Gives a new formula for the Bethe vectors of such models

2- All calculations are well described

Report

I do recommend the publication of the paper.

Requested changes

The author should precise the way he move the twist matrix trough the R matrix in the proof of Theorem 3.3 (from 3.22 and 3.23 to the nested transfert matrix in the next equation) and in the proof of Theorem 4.4 (from 4.29 and 4.30 to the nested transfert matrices in the next equation).

  • validity: high
  • significance: high
  • originality: good
  • clarity: high
  • formatting: -
  • grammar: -

Author:  Vidas Regelskis  on 2021-11-01  [id 1899]

(in reply to Report 2 on 2021-10-12)
Category:
answer to question

I thank the referee for raising the point about way the twist matrices are moved through the R-matrices in the proof of Theorem 3.3 and Theorem 4.4. This was indeed overlooked in the first version paper. This issue has been fixed. The twist matrices are now included in the definition of the monodromy matrices, given by eqs. (3.2) and (4.3). (Tables on pages 19 and 26 were updated accordingly.) This is a valid construction since the twist matrices satisfy the fundamental exchange relations, (2.65) and (2.70). This yields the wanted results without needing to deal with the twist matrices explicitly.

---

## Round 1 · Referee Report · Anonymous (Referee 3) · 2021-10-25

Strengths

1- New construction of R-matrices (of type spinor-vector and spinor-spinor) for the quantum algebras $U_q(\widehat{so_N})$ using the formalism of super-matrices. 2- Calculation of the the transfer matrices of closed spin-chains based on these algebras 3- Explicit diagonalization by the algebraic Bethe ansatz (which is new) 4- Clearly written

Weaknesses

1- Rather technical

Report

The paper is interesting and clearly written, despite its technicality.
I think it is worth publishing

Requested changes

1- I find surprising that the $R$-matrices corresponding to $N$ even have a deformation parameter $q^2$, while the ones corresponding to $N$ odd have a deformation parameter $q$. There should be an explanation for this point. In particular, it seems to me that removing the index 0 from the construction for $so(2n+2)$, one should get the construction for $so(2n+1)$. Then, how the change in the deformation parameter occurs? 2-It would be nice to present also a super-matrix construction of the vector-vector $R$-matrix. In particular, the fusion of spinor-vector $R$-matrices should go back to the vector-vector $R$-matrix in a super-matrix form. 3-Since the presentation is quite technical, it is better not to use the same notation for different objects. -- It is the case for the transposition $w$, defined in eq. 2.3 and differently on eq. 2.6. Then, in eq. 2.55, one does not know which transposition is used. On the transposition, the one introduced eq. 2.30 should be related to the ones already introduced. -- In the same way, the operators in eq. 3.9 have nothing to do with the function $\beta$ used in eq. 2.36. A variation on the letter b (e.g. the mathfrak or mathbb styles, or any other style) would be better. -- The notation used in eq. following line 138 is ambiguous: from what I understand $x_j^{m}$ is not $x_j$ to some power, but rather $x_j^{(m)}$ (which would be more consistent with the notation used elsewhere). 4-In the same spirit, some points should be precised to ease the reading of the manuscript: -- When writing products of non-commuting operators, it should be indicated that the product is from left to right starting from the bottom index to the upper one in the product sign. This is particularly important for instance in eq. 3.8 (and should be reminded here), but it should be already stated in eq. 3.2 -- If I understand correctly, the notation $(\dot{a}\ddot{a})_n^j$ in eq. 3.10 means $\dot{a}_n^j\,\ddot{a}_n^j$. It should be stated explicitly. -- I think also that it would be fair to indicate in the title that the article deals with $U_q(\widehat{so_N})$ algebras, but I leave it to the author.

  • validity: -
  • significance: -
  • originality: -
  • clarity: -
  • formatting: -
  • grammar: -

Author:  Vidas Regelskis  on 2021-11-01  [id 1900]

(in reply to Report 3 on 2021-10-25)

I thank the referee for the points they have raised. I have provided an answer in the attached pdf.

Attachment:

referee_3.pdf

---

## Round 2 · Referee Report · Anonymous · 2021-11-2

Report
I do think that the new version is good for publication.

---

## Round 2 · Referee Report · Anonymous · 2021-11-2

Report
After author's corrections I think that the paper can be published in SciPost Physics.
Vidas Regelskis on 2021-11-02 [id 1903]
Dear Editor,
I would like to thank the referees for carefully reading the manuscript and for their useful comments and suggestions. I have corrected the typos and implemented the suggested improvements.
Response to Referee 1
I thank the referee for raising the point about the possible contradiction between the commutation relations of fundamental $L$-operators and their coproduct rule. Recall that the universal $R$-matrix $\mathcal{R}$ is an element in a completion of $U_q(\mathfrak{b}_+) \otimes U_q(\mathfrak{b}_-)$ where $\mathfrak{b}_\pm$ are the standard Borel subalgebras. There is a number of ways of defining the $L$-operators consistently. For instance,
\[ L^+(u) = (\pi_u \otimes \text{id})\, \mathcal{R}^{-1} , \qquad L^-(u) = (\text{id} \otimes \pi_u)\, \mathcal{R} , \qquad R(u,v) = (\pi_u \otimes \pi_v)\, \mathcal{R}^{-1} \]gives the wanted defining relations and the wanted coproduct rule,
\[ \Delta( L^\pm_{ij}(u) ) = \sum_{k} L^\pm_{ik}(u) \otimes L^\pm_{kj}(u). \]For $U_q(\widehat{\mathfrak{gl}}_N)$ this was explicitly demonstrated by J. Ding and I. B. Frenkel in Comm. Math. Phys. 156, 277-300 (1993) and by E. Frenkel and E. Mukhin in Sel. Math. 8, 537-635 (2002). For $U_q(\widehat{\mathfrak{so}}_N)$ this could be verified using the isomorphism constructed recently by N. Jing, M. Liu and A. Molev in SIGMA 16, 043, (2020).
Response to Referee 2
I thank the referee for raising the point about way the twist matrices are moved through the $R$-matrices in the proof of Theorem 3.3 and Theorem 4.4. This was indeed overlooked in the first version paper. This issue has been fixed. The twist matrices are now included in the definition of the monodromy matrices, given by eqs. (3.2) and (4.3). (Tables on pages 19 and 26 were updated accordingly.) This is a valid construction since the twist matrices satisfy the fundamental exchange relations, (2.65) and (2.70). This yields the wanted results without needing to deal with the twist matrices explicitly.
Response to Referee 3
I thank the referee for the points they have raised. Below I list my responses and the changes I have made:
1.1. The deformation parameter of $U_{q^2}(\mathfrak{so}_{2n+1})$ is set to $q^2$ to avoid having $\sqrt{q}$ in the spinor-spinor $R$-matrix and the corresponding exchange relations. The square root of the deformation parameter arises because the root system of $\mathfrak{so}_{2n+1}$ has a simple short root. (This is not the case for $\mathfrak{so}_{2n+2}$ since its simple roots are all of the same length.) This explanation was added to the Introduction (page 2, line 45) and Section 2.5 (page 8, line 176).
2.1. It is indeed possible to obtain the vector-vector $R$-matrix by fusing spinor-vector $R$-matrices. However this construction is not needed for the goals of this paper, hence is not included.
3.1. The $q$-transposition defined by (2.6-2.7) and all instances of it were renamed to a new symbol.
3.2. The ambiguous notation of the creation operators was fixed. The repeated symbol $\beta$ was replaced by $\mathscr{b}$.
3.3. The notation below line 140 on page 6 is correct. Here $x_j^{m_j}$ with $m_j = 0, 1$ denote elements of the exterior algebra $\Lambda$ defined in line 138. In particular, $x_j^0 = 1$ and $x_j^1=x_j$.
3.4. The notation in equation (2.30) is correct. Here $\omega_i$ is an element of the deformed Clifford algebra $\mathscr{C}^n_q$.
4.1. I have added an explanation of the product notation below line 109 on page 5.
4.2. The ambiguous notation $(\dot a \ddot a)^j_n$ was replaced by $a^j_n$. An explanation of this notation was added at the beginning of Section 3.1 on page 18 and at the beginning of Section 4.1 on page 25.
4.3. The algebras $U_{q^2}(\mathfrak{so}_{2n+1})$ and $U_q(\mathfrak{so}_{2n})$ are explicitly mentioned in the Abstract. I have not included them in the title to avoid having mathematical symbols and thus help the search engines to index the paper.
Kind regards, Vidas Regelskis

---

## Round 2 · Referee Report · Anonymous · 2021-12-3

Report
I think the new version is now ready to be published

---

## Round 2 · Author Response

I would like to thank the referees for carefully reading the manuscript and for their useful comments and suggestions. I have corrected the typos and implemented the suggested improvements.

---

## Round 2 · List of Changes

The changes made are listed in the replies to the referee reports.

---

## Editorial Decision

published